# A new AMPK isoform mediates glucose-restriction induced longevity non-cell autonomously by promoting membrane fluidity

Jin-Hyuck Jeong [1,2,7], Jun-Seok Han[1,7], Youngae Jung[3,7], Seung-Min Lee [1], So-Hyun Park[1,2], Mooncheol Park[1], Min-Gi Shin[1], Nami Kim[3], Mi Sun Kang[3], Seokho Kim[4], Kwang-Pyo Lee [1,2], Ki-Sun Kwon [1,5], Chun-A. Kim[1], Yong Ryoul Yang[1], Geum-Sook Hwang [3,6] ✉ & Eun-Soo Kwon [1] ✉

Dietary restriction (DR) delays aging and the onset of age-associated diseases. However, it is yet to be determined whether and how restriction of specific nutrients promote longevity. Previous genome-wide screens isolated several *Escherichia coli* mutants that extended lifespan of *Caenorhabditis elegans*. Here, using $^1$H-NMR metabolite analyses and inter-species genetics, we demonstrate that *E. coli* mutants depleted of intracellular glucose extend *C. elegans* lifespans, serving as bona fide glucose-restricted (GR) diets. Unlike general DR, GR diets don't reduce the fecundity of animals, while still improving stress resistance and ameliorating neuro-degenerative pathologies of A$\beta_{42}$. Interestingly, AAK-2a, a new AMPK isoform, is necessary and sufficient for GR-induced longevity. AAK-2a functions exclusively in neurons to modulate GR-mediated longevity via neuropeptide signaling. Last, we find that GR/ AAK-2a prolongs longevity through PAQR-2/NHR-49/Δ9 desaturases by promoting membrane fluidity in peripheral tissues. Together, our studies identify the molecular mechanisms underlying prolonged longevity by glucose specific restriction in the context of whole animals.

Reduced nutrient intake, known as dietary restriction (DR) delays aging and the onsets of age-associated diseases in multiple species[1–6]. A lower intake of particular nutrients rather than reduced calorie intake seems critical for its health benefits, with the restriction of specific amino acids playing a significant role[7]. However, aging studies using diets with defined nutrient composition are limited, because even simple animal models, such as worms and flies feed on complex nutrient source, *Escherichia coli*, and yeast, respectively. In addition, the specific and(or) common players are implicated in promoting longevity by different nutrient restrictions such as carbohydrates, amino acids, and fat. Thus they function in highly complicated manners when whole diets are restricted. Therefore, the contribution of a specific nutrient to longevity and the molecular mechanisms involved are poorly understood. To dissect the role of specific nutrients in

[1]Aging Convergence Research Center, Korea Research Institute of Bioscience and Biotechnology (KRIBB), Daejeon 34141, Korea. [2]Biomolecular Science, KRIBB School of Bioscience, Korea University of Science and Technology (UST), Daejeon 34141, Korea. [3]Integrated Metabolomics Research Group, Western Seoul Center, Korea Basic Science Institute, Seoul 03759, Korea. [4]Department of Medicinal Biotechnology, College of Health Sciences, Dong-A University, Busan 49315, Korea. [5]Functional Genomics, KRIBB School of Bioscience, Korea University of Science and Technology (UST), Daejeon 34141, Korea. [6]Department of Chemistry & Nano Science, Ewha Womans University, Seoul 03760, Korea. [7]These authors contributed equally: Jin-Hyuck Jeong, Jun-Seok Han, Youngae Jung. ✉e-mail: gshwang@kbsi.re.kr; eunsoo.kwon@kribb.re.kr

modulating longevity at the organismal level, developing a DR regimen for particular nutrient-restricted diets is essential.

Glucose, the primary energy source of most living organisms, is one of the best-studied carbohydrates that affect aging. Increased glucose intake accelerates aging in various organisms, including yeasts, worms, flies, and mammals. Glucose-rich diets shorten the lifespan of *Caenorhabditis elegans* by affecting the activity of pro-longevity proteins, including AMP-activated protein kinase (AMPK)[8], and the FOXO transcription factor[9] and through toxic metabolite generation[10]. By contrast, 2-deoxy-glucose, a non-metabolizable glucose analog, extends the lifespan by hermetic generation of reactive oxygen species (ROS)[8]. However, to our knowledge, a bona fide glucose-restricted dietary regimen for animal models does not exist at present, that enable the studies on lifespan regulation by glucose-specific restriction.

*C. elegans* is a widely used model organism for aging research[11], because many genes and signaling pathways that regulate longevity are conserved. In *C. elegans*, DR-mediated longevity requires multiple factors, including sirtuins[12], SKN-1[13], PHA-4[14], AMPK[15], DAF-16[15], SEK-1[16], and HSF-1[17]. Interestingly, the extent of their requirement for DR-induced longevity varies, depending on each regimen[18]. Thus, the mechanisms through which they extend lifespan can differ.

Among DR regulators, AMPK is a highly conserved protein kinase, which regulates energy homeostasis by increasing energy production, and reducing energy wastage under low energy conditions. In unicellular eukaryotes, such as yeasts, AMPK directly senses the intracellular energy status and modulates fitness in response to nutritional conditions[19]. However, in multicellular eukaryotes, AMPK is ubiquitously expressed and has evolved to perform specific and overlapping functions depending on the tissues, such as neuronal and peripheral tissues. In mice, hypothalamic AMPK integrates signals from the peripheral tissues through hormones, including leptin and ghrelin, which in turn modulates feeding as well as metabolism[20]. In peripheral tissues, such as the liver and muscles, AMPK autonomously regulates fatty acid oxidation, lipogenesis, glucose uptake, gluconeogenesis, and protein synthesis through the phosphorylation of various targets, including ACC, HMGR, GS, and mTOR[19]. These indicate the divergent evolution of AMPK into sensor and effector to control whole-body energy homeostasis. However, whether AMPK promotes the organismal longevity mainly in peripheral tissues or through endocrine signaling initiated in neurons remains poorly understood. *C. elegans* provides a genetically tractable system for studying the physiological roles of AMPK in the context of the whole animal. Similar to mammals, in *C. elegans*, the two genes *aak-1* and *aak-2* separately encode the catalytic α subunit of AMPK. Thus far, *aak-2* was shown to rescue most AMPK-deficient phenotypes, such as regulation of feeding, fat metabolism, dauer maintenance, and longevity[21]. Of note, *aak-2* is required for only a subset of the DR regimen, in which a diluted bacteria was provided on agar plates in *C. elegans* (sDR)[18]. Previously, neuronal AMPK was suggested to modulate longevity non-cell autonomously by inhibiting neuronal CRTC-1[22]. However, direct evidence is lacking, as neuronal rescue of constitutively active AMPK failed to extend the lifespan[22].

The nervous system is the central integrator of information obtained both internally and externally, and communicates with peripheral tissues through secreted signaling molecules. Neuropeptides are small peptides produced and released by the neurons. They play a variety of roles in regulation of food intake, metabolism, reproduction, social behavior, aging, learning and memory[23–27].

Mounting evidence suggests that lipid metabolism has an important role in aging. Direct supplementation of polyunsaturated fatty acids (PUFAs), such as arachidonic acid, di-homo-γ-linoleic acid, and α-linolenic acid can extend lifespan[28,29] and improves resistance to starvation, oxidative stress, and heat stress[30]. Dietary supplementation with monounsaturated fatty acids (MUFAs), such as oleic, palmitoleic,

or cis-vaccenic acids, is sufficient to increase lifespan[31,32]. Further, MUFAs are abundant in long-lived *C. elegans* mutants and the plasma of long-lived humans[33,34], suggesting that unsaturated fatty acids (UFA) may be beneficial for longevity across species. However, the mechanism involved remains unknown.

The composition of fatty acids in the cell membrane greatly influences membrane properties, such as fluidity[35]. Saturated fatty acids make the membranes rigid, whereas unsaturated fatty acids promote fluidity, which consequently affects cellular physiology impinging upon cell-to-cell communication, intracellular signaling, organellar homeostasis, and membrane dynamics[36]. Interestingly glucose-rich diets promote the saturation of fatty acids[37], hence reducing membrane fluidity, which may contribute to glucose toxicity[38]. Of note, membrane fluidity has been reported to decrease with age in several tissues, including intestinal cells, macrophages, lymphocytes, neurons, the heart, skeletal muscles and blood vessels in mammals[39–41], suggesting that the maintenance of membrane fluidity may promote longevity.

Recently, *C. elegans* and its bacterial diet have become a powerful model system to study the effects of diets on animal life history traits[42,43]. In this study, using ¹H-NMR metabolite analyses coupled with inter-species genetic analyses, we have demonstrated that glucose-depleted *E. coli* mutants extend the lifespan and prolong the healthspan of *C. elegans*. Interestingly, neuronal AAK-2a, a new isoform of AMPK is necessary and sufficient for this GR-mediated longevity. Mechanistically, GR/neuronal AAK-2a prolongs longevity through a series of actions including neuropeptide, PAQR-2/AdipoR, NHR-49/PPARα, and Δ9 desaturases. Further neuronal AAK-2a regulates fat metabolism and promotes membrane fluidity non-cell autonomously. In summary, we provide the molecular mechanisms by which organisms respond to GR condition to promote longevity using conserved factors across species.

## Results

### Glucose-depleted bacteria prolong longevity in *C. elegans*

Previously, we isolated several *E. coli* mutants that prolonged *C. elegans* longevity[44], including Δ*sucA E. coli*. The *sucA* gene encodes α-ketoglutarate (αKG) decarboxylase, the E1 subunit of the αKG dehydrogenase complex that catalyzes the conversion of αKG to succinyl-CoA in the tricarboxylic acid (TCA) cycle. Consistent with previous screen data, the lifespan of N2 control animals was significantly extended by feeding Δ*sucA E. coli* mutants compared with control *E. coli* K12 parental strains (Fig. 1a; Supplementary Table 1a). This lifespan extension was completely suppressed by wild-type *sucA*⁺ complementation (Supplementary Fig. 1a; Supplementary Table 1b), demonstrating that *E. coli* Δ*sucA* mutation causally extended the lifespan of *C. elegans*. Because the TCA cycle functions as a metabolic hub for energy production and a source of biosynthetic precursors, we asked whether the impaired TCA cycle flux in *E. coli* could extend the lifespan of *C. elegans*. We evaluated the lifespans of *C. elegans* that fed six additional mutants of *E. coli* TCA cycle (Fig. 2a), including Δ*acnA*, Δ*acnB*, Δ*icd*, Δ*sucC*, Δ*sdhC*, and Δ*mdh*. Interestingly, none of these mutants affected the *C. elegans* lifespan (Supplementary Table 1c), suggesting that the simple impairment of TCA cycle in *E. coli* did not causally extend the lifespan of *C. elegans*. To systemically study molecular cues that promoted *C. elegans* longevity, we compared the intracellular metabolites extracted from Δ*sucA*, Δ*icd* and control *E. coli* using proton nuclear magnetic resonance (¹H-NMR) (Supplementary Table 2). A Δ*icd E. coli* was included as an additional negative control because TCA cycle impairment likely changed the levels of various metabolites which were not associated with *C. elegans* longevity. Principal component analysis (PCA) revealed that the metabolite profile of Δ*sucA E. coli* was quite distinct from that of wild-type *E. coli* (Fig. 1b). By contrast, the metabolome of Δ*icd E. coli* resembled that of wild-type *E. coli* with minor differences (Fig. 1b), suggesting that

metabolite changes by *ΔsucA* mutation might causally extend the lifespan of the animals. Among the glycolysis-TCA cycle intermediates, *ΔsucA E. coli* accumulated 7-fold more αKG and contained less glucose and TCA cycle intermediates, compared to control *E. coli* (Fig. 1c; Supplementary Table 2). Regarding amino acids, glycine and lysine accumulation increased by 6.67 and 2.93-folds respectively in *ΔsucA E. coli* (Fig. 1c; Supplementary Table 2). As reported[45], glutamate was the most abundant amino acid in wild-type *E. coli* (Supplementary Table 2), supporting the validity of our metabolite analysis.

Previously, αKG was reported to extend the lifespan of *C. elegans* by inhibiting ATPase and TOR[46], suggesting that the accumulated αKG in *ΔsucA E. coli* might cause the lifespan extension of *C. elegans*. First, we performed a series of lifespan assays with αKG supplementation as follows; 2, 4, 6, 8, or 12 mM treatment from L1 larvae or adults; treatment every other day; and treatment on standard *E. coli* B strain OP50 or K12 strains. However, αKG did not extend the *C. elegans* lifespan under the assay conditions we tested (Supplementary Fig. 1b; Supplementary Table 1d). Because bacterial growth was greatly enhanced and the bacteria made a thick lawn by αKG treatment, *E. coli* might have consumed αKG and abolished the beneficial effects of αKG. To exclude this possibility, we used *ΔkgtP E. coli* mutants, which were unable to uptake extracellular αKG and grew similarly to control *E. coli* upon αKG treatment. However, αKG failed to extend the lifespan of worms fed *ΔkgtP E. coli* (Supplementary Fig. 1c; Supplementary Table 1e). Lastly, we asked whether *C. elegans* could uptake αKG from media. When supplemented in various concentrations (0, 2, 4, 10 mM), αKG levels in *C. elegans* increased in a dose-dependent manner (Supplementary Fig. 1d). Together, these data strongly suggested that the accumulated αKG in *ΔsucA E. coli* did not causally promote longevity in *C. elegans*. Another enriched metabolite, glycine was reported to promote longevity in *C. elegans* in a methionine metabolism-dependent manner, as glycine failed to promote longevity in *metr-*

*1(ok521)* and *sams-1(ok3033)* mutants[47]. Notably, *ΔsucA E. coli* extended the lifespan of both *metr-1(ok521)* and *sams-1(ok3033)* mutants (Supplementary Fig. 1e, and 1f; Supplementary Table 1f). Lysine was shown to extend the lifespan in a particular regimen such as liquid medium[48]. To validate the effect of lysine on longevity, we tested whether supplemented lysine could extend the lifespan on a standard solid agar medium in wild-type *C. elegans*. Lysine did not extend lifespan on agar medium at the concentrations previously known to increase lifespan (Supplementary Fig. 1g, h; Supplementary Table 1g), indicating that *ΔsucA E. coli* promotes longevity through a distinct mechanism.

Glucose is the primary energy source for most organisms. Increased glucose intake accelerates aging and the onset of age-associated diseases, such as diabetes and hypertension[49]. However, the effect on aging by glucose restriction is relatively unknown due to the lack of relevant models. To evaluate whether low glucose levels in *ΔsucA E. coli* causally extended the lifespan of *C. elegans*, we first asked whether glucose supplementation could suppress lifespan extension by *ΔsucA E. coli*. As reported[9], 2% glucose (111 mM) significantly shortened the lifespan of control N2 animals (Fig. 1d; Supplementary Table 1h). Further, 2% glucose completely suppressed the lifespan extension by *ΔsucA E. coli* (Fig. 1d). Notably, glucose robustly enhanced the growth of control and *ΔsucA E. coli*, which could indirectly affect *C. elegans* survival[50]. To exclude this possibility, we tested whether a low concentration of glucose could specifically reduce lifespan extension mediated by *ΔsucA E. coli*. Notably, 5 mM glucose suppressed lifespan extension by *ΔsucA E. coli* with a minor effect on *C. elegans* lifespan when treated on control *E. coli* (Fig. 1e; Supplementary Table 1i). Once transported into *E. coli*, glucose is converted into various metabolites, which may affect the lifespan of *C. elegans*. To exclude this possibility, we generated *ΔPTS^Glc* (*ΔptsH-ΔptsI-Δcrr*) mutants in a wild-type or *ΔsucA* background, which were deficient in glucose uptake[9,51]. Importantly, 5 mM glucose reduced lifespan extension mediated by

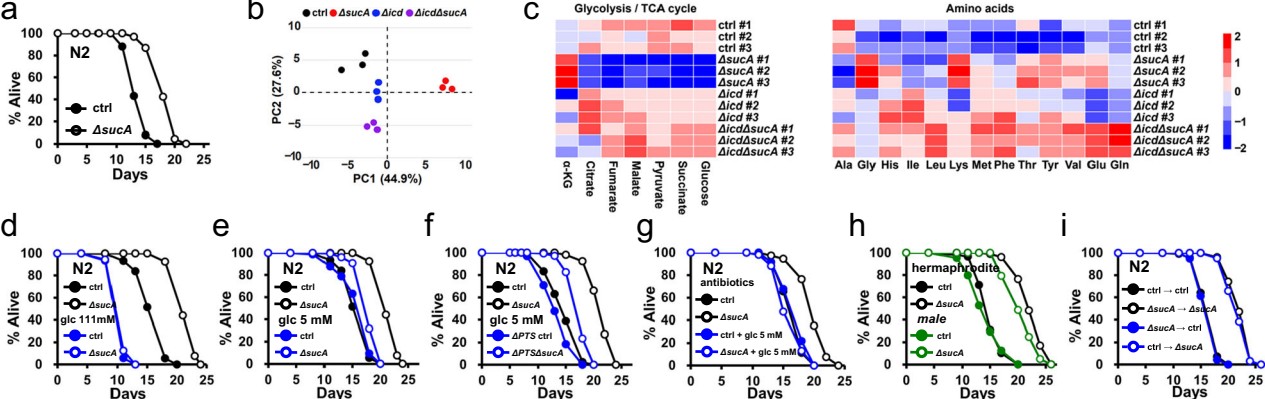

**Fig. 1 | Glucose-depleted *E. coli* mutants promote longevity of *C. elegans*.**
**a** Lifespan showing that *ΔsucA Escherichia coli* mutants extend lifespan of *Caenorhabditis elegans* ($P < 1.0 \times 10^{-10}$, *P* value determined by two-tailed Student's *t* test). *E. coli* K12 parental strain (BW25113) is indicated as ctrl. Results from one of three independent experiments are shown. **b** PCA analysis of metabolites from control, *Δicd*, *ΔicdΔsucA* and *ΔsucA E. coli*. Metabolites from three biological repeats are analyzed using ¹H-NMR. PC1 and PC2 indicate principal component 1 and principal component 2. Each point represents an independent biological sample. **c** Heat map analysis of metabolite quantification data. The row represents the sample and the column represents metabolite. Metabolites significantly decreased are displayed in blue, whereas metabolites significantly increased are displayed in red. The brightness of each color corresponds to the magnitude of the difference when compared with average values. The glycolysis-TCA metabolites and amino acids are presented. For total metabolites data, see also Supplementary Table 2. **d** Lifespan showing that 2% glucose (111 mM) treatment completely suppress the lifespan extension by GR diets ($P = 0.3318$, *P* value determined by two-tailed

Student's *t* test). Results from one of three independent experiments are shown. **e**, **f** Lifespan showing that glucose supplementation (5 mM) shortens lifespan extension by *ΔsucA E. coli* mutants (**e**) and by *ΔPTS^Glc ΔsucA E. coli* mutants (**f**) ($P < 1.0 \times 10^{-10}$, *P* value determined by two-tailed Student's *t* test). Results from representative experiments are shown with additional repeats. **g** Lifespan showing that 5 mM glucose treatment significantly reduce the lifespan extension by *ΔsucA E. coli* ($P = 0.0197$, *P* value determined by two-tailed Student's *t* test). All *E. coli* strains are metabolically inactivated by antibiotics, ampicillin and kanamycin. Results from representative experiments are shown with additional repeats. **h** Lifespan showing that *ΔsucA E. coli* extend lifespan in both hermaphrodites and male *C. elegans* ($P < 1.0 \times 10^{-10}$, $P < 1.0 \times 10^{-10}$, respectively, *P* value determined by two-tailed Student's *t* test). Results from representative experiments are shown with additional repeats. **i** Lifespan showing that adult stage diets determine the lifespan of *C. elegans*. Adult-only GR diets sufficiently extend lifespan ($P < 1.0 \times 10^{-10}$, *P* value determined by two-tailed Student's *t* test). Results from one of three independent experiments are shown. Source data are provided as a Source Data file.

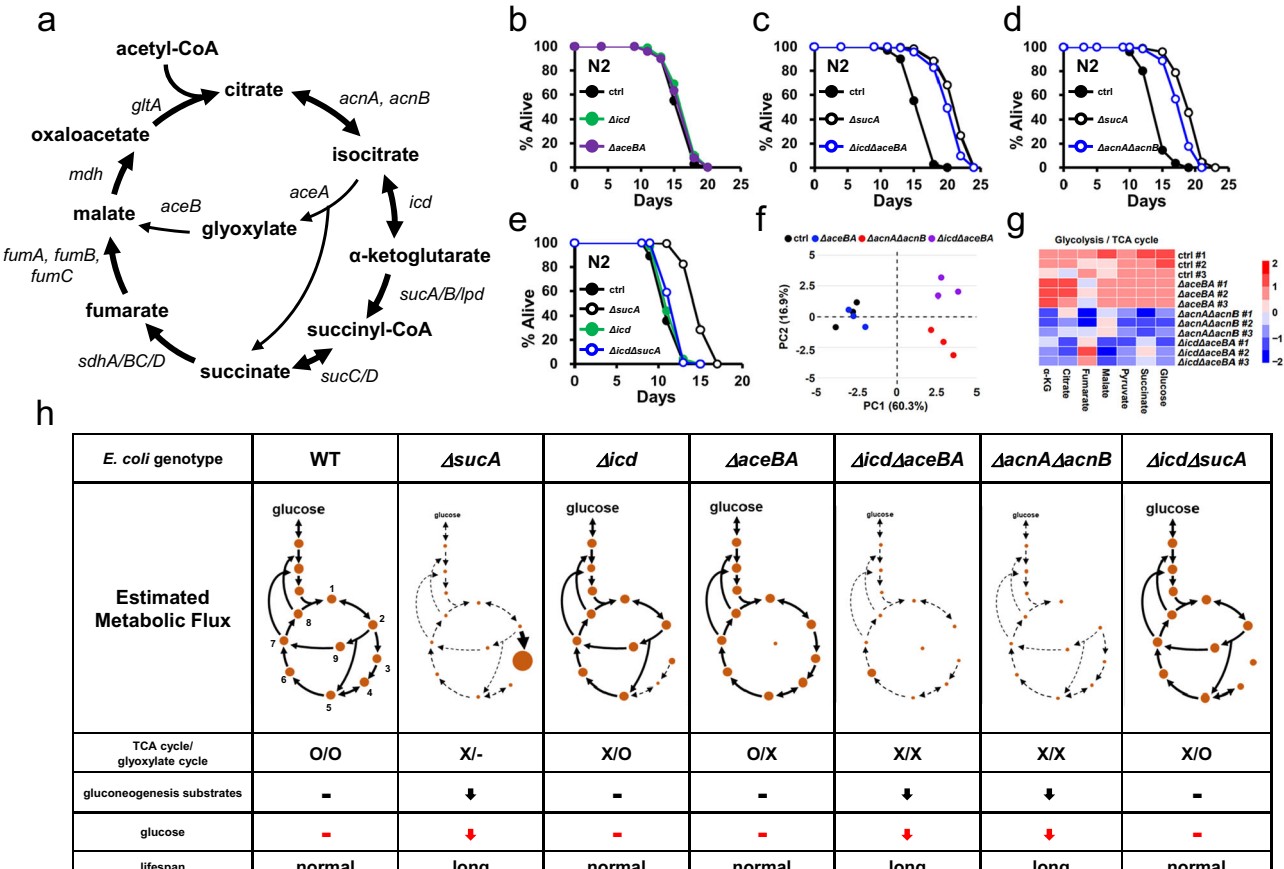

**Fig. 2 | *E. coli* mutants simultaneously impaired in TCA and glyoxylate cycle prolong the longevity of *C. elegans*. a** The TCA and glyoxylate cycles in *E. coli*. Metabolites are indicated in bold and genes encoding metabolic enzymes are indicated in italic. **b**–**e** Lifespan showing the effects of *E. coli* TCA and (or) glyoxylate cycle mutations on *C. elegans* longevity. Neither *Δicd* nor *ΔaceBA E. coli* mutants affect lifespan (**b**) while *ΔicdΔaceBA E. coli* triple mutants extend lifespan (**c**). *ΔacnAΔacnB E. coli* double mutants extended lifespan (**d**). *Δicd* mutation abolished lifespan extension by *ΔsucA E. coli* mutants (**e**). Results from representative experiments are shown with additional repeats. **f** PCA analysis of metabolites from control, *ΔaceBA*, *ΔicdΔaceBA*, and *ΔacnAΔacnB E. coli*. Metabolites from three biological repeats are analyzed using ¹H-NMR. PC1 and PC2 indicate principal component 1 and principal component 2. Each point represents an independent biological sample. **g** Heat map analysis of metabolite quantification

data. The row represents the sample and the column represents metabolite. Metabolites significantly decreased are displayed in blue, whereas metabolites significantly increased are displayed in red. The brightness of each color corresponds to the magnitude of the difference when compared with average values. For total metabolites data, see also Supplementary Table 3. **h** The estimated metabolic fluxes of gluconeogenesis, TCA, and glyoxylate cycles in *E. coli* mutants. The thickness of lines corresponds to the magnitude of metabolic fluxes. The dotted line represents weak metabolic flux. Circle size represents the level of metabolic intermediates. Numbers indicate citrate (1), isocitrate (2), αKG (3), succinyl-CoA (4), succinate (5), fumarate (6), malate (7), oxaloacetate (8), and glyoxylate (9) in WT panel. The effects on *C. elegans* lifespan are also indicated. Source data are provided as a Source Data file.

*ΔPTS^Glc^ΔsucA E. coli* to a similar extent as *ΔsucA E. coli* (Fig. 1f; Supplementary Table 1j). Further, we found that glucose supplementation suppressed lifespan extension by *ΔsucA E. coli* that was metabolically inactivated by antibiotics (Fig. 1g; Supplementary Table 1k).

Previously glucose-induced toxicity in *paqr-2 C. elegans* that is compromised in lipid metabolism[38]. Interestingly this glucose toxicity requires the conversion of glucose into saturated fatty acids (SFA) by *E. coli*, as *ΔPTS^Glc^ E. coli* mutants prevent the toxic glucose effects on *paqr-2 C. elegans*. Although we show that glucose still reduce the lifespan of N2 animals on *ΔPTS^Glc^ΔsucA E. coli* as well as metabolically inactivated *ΔsucA E. coli*, glucose depleted *ΔsucA E. coli* may contain less SFA and more unsaturated FA (UFA). This change of FA composition in *E. coli* could extend lifespan of *C. elegans*. To test this idea, we measured the FA composition of *ΔsucA E. coli* using UPLC/QTOF mass spectrometry. Of note, the SFA/UFA ratio in phosphatidylethanolamine is not lower in *ΔsucA E. coli*. It is slightly higher, compared to the control (Supplementary Fig. 1i; Supplementary Table 1l), suggesting that the FA composition in *ΔsucA E. coli* did not causally promote longevity of *C. elegans*. Taken altogether,

glucose abrogated lifespan extension by *ΔsucA E. coli* by directly acting on *C. elegans*.

Sex-differences in aging manipulations are recognized across species, including worms[52], flies[53], and mouse[54,55]. Interestingly, *C. elegans* males are almost unresponsive to several DR regimens[56]. To investigate whether our glucose restricted (GR) regimens could be applicable to two sexes, we performed lifespan assays using male *C. elegans* on GR diets vs. AL diets. Similar to *C. elegans* hermaphrodites, *ΔsucA E. coli* extended the lifespan of male *C. elegans* (Fig. 1h; Supplementary Table 1m). Finally, to investigate when *ΔsucA E. coli* act to prolong longevity, we conducted diet-switch experiments. After larval development, the worms were transferred from wild-type control *E. coli* to *ΔsucA E. coli* or vice versa. Interestingly, adult-only *ΔsucA E. coli* diets sufficiently extended the lifespan (Fig. 1i; Supplementary Table 1n). By contrast, developmental stage-only *ΔsucA E. coli* diets exhibited no lifespan extension (Fig. 1i; Supplementary Table 1n).

Overall, these data suggest that the glucose-depleted *ΔsucA E. coli* mutants prolong the longevity of *C. elegans*, serving as GR diets and adult-only GR diets sufficiently prolong the lifespan of *C. elegans*.

### *E. coli* mutants simultaneously impaired in TCA and glyoxylate cycle prolong the longevity of *C. elegans*

In the laboratory, *C. elegans* feed *E. coli* that grows on peptone-based nematode growth media plates, which lacks carbohydrates, such as glucose. Therefore, *E. coli* must synthesize the six-carbon sugar to make various cellular macromolecules from non-sugar compounds, such as amino acids. Amino acids are converted to glycolysis/TCA cycle intermediates, which could serve as substrates for gluconeogenesis. Notably, metabolites from central carbon metabolism, including glucose, pyruvate, succinate, fumarate, and malate were significantly less in *ΔsucA E. coli*, compared to the control (Fig. 1c; Supplementary Table 2), thereby implying lowered gluconeogenesis. By contrast, in TCA cycle-deficient *Δicd E. coli*, the levels of these metabolites were comparable to those in control (Fig. 1c; Supplementary Table 2). These levels of metabolites could be due to glyoxylate bypass, which enables alternative metabolic cyclic flux by converting isocitrate to succinate and malate (Fig. 2a). Therefore, we hypothesized that impairment of the glyoxylate cycle in *Δicd E. coli* should extend the lifespan of *C. elegans*, as it mimicked a reduced gluconeogenesis in *ΔsucA E. coli*. The *E. coli* genome contains the *aceB-aceA* operon, which encodes the glyoxylate cycle enzymes, malate synthase and isocitrate lyase, respectively (Fig. 2a). We generated *ΔaceBA* mutants in either wild-type or *Δicd* mutant backgrounds. While the lifespans of *C. elegans* fed on TCA cycle-deficient *Δicd* or glyoxylate cycle-deficient *ΔaceBA E. coli* were comparable to that on control *E. coli* (Fig. 2b; Supplementary Table 1o), *ΔicdΔaceBA* triple mutants remarkably extended the lifespan of *C. elegans*, similarly to *ΔsucA E. coli* (Fig. 2c; Supplementary Table 1o). This suggests that *E. coli* mutants, which are deficient in both TCA and glyoxylate cycle, prolong *C. elegans* lifespan by serving as low-glucose diets.

Aconitase converts citrate to isocitrate, being a shared enzyme in the TCA and glyoxylate cycle. Therefore, we reasoned that aconitase-deficient *E. coli* should extend *C. elegans* lifespan. *E. coli* contains two aconitase encoding genes, *acnA* and *acnB* (Fig. 2a). Previously, we showed that single mutants of either gene did not affect the lifespan of the animals (Supplementary Table 1c). We generated a *ΔacnAΔacnB* double mutant and assessed its effects on *C. elegans* lifespan. Surprisingly, it significantly extended *C. elegans* lifespan, similarly to *ΔsucA E. coli* (Fig. 2d; Supplementary Table 1p). To validate the metabolic features of newly generated *E. coli* mutants, we analyzed the intracellular metabolites of *ΔacnAΔacnB*, *ΔicdΔaceBA*, *ΔaceBA E. coli* using $^1$H-NMR. Consistent with our idea, two *E. coli* mutants that extend the lifespan of *C. elegans* exhibit similar metabolite profiles to *ΔsucA E. coli*, with low level of glucose and TCA intermediates (Fig. 2f; Supplementary Table 3). In contrast, the metabolite profile of *ΔaceBA* mutants only deficient in the glyoxylate cycle resembles that of control. Thus, *C. elegans* lifespan is extended by *E. coli* mutants with a simultaneous impairment of TCA and glyoxylate cycle.

Pyruvate, a central metabolite in glycolysis and TCA cycle, is reduced in all *E. coli* mutants that extend the lifespan of *C. elegans* (Fig. 1c; Fig. 2g; Supplementary Table 2). Next we asked if pyruvate could suppress the lifespan extension by *ΔsucA E. coli* similarly with glucose. To our surprise, unlike glucose, pyruvate did not restore the growth retardation of *ΔsucA E. coli* to normal (Supplementary Fig. 1j). This implies that supplemented pyruvate might not be actively converted to glucose in *ΔsucA E. coli*, potentially due to active consumption by glycolysis and TCA cycle. Consequently, pyruvate did not abolish the lifespan extension by *ΔsucA E. coli* (Supplementary Fig. 1k; Supplementary Table 1q). Glycerol enters into gluconeogenesis via dihydroxyacetone phosphate and the upper part of glycolysis[57]. Recently it has been reported that glycerol is a better substrate for gluconeogenesis than pyruvate in fasting mice[58]. Therefore, we asked whether glycerol could suppress lifespan extension by *ΔsucA E. coli*. Unlike pyruvate, glycerol enhanced the growth of *ΔsucA E. coli* (Supplementary Fig. 1j) and the lifespan extension by *ΔsucA* was

significantly reduced by glycerol (Supplementary Fig. 1l; Supplementary Table 1r). These data demonstrate that *ΔsucA*-induced longevity can be suppressed by metabolite that is easily converted to glucose.

Isocitrate is a branch point metabolite, which is converted to αKG through the TCA cycle or to succinate and malate through the glyoxylate cycle (Fig. 2a). Given the low level of succinate and malate and the high level of αKG in *ΔsucA* mutants (Fig. 1c; Supplementary Table 2), TCA cycle flux must override that of the glyoxylate cycle in *ΔsucA E. coli*. If that is the case, the inhibition of TCA cycle flux in *ΔsucA E. coli* by introducing an *Δicd* mutation can facilitate the glyoxylate cycle (Fig. 2h) and, in turn, abolish the lifespan extension by *ΔsucA* mutation. As hypothesized, the metabolite profile of the *ΔicdΔsucA* double mutants became similar to those of the control and *Δicd* mutants (Fig. 1b; Supplementary Table 2), recovering the levels of TCA intermediates and glucose (Fig. 1c). Of note, *Δicd* mutation completely abolished the lifespan extension by *ΔsucA E. coli* (Fig. 2e; Supplementary Table 1s). The levels of metabolites and metabolic fluxes in six respective *E. coli* mutants are presented in Fig. 2h. Hereafter, *ΔsucA E. coli* and wild-type *E. coli* will be referred to as GR diets and *ad libitum* (AL) diets, respectively, in this study. Taken together, these results demonstrate that *E. coli* mutants impaired in both TCA and glyoxylate cycle serve as low glucose diets and promote longevity in *C. elegans*.

### GR ameliorates age-associated pathologies with no trade-off of fecundity

DR not only extends the lifespan but also delays various age-associated diseases, including neurodegenerative diseases[59]. We tested whether GR diets could ameliorate age-associated pathologies using Alzheimer's disease (AD) model animals. The 42 amino acids of beta amyloid (Aβ$_{42}$) play a pivotal role in the pathogenesis of AD[60]. *C. elegans* expressing human Aβ$_{42}$ exhibited an early decline in physical activity[61]. As expected, we found that GR diets substantially delayed age-associated paralysis (Fig. 3a; Supplementary Table 1t). Further, *C. elegans* grown on GR diets maintained health with aging, as determined by enhanced stress resistance (Fig. 3b; Supplementary Table 1u). By contrast to its beneficial effects, DR often reduced fecundity likely due to a soma/germline trade-off[7]. However, interestingly GR diets were not generally dietary restricted, as brood size of *C. elegans* feeding GR diets was comparable with that on AL diets during two consecutive generations (Fig. 3c, d; Supplementary Table 1v). Taken together, our findings reveal that GR diets improve many indices of healthspan without loss of fecundity.

### New AMPK isoform, AAK-2a mediates GR lifespan extension

The non-metabolizable glucose analog, 2-deoxy glucose (2-DG) inhibits glucose metabolism, indicating its potential as a DR mimetic. Given that antioxidant, *N*-acetyl cysteine (NAC), suppressed the lifespan extension by 2-DG, hormesis, a beneficial effect of mild stress, is critical for 2-DG-induced longevity in *C. elegans*[8]. However, we found that NAC did not suppress lifespan extension by GR diets (Supplementary Fig. 2a; Supplementary Table 4a), suggesting that a distinct mechanism was involved. Therefore, to gain insight into the pro-longevity mechanism of GR diets, we asked if GR diets extended lifespan through known DR effectors, including FOXA, FOXO, TOR, HIF, AMPK, Sirtuin, HSF, SEK-1, and NRF2 as well as other longevity pathway components. *C. elegans* mutants tested were as follows, *pha-4(RNAi)*, *daf-16(mgDf50)*, *raga-1(ok386)*, *rict-1(ft7)*, *hif-1(ia4)*, *aak-2(ok524)*, *sir-2.1(ok434)*, *hsf-1(sy441)*, *skn-1(zu135)*, *sek-1(km4)*, *pdr-1(gk448)*, *pink-1(ok3538)*, and *isp-1(qm150)*. We found that most DR effectors were dispensable for GR-induced longevity (Supplementary Fig. 2b–i; Supplementary Table 4b). In addition, stress response and mitochondrial longevity pathway were not implicated (Supplementary Fig. 2j–m; Supplementary Table 4b). Interestingly, GR-mediated longevity was completely abolished only in *aak-2(ok524)* mutants (Fig. 4a; Supplementary Table 4c). An *aak-2* encodes the catalytic α-subunit of AMPK,

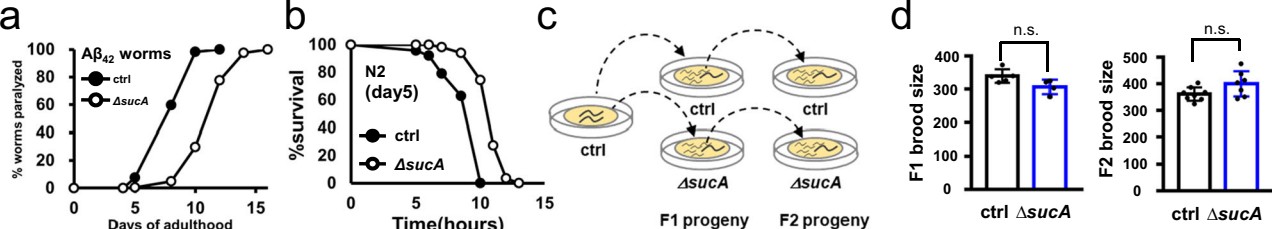

**Fig. 3 | GR delays Age-associated pathologies with no germline trade-off.**
**a** Paralysis assay showing that GR diets delayed age-associated paralysis in Aβ$_{42}$ animals ($P < 1.0 \times 10^{-10}$, $P$ value determined by two-tailed Student's $t$ test). Results from one of three independent experiments are shown. **b** Survival assay showing that GR diets enhance heat stress resistance of worms at day 5 ($P < 1.0 \times 10^{-10}$, $P$ value determined by two-tailed Student's $t$ test). Results from representative experiments are shown with additional repeats. **c, d** Brood size analysis during

consecutive generations on the indicated diets (**c**). GR diets do not reduce F1 brood size (AL diets: $n = 5$, GR diets: $n = 4$, $P = 0.0532$, $P$ value determined by two-tailed Student's $t$ test) and F2 brood size (AL diets: $n = 8$, GR diets: $n = 7$, $P = 0.0653$, $P$ value determined by two-tailed Student's $t$ test) (**d**). Results from representative experiment are shown one of two independent repeats. Error bars indicate mean ± s.d. Source data are provided as a Source Data file.

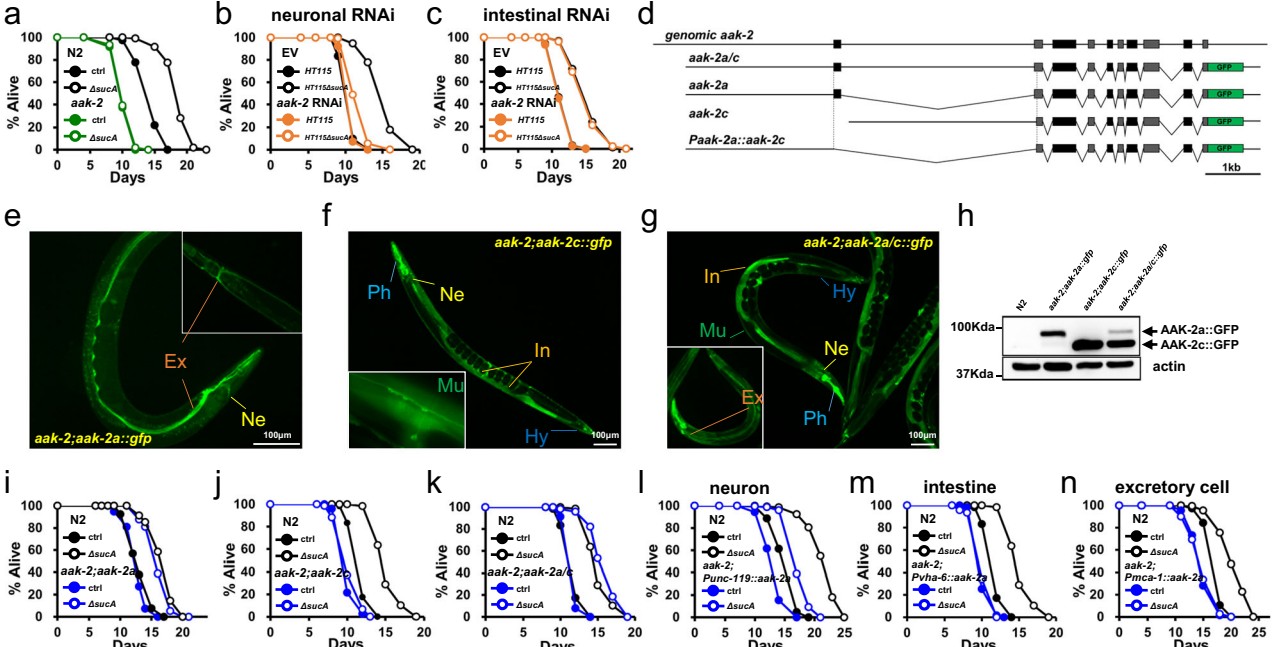

**Fig. 4 | New AMPK isoform, AAK-2a modulates GR-mediated longevity in neurons. a** Lifespan showing that *aak-2(ok524)* abolishes GR-induced longevity ($P = 1$, $P$ value determined by two-tailed Student's $t$ test). Results from one of three biological independent experiments are shown. **b, c** Lifespan showing that neuronal AAK-2 is required for GR-induced longevity. Tissue-specific RNAi animals are used for neuronal *aak-2* RNAi knockdown (**b**) and intestinal *aak-2* RNAi knockdown (**c**). Results from one of three biological independent experiments are shown. **d** Structure of *aak-2*⁺ gene and *aak-2 isoform::gfp* constructs. Coding regions are indicated as closed boxes; noncoding regions as line; introns spliced-out in DNA constructs as ∧. Locus of start codon for AAK-2 isoforms are indicated as dotted lines. **e–g** GFP image showing the expression of AAK-2 isoforms. GFP images are taken using *aak-2(ok524);aak-2 isoform::gfp* animals. AAK-2a is expressed in the head neurons (Ne) and excretory cells (Ex) (**e**). AAK-2c is expressed in multiple tissues, including the intestine (In), pharynx (Ph), muscles (Mu), hypodermis (Hy),

and neuron (Ne) (**f**). AAK-2a/c transgenic animals show the combined expression patterns (**g**). Representative images obtained from at least four biological independent repeats. **h** Immunoblotting analysis showing the existence and relative levels of AAK-2 isoforms in the indicated animals. Worm lysates were detected with anti-GFP (top panel), and anti-α-tubulin (bottom panel). Representative images obtained from four biological independent repeats. **i–k** Lifespans showing contribution to GR-induced longevity by AAK-2 isoform(s). The lifespans of *aak-2(ok524)* expressing AAK-2 isoform(s) under endogenous promoter are shown; AAK-2a (**i**), AAK-2c (**j**) and AAK-2a/c (**k**). Results from one of three biological independent experiments are shown. **l–n** The lifespans of *aak-2(ok524)* transgenic animals expressing AAK-2a isoform in indicated tissues; AAK-2a in the neurons (**l**), the intestine (**m**) and the excretory cells (**n**). Results from one of three biological independent experiments are shown. Source data are provided as a Source Data file.

which rescues most AMPK-deficient phenotypes in *C. elegans*[62], indicating that AMPK is specifically required for GR-induced lifespan extension. AAK-2 is expressed in various tissues including neurons, muscles, excretory cells, and intestinal cells[62]. Although AMPK performs specific and overlapping functions in organismal energy homeostasis depending on the expressed tissues[63], it is relatively unknown whether AMPK activity, in the neurons or peripheral tissues (or both), plays a critical role in longevity in the context of whole

animals. Interestingly, although GR-mediated longevity was completely suppressed in *aak-2(ok524)* mutants, it was not affected by *aak-2* RNAi (Supplementary Fig. 2n; Supplementary Table 4d). The efficiencies of *aak-2* RNAi were confirmed by the reduced transcript of *aak-2* (Supplementary Fig. 2p) and the GFP intensity in AAK-2::GFP animals (Supplementary Fig. 2q, r). Because *C. elegans* neurons are refractory to RNAi[64,65], this data suggests that neuronal AMPK activity is crucial for GR-mediated longevity. Consistently, *aak-2* RNAi

impaired lifespan extension by GR diets in *rrf-3(pk1426)* mutants that were hyper-sensitive to RNAi[66] (Supplementary Fig. 2o; Supplementary Table 4e). To further validate this data, we utilized neuron-specific RNAi transgenic strain, TU3401 which express the dsRNA transporter in neurons[67] and intestine-specific RNAi strain, VP303 which enable RNAi only in the intestine[68], for *aak-2* knockdown. As shown in Fig. 4b, the knockdown of *aak-2* in neuronal cells significantly reduced GR-mediated lifespan extension (Supplementary Table 4f). By contrast, *aak-2* RNAi in intestinal cells did not affect GR-mediated lifespan extension (Fig. 4c; Supplementary Table 4f), similarly to that in N2. These data support that AMPK play an essential role in GR-induced longevity exclusively in neurons. According to expressed sequence tag database, *C. elegans* expresses at least two isoforms of *aak-2* using distinct upstream promoters and start sites, generating AAK-2a and AAK-2c isoforms (Fig. 4d). Thus far, an AAK-2 isoform-specific study has not been conducted, such as the tissue-specific expressions and potential distinct functions of isoforms. To evaluate the role of AMPK isoforms in GR-mediated longevity, next we generated transgenic animals specifically expressing AAK-2a, AAK-2c, or AAK-2a/c tagged with GFP using each endogenous promoter in an *aak-2(ok524)* background (Fig. 4d). AAK-2c was expressed in various tissues, including the pharynx, intestine, muscles, hypodermis, and neurons (Fig. 4f), consistent with previous AAK-2 expression[62]. This implies that previous AMPK studies utilized the AAK-2c isoform. Interestingly AAK-2a was expressed in only a few head neurons and excretory cells (Fig. 4e), with the combined expression patterns being observed in *aak-2(ok524);aak-2a/c::gfp* animals (Fig. 4g). To verify the existence and the levels of each isoform, we detected AAK-2 protein in transgenic animals expressing either of or both isoforms, using western blotting. Consistent with GFP intensities, the protein level of AAK-2c is much higher than that of AAK-2a (Fig. 4h). As AAK-2a is larger with additional 62 amino acids at its N-terminus, two discrete bands of AAK-2a and AAK-2c were detected in AAK-2a/c transgenic animals (Fig. 4h), demonstrating that two different AAK-2 isoforms are endogenously expressed in *C. elegans*.

To determine which AMPK isoform(s) was required for GR-induced longevity, we explored the lifespan of wild-type N2, *aak-2(ok524)*, and *aak-2(ok524)* mutants expressing either isoform or both, which were fed either AL or GR diets. Surprisingly, AAK-2a fully rescued GR-induced longevity (Fig. 4i; Supplementary Table 4g), whereas the widely expressed AAK-2c was dispensable for GR-induced longevity (Fig. 4j; Supplementary Table 4h). The lifespans of *aak-2(ok524)* transgenic animals expressing both AAK-2a/c isoforms were comparable to that of *aak-2(ok524);aak-2a* transgenic animals (Figs. 4i, and 4k; Supplementary Table 4i). Importantly *aak-2(ok524);aak-2a* transgenic animals lived longer compared to *aak-2(ok524); aak-2c* transgenic animals on control diets (Supplementary Fig. 3a; Supplementary Table 4j), demonstrating that the AAK-2a isoform plays a critical role not only in GR induced longevity but also in the normal lifespan. To further determine whether *aak-2a* is sufficient for wild-type *aak-2*⁺ for lifespan regulation, we generated *aak-2*⁺;*aak-2a::gfp* animals and compared their lifespans with *aak-2(ok524);aak-2a::gfp* animals on AL as well as GR diets. We found that their lifespans were comparable on either AL or GR diets (Supplementary Fig. 3b; Supplementary Table 4k), indicating that the AAK-2a isoform was sufficient for endogenous AMPK-mediated lifespan regulation. Notably, *aak-2(ok524); aak-2c* transgenic animals lived longer than *aak-2* mutant animals (Supplementary Fig. 3a; Supplementary Table 4j). This indicates that the AAK-2c isoform also has a role in longevity on AL diets, although its role is minor. Together, our data reveal that the new AMPK isoform AAK-2a plays the major role in longevity. Since AAK-2a exhibited a distinct tissue expression pattern and contained an additional amino acids at N-terminus, compared with AAK-2c (Fig. 4d), next we asked what determined GR-mediated lifespan extension by AMPK. We generated transgenic animals expressing AAK-2c::GFP using *aak-2a*

promoter in *aak-2(ok524)* background (Supplementary Fig. 3c). Notably, GR diets extended the lifespan of these animals (Supplementary Fig. 3d; Supplementary Table 4l), demonstrating that the expression pattern of AAK-2 determined the GR-mediated lifespan extension. AAK-2a is expressed in the neurons and excretory cells (Fig. 4e). Although our data strongly support the neuronal function of AAK-2a in GR-induced longevity, it was shown that AAK-2 in excretory cells plays roles in longevity[69] and germ cell proliferation[70]. In their studies, the *sulp-5* promoter was used to drive *aak-2* expression in the excretory cells. Therefore, to further explore in which cells AMPK functioned for GR-mediated lifespan extension, we generated a series of *aak-2(ok524); tissue-specific aak-2a::gfp* animals using the pan-neuronal *Punc-119*, *Prab-3*; excretory *Pmca-1*, *Psulp-5*; and intestinal *Pvha-6* promoters (Supplementary Fig. 3e–i) and measured their lifespans on AL vs. GR diets. Consistent with our tissue-specific RNAi lifespan data, neuronal AAK-2a restored lifespan extension by GR diets, whereas AAK-2a in intestine did not (Fig. 4l, m; Supplementary Fig. 3j; Supplementary Table 4m).

The role of excretory cells was more complex, as *Psulp-5::aak-2a* transgene rescued GR-induced longevity while *Pmca-1::aak-2a* did not (Fig. 4n; Supplementary Fig. 3k; Supplementary Table 4m, n). Of note, in contrast to the exclusive expression in excretory cells by *Pmca-1* (Supplementary Fig. 3g), *Psulp-5* promoter driven AAK-2 is not only in excretory cells but also in neurons (Supplementary Fig. 3h)[71]. To knock down AAK-2a specifically in the excretory cells, we took advantage of RNAi feeding, which is refractory in neurons. As expected, *gfp* RNAi abolished the expression of AAK-2 only in excretory cells in *aak-2;Psulp-5::aak-2a* animals, while having no effect on neuronal *aak-2a* expression (Supplementary Fig. 3l). Importantly, this RNAi did not suppress GR mediated longevity (Supplementary Fig. 3m, n; Supplementary Table 4o), demonstrating that AAK-2 in neurons is sufficient to extend lifespan by GR diets.

Taken together, we have demonstrated that a new AMPK isoform, AAK-2a modulates GR-mediated longevity in a neuron-dependent manner.

## AAK-2a modulate longevity non-cell autonomously via neuropeptide signaling

In mammals, hypothalamic AMPK modulates feeding, thermogenesis as well as fat metabolism in peripheral tissues in an endocrine manner[72]. In *C. elegans*, constitutively active AMPK (CA-AMPK) was reported to extend the lifespan[22]. CA-AMPK inhibited CRTC-1 in the neurons, thereby inhibiting pro-aging catecholamine signaling to prolong longevity. However, the neuronal function of AMPK in longevity remains elusive, as neuron-specific expression of CA-AMPK was unable to extend lifespan[22]. Since AAK-2a is only required in the neurons to mediate GR-induced longevity, we first asked which endocrine activity was required for GR-mediated lifespan extension. We utilized the *unc-13* mutants, which were defective in neurotransmitter release through small clear vesicles[73], and *unc-31* mutants, defective in release of dense core vesicles that mainly contained neuropeptides[73]. Interestingly, GR diets extended the lifespans of multiple *unc-13* mutant alleles, including *e51*, *s69*, *e450*, and *e1091* (Fig. 5a, and Supplementary Fig. 4a–c; Supplementary Table 5a). By contrast, *unc-31 (e928)* completely abolished the GR-mediated lifespan extension (Fig. 5b; Supplementary Table 5b). Next, the lifespan extension by GR diets was significantly impaired in *egl-21 (n476)*, carboxypeptidase E mutants (Fig. 5c; Supplementary Table 5c), which was deficient in neuropeptide processing[74]. Thus, GR/neuronal AAK-2a modulates longevity non-cell autonomously through neuropeptides. To further exclude the possibility that neurotransmitters are involved in GR-mediated longevity, we measured the lifespan of various neurotransmitter signaling mutants, including mutants for octopamine; *tdc-1(ok914)*, serotonin; *tph-1(n4622)*, and dopamine; *cat-2(e1112)*. Notably, in contrast to the dependence on TDC-1 in CA-AMPK/CRTC-1-mediated longevity[22], GR

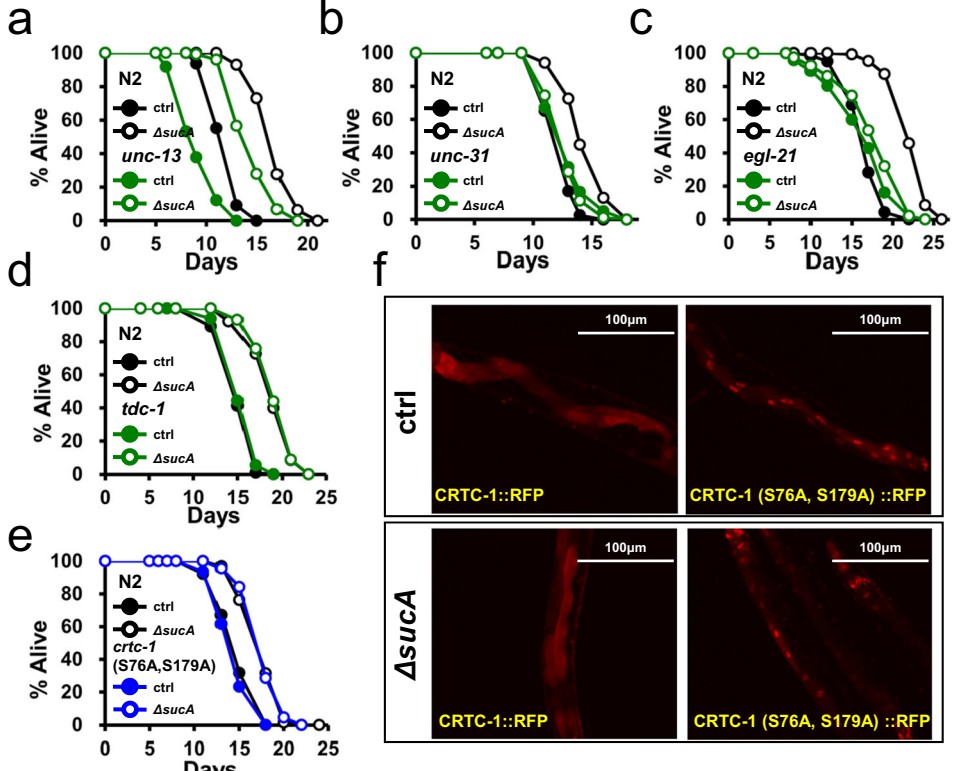

**Fig. 5 | GR promotes longevity non-cell autonomously via pro-longevity neuropeptide. a** Lifespan showing that GR diets extend lifespan of *unc-13 (e51)* ($P < 1.0 \times 10^{-10}$, *P* value determined by two-tailed Student's *t* test). Results from one of three biological independent experiments are shown. **b** Lifespan showing that GR diets fail to extend lifespan of *unc-31(e928)* ($P = 0.3647$, *P* value determined by two-tailed Student's *t* test). Results from one of three biological independent experiments are shown. **c** Lifespan showing that GR-induced longevity is significantly shortened in *egl-21(n476)* animals ($P = 0.006$, *P* value determined by two-tailed Student's *t* test). Results from one of three biological independent experiments are shown. **d** Lifespan showing that GR diets extend the lifespan of *tdc-1(ok914)* mutant animals ($P < 1.0 \times 10^{-10}$, *P* value determined by two-tailed Student's *t* test). Results from one of three biological independent experiments are shown. **e** Lifespan showing that GR diets extend the lifespan of CRTC-1(S76A, S179A) animals ($P < 1.0 \times 10^{-10}$, *P* value determined by two-tailed Student's *t* test). Results from one of three biological independent experiments are shown. **f** GR diets do not affect the nuclear/cytoplasmic localization of CRTC-1::RFP and CRTC-1(S76A, S179A)::RFP. Representative images obtained from three biological independent repeats. Source data are provided as a Source Data file.

diets extended the lifespan of all tested mutants, including *tdc-1(ok914)* mutants (Fig. 5d, Supplementary Fig. 4d, and 4e; Supplementary Table 5d). Lastly, we asked whether GR diets extended the lifespan of CRTC-1 (S76A, S179A) mutants, in which CRTC-1 is constitutively nuclear and refractory to the lifespan extension by CA-AAK-2[73]. Notably, GR diets extended the lifespan of CRTC-1(S76A, S179A) animals (Fig. 5e; Supplementary Table 5e), demonstrating that GR-mediated pro-longevity signaling is dominant over the CRTC-1-mediated pro-aging signaling. GR diets did not affect the nuclear/cytoplasmic localization of CRTC-1, further supporting the independence of CRTC-1 in GR-mediated longevity (Fig. 5f). Together, our data demonstrate that GR/neuronal AMPK prolongs longevity non-cell autonomously through neuropeptide signaling.

**GR-mediated lifespan extension requires NHR-49**

Organelle homeostasis was often associated with AMPK-mediated longevity, as CA-AMPK did not extend the lifespan of mitochondrial dynamics mutants and peroxisome-depleted animal, such as *fzo-1 (tm1133)*, *drp-1 (tm1108)*, and *prx-5* RNAi animals respectively[75]. We asked whether regulators of organelle dynamics were required for GR-mediated longevity. Surprisingly, GR diets extended the lifespans of *drp-1(tm1108)*, *fzo-1(tm1133)*, and *prx-5* RNAi animals (Supplementary Fig. 5a–c; Supplementary Table 6a), demonstrating that GR prolongs longevity through different mechanisms. In response to low nutrient intake, AMPK remodels fat metabolism through peroxisome proliferator-activated receptor alpha (PPARα) across species[76,77]. In

*C. elegans*, NHR-49, a functional PPARα orthologue, was shown to be required for CA-AMPK-mediated longevity[22]. We, therefore, asked whether NHR-49 was required for GR-mediated longevity. As shown in Fig. 6a, *nhr-49(nr2041)* completely abolished GR-mediated longevity (Supplementary Table 6b).

Both AMPK and NHR-49 control energy homeostasis, lipid metabolism and longevity[22,78]. However, their relative interactions remain elusive. Since either *aak-2* or *nhr-49* mutation shortened normal lifespans on control diets (Figs. 4a and 6a), we asked if the effects on lifespan by either mutation are additive or not. An *aak-2* mutation in the *nhr-49(nr2041)* mutants did not further reduce the lifespan of *nhr-49(nr2041)* animals (Fig. 6b; Supplementary Table 6d), suggesting that AAK-2 and NHR-49 might function in the same pathway. For a genetic epistasis analysis, we introduced the *nhr-49* gain-of-function (*gof*) mutations, *et7* and *et13*, in *aak-2(ok524)* animals. Despite the unresponsiveness of *aak-2* mutants to GR diets, *nhr-49$^{gof}$;aak-2* animals lived longer on GR diets, compared to AL diets (Fig. 6c, d; Supplementary Table 6e). Furthermore, the *nhr-49$^{gof}$(et7 and et13)* transgenes conferred the GR-induced longevity to *aak-2* mutant animals (Fig. 6e, f; Supplementary Table 6f). This data suggests that *nhr-49* functions at least genetically downstream of *aak-2* for GR-induced longevity. Due to lack of direct evidence, our genetic data do not exclude the possibility that AAK-2a and NHR-49 function to promote metabolic remodeling for GR-induced longevity in parallel.

Given that neuronal AMPK activity was sufficient for GR-mediated longevity, we next asked in which tissues NHR-49 functioned for GR-

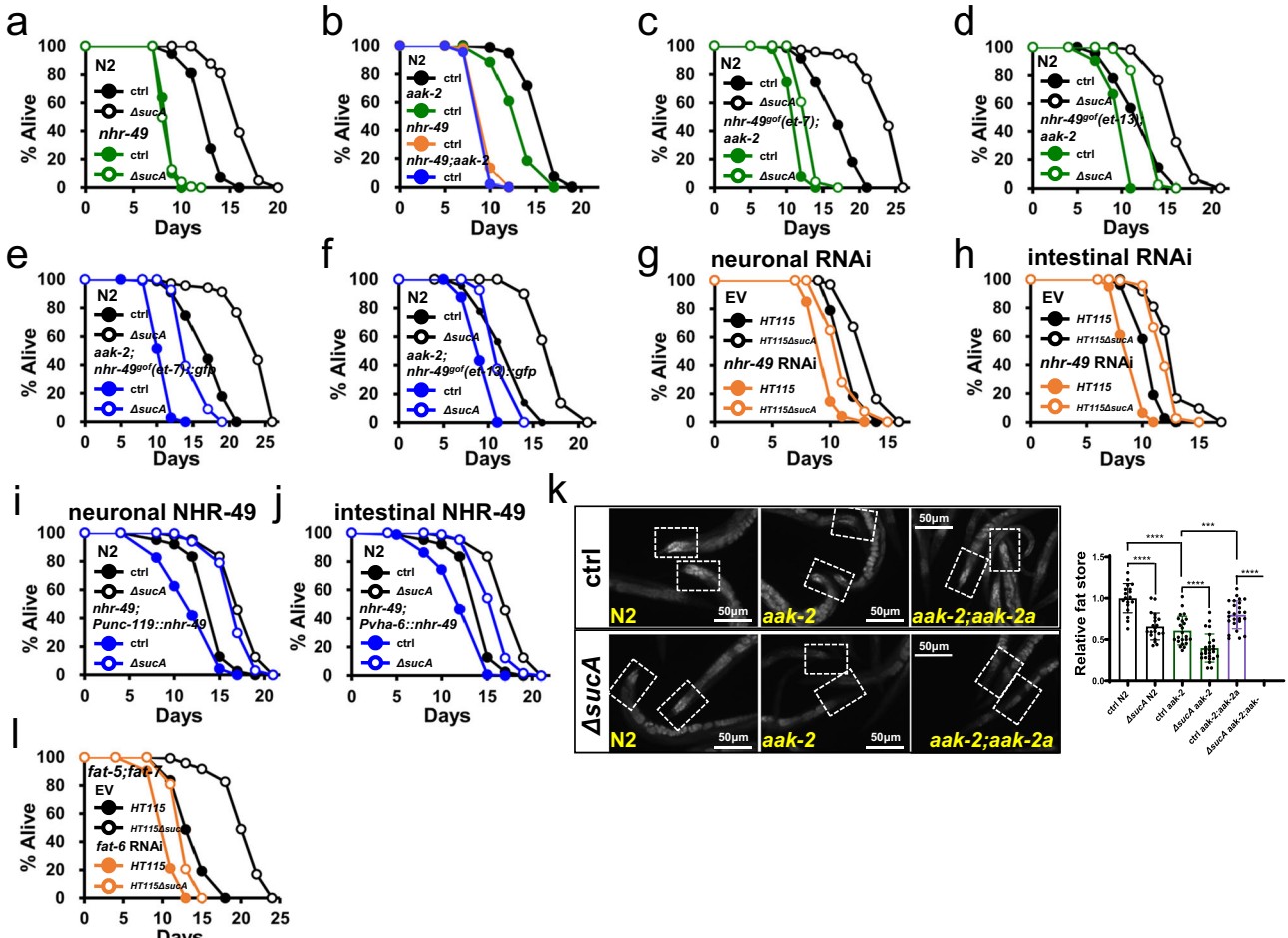

**Fig. 6 | NHR-49 is required for GR-induced longevity. a** Lifespan showing that NHR-49 is required for GR-induced longevity. *nhr-49 (nr2041)* completely abolished GR-induced longevity ($P = 1$, $P$ value determined by two-tailed Student's $t$ test). Results from one of three independent experiments are shown. **b** Lifespans showing that the effects of *aak-2* and *nhr-49* mutation on lifespan are not additive. The mean lifespan of *nhr-49;aak-2* is comparable to that of *nhr-49* animals ($P = 1$, $P$ value determined by two-tailed Student's $t$ test). Results from one of three biological independent experiments are shown. **c, d** Lifespans showing that *nhr-49* gain-of-function (*gof*) mutations, *et7* (**c**) and *et13* (**d**), in *aak-2(ok524)* animals gain the GR-mediated longevity ($P < 1.0 \times 10^{-10}$, $P$ value determined by two-tailed Student's $t$ test). Results from one of three biological independent experiments are shown. **e, f** Lifespans showing that *nhr-49gof* transgenes, *et7* (**e**) and *et13* (**f**), rescued GR induced longevity in *aak-2* mutants ($P < 1.0 \times 10^{-10}$, $P$ value determined by two-tailed Student's $t$ test). Results from representative experiments are shown with additional repeats. **g, h** Lifespans showing that neither neuronal nor intestinal *nhr-49* RNAi reduced the lifespan extension by GR diets ($P < 1.0 \times 10^{-10}$, Student's $t$ test). Tissue-specific RNAi animals are used for neuronal *nhr-49* RNAi knockdown (**g**) and

intestinal *nhr-49* RNAi knockdown (**h**). Results from representative experiments are shown with additional repeats. **i, j** Lifespans showing that restoring NHR-49 to either neuron or intestine fully restored the lifespan extension by GR in *nhr-49* mutants ($P < 1.0 \times 10^{-10}$, $P$ value determined by two-tailed Student's $t$ test). The lifespans of *nhr-49(nr2041)* expressing NHR-49 in the indicated tissue are shown; pan-neuronal *nhr-49* (**i**) and intestinal *nhr-49* (**j**). Results from one of three independent experiments are shown. **k** Fat store of animals feeding AL vs. GR diets. Fixed worms are stained with Nile red dye. Intensity of posterior parts of intestine are quantified as fat contents (dotted boxes) using ImageJ software (\*\*\*$P < 0.001$, \*\*\*\*$P < 0.0001$, $P$ value determined by two-tailed Student's $t$ test). Error bars indicate standard error of mean (SEM). Results from representative experiments are shown with additional repeats. ($n \geq 18$ for each condition). **l** Lifespan showing that Δ9 desaturases are implicated in GR-induced longevity. Lifespan extension by GR diets are significantly reduced in *fat-5(tm420); fat-7(wa36); fat-6(RNAi)* animals. Results from one of three independent experiments are shown. Source data are provided as a Source Data file.

mediated longevity. To this end, we used tissue-specific RNAi sensitive strains for *nhr-49* knockdown in neurons and intestinal cells. Surprisingly, none of the tissue-specific knockdowns attenuated GR-induced longevity (Fig. 6g, h; Supplementary Table 6g), suggesting that residual NHR-49 activity in tissues unaffected by RNAi was sufficient to extend the lifespan by GR. To test this, we selectively restored NHR-49 activity to neuronal or intestinal cells of *nhr-49(nr2041)* mutants using *Punc-119/Prab-3* and *Pvha-6*, respectively. Unlike neuron-specific AMPK activity for GR-induced longevity, either neuronal or intestinal rescue of NHR-49 restored GR-induced longevity in *nhr-49(nr2041)* mutants (Fig. 6i, j; Supplementary Fig. 5h; Supplementary Table 6h). Interestingly, there are multiple publications that NHR-49 in a single cell type is sufficient for a non-cell autonomous effect on lifespan[22,79,80]. These

imply that some of downstream effects of NHR-49 are non-cell autonomous. As previously suggested[81], it is possible that NHR-49 in one tissue functions non-cell autonomously via lipid exchange between distal cells. Taken altogether, we demonstrate that NHR-49 modulates GR-induced longevity, communicating between the tissue of expression and the rest of body in the organismal context.

## Δ9 desaturases are required for GR-induced longevity

Under low energy status, AMPK maintains energy homeostasis through remodeling fat metabolisms across species[82]. NHR-49 functions as a key modulator of these processes by regulating genes involved in fat oxidation and fatty acid desaturation[77,79]. First, we measured the fat content of animals on AL vs. GR diets using Nile red

and Oil Red O staining. Wild-type control N2 animals stored less fat on GR diets (Fig. 6k; Supplementary Fig. 5f), likely due to enhanced fat usage. As reported[83], *aak-2* mutants stored significantly less fat on AL diets, compared to N2 control (Fig. 6k; Supplementary Fig. 5f). However, fat content in *aak-2(ok524)* mutants was further reduced by GR diets compared to AL diets (Fig. 6k; Supplementary Fig. 5f), suggesting that enhanced fat usage under GR conditions was independent of AAK-2 and might not directly cause lifespan extension. Notably, AAK-2a restored fat storage in the intestine (Fig. 6k; Supplementary Fig. 5f), demonstrating that neuronal AMPK enhanced fat stores in peripheral tissues non-cell autonomously. To validate fat content analyzed by staining methods, we conducted biochemical assays for triacylglycerol (TAG) levels. Consistent with dye staining data, GR diets reduced TAG levels in all tested animals, including wild-type control, *aak-2(ok524)* mutants, and *aak-2(ok524);aak-2a* transgenic animals (Supplementary Fig. 5g). To examine whether GR-enhanced fat oxidation might not causally extend lifespan, GR-induced longevity was assessed when fat oxidation was inhibited. For stored fat to be mobilized, TAG is hydrolyzed by adipose triglyceride lipase 1 (ATGL-1) and hormone sensitive lipase homolog (HOSL-1) in *C. elegans*[84]. We found that GR diets similarly extended *C. elegans* lifespan whether *atgl-1* or *hosl-1* was knocked down or not (Supplementary Fig. 5i, j; Supplementary Table 6i). The mitochondrial carnitine palmitoyltransferase-1 (CPT-1) transports long chain fatty acids into the mitochondria for fatty acid oxidation[85]. The *cpt-1* RNAi did not affect GR-induced longevity (Supplementary Fig. 5k; Supplementary Table 6j). Together, these data demonstrate that enhanced fat oxidation accompanied by GR diets does not causally mediate lifespan extension by GR.

In addition to β-oxidation, NHR-49 promotes fatty acid desaturation via Δ9 desaturases[77]. To investigate whether fatty acid desaturation might be required for GR-mediated longevity, we assessed their involvement in GR-induced longevity using Δ9 desaturase mutants. *C. elegans* has one C16 Δ9 desaturase, FAT-5, and two C18 Δ9 desaturases, FAT-6 and FAT-7[86]. Due to their potential redundancy, we measured the lifespans on AL vs. GR diets of all possible combinations using mutants or RNAi as follows; *fat-5(tm420)*, *fat-6(tm331)*, *fat-7(wa36)*, *fat-5(tm420);fat-6(tm331)*, *fat-5(tm420);fat-7(wa36)*, *fat-7(wa36);fat-6(RNAi)*, and *fat-5(tm420);fat-7(wa36);fat-6(RNAi)*. Interestingly the slow growth phenotype of *fat-6(tm331);fat-7(wa36)* double mutants[87] was significantly exacerbated by GR diets, implying that fat remodeling was essential under GR conditions. As *fat-5;fat-6;fat-7* triple mutant is lethal[88], *fat-6* was post-developmentally knocked down by RNAi in *fat-5(tm420);fat-7(wa36)* animals. GR diets extended the lifespan of single *fat* mutants and all three double mutant (or RNAi) combinations, similar to control animals (Supplementary Fig. 5l–q; Supplementary Table 6k). However, we found that *fat-6* RNAi reduced the GR-mediated longevity in *fat-5(tm420);fat-7(wa36)* animals by 61.03 % (Fig. 6l; Supplementary Table 6l), demonstrating that fatty acid desaturation was implicated in GR-mediated longevity. Notably, oleic acid, a monounsaturated fatty acid (MUFA), is associated with lifespan extension in *C. elegans* under various conditions, such as ER stress, germ cell loss, altered chromatin, and low temperature[31,32,89,90]. Given these pro-longevity effects of MUFA, our data demonstrate that GR prolongs longevity by remodeling the fat composition through Δ9 desaturases.

## GR enhances membrane fluidity to prolong longevity

UFAs are abundant fatty acids in cell membranes, critical for the membrane fluidity with their physical properties, double bonds, interfering with the tight stacking of membrane lipids[35]. Any one of C16 or C18 Δ9 desaturases functions sufficiently for GR-mediated longevity (Supplementary Fig. 5l–q; Supplementary Table 6h), implying that a specific lipid species might not be necessary for GR-mediated longevity. Instead, any type of UFA could mediate the beneficial effects by GR. In *C. elegans* glucose supplementation promotes the saturation of membrane lipids, hence membrane rigidity[37], which might contribute to high glucose-induced short lifespan. In addition, membrane fluidity decreases with age in mammals[40,41]. Therefore, we hypothesized that GR diets might enhance membrane fluidity to promote longevity. To address this, we measured in vivo membrane fluidity in animals fed AL vs. GR diets using fluorescence recovery after photobleaching (FRAP). Using FRAP on transgenic worms expressing GFP incorporated in the intestinal membrane[37], membrane fluidity is directly determined as the rate and capacity of fluorescence recovery in an area bleached by a laser. Remarkably, GR diets promoted membrane fluidity in wild-type background as shown by shorter $T_{half}$, compared to AL diets (Fig. 7a; supplementary videos 1, 2). Moreover, the maximum recovered fluorescence was greater on GR diets. This indicates that not only fluidity is improved but also mobile membrane fraction increases by GR diets. C-Laurdan dye is commonly used to measure membrane fluidity[91–93]. It exhibits the spectral shift in emission spectrum according to the levels of membrane order, enabling a straightforward method to monitor the membrane fluidity. We found that the spectrum of C-Laurdan was shifted from solid ordered phases to liquid-disordered phases by GR diets, as shown by generalized polarization (GP) index[94] (Fig. 7b, c). Taken all, FRAP and C-Laurdan analyses reveal that GR diets promote membrane fluidity in *C. elegans*. Given that fatty acid desaturases were implicated in GR-induced longevity, we asked whether GR diets increased levels of unsaturated fatty acids in *C. elegans*. Using UPLC/QTOF mass spectrometry, we observed a slight increase in PUFA in free fatty acids by GR diets, although it was not significant (Supplementary Fig. 6a; Supplementary Table 7). Although GR diets caused no statistically significant change in the level of individual fatty acids in phosphatidylethanolamine (PE) and phosphatidylglycerol (PC), some PCs in which the acyl groups contained multiple double bonds were increased by GR diets, such as PC (34:2), (36:5), (38:3), (38:5) and (40:5) (Supplementary Fig. 6d; Supplementary Table 7). Given that the SFA content of cellular membranes is resistant to dietary challenges[95], these small changes might collectively contribute to GR-induced membrane fluidity.

Importantly, GR-enhanced membrane fluidity was completely abrogated in *aak-2(ok524)* mutants, with no effect on the rate and capacity of fluorescence recovery by GR diets (Fig. 7d; supplementary videos 3, 4). Thus, AMPK is a crucial modulator of membrane fluidity in response to glucose restriction. This is in contrast to the fact that GR further enhanced fat oxidation even in the *aak-2(ok524)* background (Fig. 6k). Next, we asked whether the AAK-2a isoform could rescue the GR-induced membrane fluidity. Remarkably, *aak-2(ok524);aak-2a* animals completely restored both the recovery rate and maximum capacity of the membrane fluidity in intestinal cells (Fig. 7e; supplementary videos 5, 6). These data suggest that GR/neuronal AMPK prolongs longevity by promoting the membrane fluidity in peripheral tissues. PAQR-2 is an orthologue of mammalian adiponectin receptor AdipoR2 and is crucial to control membrane fluidity[37,96]. Therefore, we asked whether PAQR-2 was required for GR-mediated longevity. Remarkably, introducing the *paqr-2(tm3410)* mutation completely abolished GR-mediated longevity (Fig. 7f; Supplementary Table 6m), indicating that adjusting membrane fluidity is critical for GR-mediated longevity.

Of note, adipoR2 is shown to regulate lipid metabolism via PPARα signaling pathways in mice[95,97]. To investigate whether NHR-49^*gof* could bypass the requirement of PAQR-2 in GR induced longevity, we introduced the *nhr-49* gain-of-function (*gof*) mutations, *et7* and *et13*, in *paqr-2(tm3410)* animals. Similar to *nhr-49^gof;aak-2* animals, *nhr-49^gof;paqr-2* animals exhibited lifespan extension by GR diets (Fig. 7g, h; Supplementary Table 6n). This data suggests that NHR-49 may function at least genetically downstream of PAQR-2 for GR-induced longevity. Of note, the *nhr-49* mutations in *paqr-2* mutants severely impaired the general fitness of *C. elegans*, resulting in the inability to conduct lifespan assays. These data imply that *nhr-49* and *paqr-2* may

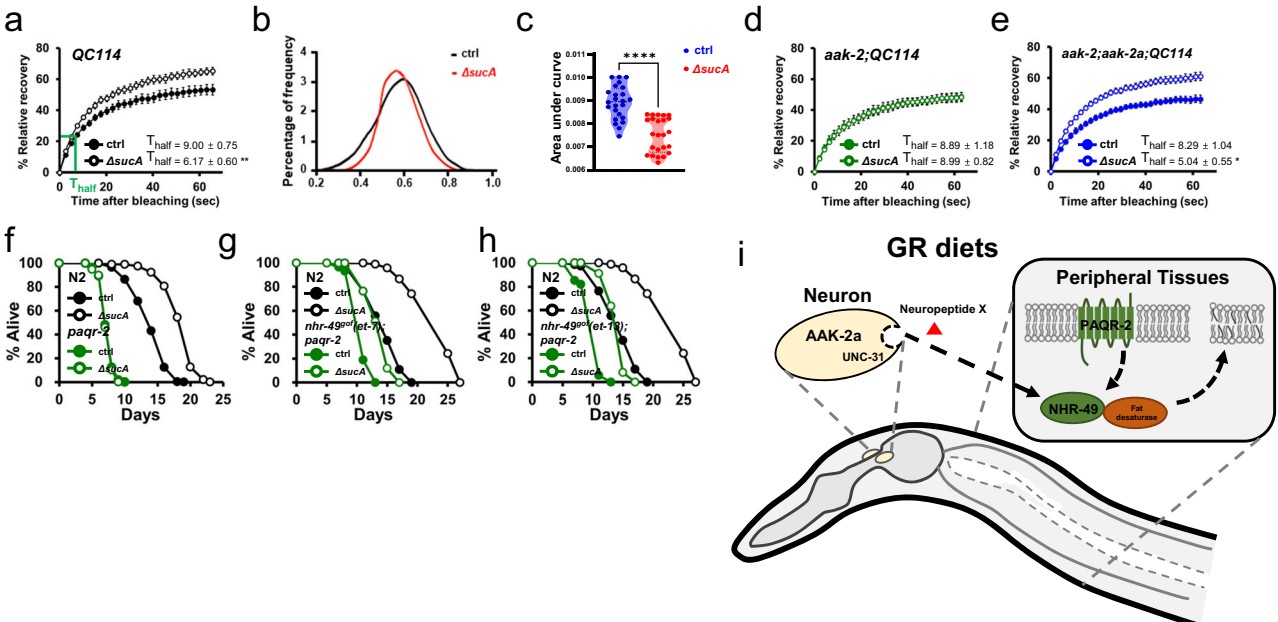

**Fig. 7 | GR diets promote membrane fluidity. a** FRAP analysis showing that GR diets enhance the membrane fluidity. Intestinal membrane fluidity is measured using control animals expressing membrane incorporated GFP (QC114). Average $T_{half}$ values (the time when half of maximum recovery is reached) are presented. Error bars indicate SEM. Results from representative experiments are shown with additional repeats ($n \geq 9$ for each condition). See also Supplementary Movie 1-2. **b**, **c** C-Laurdan staining showing that GR diets promote membrane fluidity in *C. elegans*. Distribution of the average GP index values are shown from multiple images for AL vs. GR diets ($n = 23$, 24 respectively) (**b**). Reduced area under the curve (AUC) values demonstrate that *C. elegans* on GR diets has more fluidic membrane ($P < 0.0001$, $P$ value determined by two-tailed Student's $t$ test). **d**, **e** FRAP analysis showing that GR diets enhance the membrane fluidity in an AAK-2a dependent manner. Intestinal membrane fluidity is measured using animals expressing membrane incorporated GFP (QC114) in *aak-2(ok524)* (**d**) and *aak-*

*2(ok524);aak-2a::gfp* animals (**e**). Average $T_{half}$ values (the time when half of maximum recovery is reached) are presented. Error bars indicate SEM. Results from representative experiments are shown with additional repeats ($n \geq 8$ for each condition). See also Supplementary Movie 3–6. **f** Lifespan showing that *paqr-2* is required for GR-induced longevity. *paqr-2(tm3410)* completely abolished GR-induced longevity ($P = 0.653$, $P$ value determined by two-tailed Student's $t$ test). Results from one of three independent experiments are shown. **g**, **h** Lifespans showing that *nhr-49* (*gof*) mutations, *et7* (**g**) and *et13* (**h**), in *paqr-2(tm3410)* animals gain the GR-mediated longevity ($P < 1.0 \times 10^{-10}$, $P$ value determined by two-tailed Student's $t$ test). Results from representative experiments are shown with additional repeats. **i** Model showing that GR prolongs longevity non-cell autonomously by promoting membrane fluidity through neuronal AMPK/neuropeptide/PAQR-2/NHR-49/Δ9 desaturase. Source data are provided as a Source Data file.

play non-overlapping roles in general worm health. Taken altogether, we demonstrate that GR prolongs longevity by promoting membrane fluidity via signaling including neuronal AMPK, neuropeptide, PAQR-2/AdipoR, NHR-49/PPARα, and Δ9 desaturases (Fig. 7i).

## Discussion

Dietary restriction delays aging and aging-associated diseases in multiple organisms. Nonetheless, its unsustainability and possible side-effects limit DR application for humans. Therefore, lowered intake of particular nutrients could be a realistic alternative for DR-mediated health benefits. However, studies of the effects on longevity by defined nutrients have been limited, due to the lack of a relevant model.

Glucose is the best known nutrient to affect aging and aging-associated diseases. However, the molecular mechanisms underlying its effects on aging are poorly understood. It has been proposed that a glucose-rich diet activates the IIS pathway[9], generates toxic advanced glycation end products[10], and promotes the saturation of membrane lipids[37], which may collectively mediate glucose toxicity. Glucose is known to reduce the lifespan in *C. elegans*[9]. Glucose likely functions directly on worms, as most of its effects still remain when supplied on $\Delta PTS^{Glc}$ *E. coli* deficient in glucose uptake. However, glucose toxicity is abrogated in *paqr-2* mutant *C. elegans* when the dietary *E. coli* is unable to convert the glucose into SFAs[38]. Therefore, at least in certain circumstances, glucose toxicity can be caused by gut microbes. In this study, glucose supplementation significantly increased the saturation of free fatty acids in *C. elegans* when fed $\Delta PTS^{Glc}$ *E. coli* (Supplementary

Fig. 6e), demonstrating that glucose could directly affect the lipid composition of *C. elegans*.

Studies on the longevity effects of GR diets are even less common. To mimic GR diets, the inhibitor of glycolysis 2-DG, or knocking down/blocking glycolytic enzyme has been utilized[8]. However, 2-DG represents only a subset of GR-induced phenotypes and exhibits glycolysis-independent effects, such as cell cycle arrest, apoptosis, ER stress, and AKT phosphorylation[98–101]. Therefore, a bona fide GR regimen will be valuable as it enables studies on the exact molecular mechanisms by which glucose modulates longevity and can provide a novel target for developing a GR mimetic.

Our previous genome-wide lifespan screens identified several *E. coli* mutants that extend the lifespan of *C. elegans*[44]. Among them, we demonstrate that $\Delta sucA$ *E. coli* mutants extend *C. elegans* longevity by serving as GR diets. Further, GR diets recapitulate DR benefits, such as delayed onset of Aβ42-induced pathologies and enhanced stress resistance. In contrast to DR, GR diets did not exhibit any loss of fitness such as the slow growth and lowered fecundity.

In mammals, the brain utilizes ~20% of the entire glucose supply[102], suggesting that neurons are sensitive to glucose availability and should convey the signal to peripheral tissues to maintain organismal energy homeostasis. In this study, we demonstrate that a new AMPK isoform, AAK-2a, plays a pivotal role in the neurons, orchestrating GR-induced organismal responses in *C. elegans*. GR/neuronal AAK-2a prolongs longevity non-cell autonomously via pro-longevity neuropeptide signaling. Moreover, the GR-induced pro-longevity signal is dominant over the CA-AMPK/CRTC-1-mediated pro-aging

signal[22], as the GR diets extend the lifespan of constitutively active CRTC-1 (S76A, S179A) animals.

GR/AAK-2a mediated longevity requires NHR-49. NHR-49/PPARα orthologue regulates fat metabolism with MDT-15, *C. elegans* orthologue of mammalian mediator subunit MED15[78,103] and SREBP ortholog SBP-1[104]. MDT-15 was also required for GR-mediated longevity (Supplementary Fig. 5d; Supplementary Table 6c). However, when *sbp-1* knocked down post-developmentally, *ΔsucA E. coli* reproducibly extend the lifespan (Supplementary Fig. 5e; Supplementary Table 6o). These data imply that *sbp-1* might not be involved in GR-induced longevity. Due to insufficient RNAi knock-down issues, however, we cannot rule out the possibility that SBP-1 is still involved in GR longevity.

While AAK-2 functions exclusively in neurons for GR-induced longevity, either neuronal or intestinal NHR-49 is sufficient for GR-induced longevity. Interestingly, similar to NHR-49, the expression of *paqr-2* in a single tissue or alternatively in any one tissue is sufficient to suppress systemic *paqr-2* mutant phenotypes[81]. The authors show that cell membrane homeostasis is maintained non-cell autonomously via lipid exchange between distant cells. It is possible that NHR-49 in one tissue modulates the systemic effect of GR-induced longevity in a similar manner. This could provide a plausible explanation that rescue of NHR-49 in either neurons or intestinal cells is sufficient for GR-mediated longevity.

In peripheral tissues, GR diets promote the fluidity of cellular membranes, which also requires neuronal AAK-2a. Since lipid regulators such as PAQR-2/AdipoR2, NHR-49/PPARα, and Δ9 desaturases are required for GR-induced longevity, GR diets promote whole-body lipid homeostasis including membrane fluidity.

Of interest, the membrane fluidity decreases in various cells from aged mice[39,40] and human diabetes patients[105], suggesting that the maintenance of membrane fluidity may promote longevity across species. Further, the PAQR-2 pathway functions not only for SFA tolerance[106] but also for GR-induced longevity. Taken together, our data provide a regulatory network showing how organisms activate an intrinsic pro-longevity program in response to environmental factors, such as GR (Fig. 7i). Given the conservation of factors involved in GR-mediated longevity, our studies deepen understanding on aging and age-associated diseases and identify potential therapeutic targets to improve the healthspan in humans.

## Methods

### *C. elegans*

*C. elegans* strains were maintained at 20 °C on *E. coli* using standard cultivation techniques[107], unless otherwise described. *C. elegans* strains were purchased from the Genetic Genome Center or were gifts from other laboratories. The transgenic animals were generated using standard microinjection methods[108]. Briefly, to generate *aak-2* transgenic animals, *aak-2* DNA construct was injected into *aak-2;unc-119* double mutants along with co-injection marker, *unc-119+* (200 ng/μl). Double mutants were generated using standard genetic methods as described below. In brief, to generate *nhr-49; aak-2* double mutants, *aak-2* males were mated to *nhr-49* hermaphrodites. Several F1 progenies were singled on individual plates. F2 progenies were selected and tested for homozygous *aak-2* mutation by PCR. From multiple F3 progenies, homozygous *nhr-49* mutants were selected by PCR. The primers used for genotyping are listed in Supplementary Table 8. All strains were backcrossed at least four times to control strains. The strains are listed in Supplementary Table 9.s

### *E. coli* strains

*E. coli* mutants were sourced from an *E. coli* single-gene deletion mutant library (Keio collection). Double or quadruple *E. coli* mutants were generated as previously described[109].

**DNA construction.** To generate the *aak-2a* promoter construct, a SphI/XbaI fragment containing the 2.4 kb upstream region of *aak-2a* was cloned into the pPD95.75 vector. To generate the *aak-2a* cDNA constructs, *aak-2a* cDNA was cloned into the pUC18 vector using mutagenic primers with XbaI/KpnI restriction sites. The *aak-2a* cDNA was subcloned into pPD95.75-P*aak-2a* using XbaI/KpnI. To generate the *aak-2c* promoter construct, a PstI/SalI fragment containing the 3.6 kb upstream region of *aak-2c* was cloned into the pUC18-*aak-2a* vector. The P*aak-2c::aak-2c* ORF was subcloned into pPD95.75 using PstI/KpnI. The *aak-2c* promoter was cloned into pPD95.75 *aak-2a* to make *aak-2a/c* using NheI/XbaI. To generate tissue specific expression vectors, 1.3 kb *unc-119* promoter was amplified using mutagenic oligos containing HindIII/SphI and cloned into pPD95.75 vector. The 2.4 kb *mca-1* promoter containing SphI/PstI and 1.2 kb *vha-6* promoter containing HindIII/SalI were cloned into pPD95.75. An *aak-2a* cDNA was subcloned into each tissue specific expression vector. To generate tissue specific NHR-49 expression vectors, a *nhr-49* cDNA was subcloned into pPD95.75P*unc-119*, pPD95.75P*rab-3* and pPD95.75P*vha-6* vectors using SalI/SmaI. The primers used for the DNA constructs are listed in Supplementary Table 8.

**Lifespan assay.** Lifespan assays were performed at 25 °C, unless mentioned otherwise. Synchronized worms were prepared by 3 hour-egg laying or egg preparation followed by L1 arrest. The worms were allowed to grow for several days until they reached the young adult stage. Synchronized young adult stage worms were treated with 0.1 mg/mL FUDR to prevent the proliferation of the progeny. The worms were scored as dead or alive by tapping them with a platinum wire every 2-3 day. The worms that died of vulval rupture were excluded from the analysis. Lifespan plates were freshly prepared with overnight culture of *E. coli* in LB media. For glucose-depleted *E. coli* such as *ΔsucA*, *ΔacnAΔacnB*, *ΔicdΔaceBA*, overnight culture was concentrated 15 times before seeding on NGM plates, to prevent worms from starving during lifespan assays. All lifespan assays were repeated at least three times, unless mentioned otherwise. Statistical analyses and *p*-values were calculated using the log-rank (Mantel-Cox) method, through the OASIS application (http://sbi.postech.ac.kr/oasis2/). Glucose, αKG, lysine, pyruvate, or glycerol was freshly dissolved in distilled water, followed by filter sterilization. All supplements were added on plates one day before use.

**Paralysis assay.** For the paralysis assay, CL2006 animals expressing the human Aβ42 were utilized as previously described[61,110]. In brief, synchronized L1 worms were grown on the BW25113 control *E. coli* (AL diets) or *ΔsucA E. coli* mutants (GR diets) until they reached the young adults at 20 °C. To paralyze worms, plates were moved to 25 °C and monitored until all worms were paralyzed. The worms were scored every other day as paralyzed when they did not respond upon tapping. One of representative data was shown from 3 biological replicates.

**Brood size assay.** Brood size assays were conducted as previously reported[44]. In brief, total 12 L4 stage N2 worms were moved to individual plates seeded with AL or GR diets and allowed to lay eggs for 24 h at 20 °C. Each worm was transferred to a new plate every 24 h for 5 d. The number of progeny was counted when the worms reached the L4 stage. For consecutive brood size analysis, the same procedures were repeated using worms previously grown on AL diets vs. GR diets. One of representative data was shown from three biological replicates.

**Thermotolerance assay.** Thermotolerance aassay was conducted as previously reported[111]. In brief, synchronized wild-type N2 worms were grown at 20 °C, until they reached the L4 stage, followed by an upshift to 37 °C. The plates were removed at the indicated times and the worms were allowed to recover for -12 h. The plates were then scored and discarded. Heat stress-induced mortality was determined by

tapping the worms with a platinum wire to check motility. Statistical analyses was calculated through the OASIS application (http://sbi.postech.ac.kr/oasis2/). One of representative data was shown from three biological replicates.

**RNAi knockdown.** Ahringer RNAi collection was used for RNAi knockdown of specific genes. If necessary, RNAi vectors were constructed using classical cloning techniques. RNAi knockdown conducted as previously reported[44]. In brief, HT115 bacteria with L4440 vector encoded target-gene sequence were grown at 37 °C overnight in LB media, with 50 µg/mL ampicillin and 10 µg/mL tetracycline. 1% overnight culture was inoculated in LB media containing 50 µg/mL ampicillin, and incubated for 3 hours. The culture was concentrated 10-fold and used to seed NGM plates containing 50 µg/mL ampicillin and 1 mM IPTG. For tissue-specific RNAi experiments, TU3410 and VP303 were used for neuron- and intestine-specific RNAi, respectively.

**Metabolite analysis.** Bacteria were cultured overnight in LB media, harvested, and washed with M9 solution two times. Thereafter, the bacteria were diluted with M9 solution and seeded on 90 mm NGM plates. After 3–4 day of growth, the bacteria were harvested using a scraper, suspended in pre-chilled 2X quenching solution (−20 °C, 40% ethanol, 0.8% NaCl), and centrifuged at −20 °C and 3,000 g for 10 min. After discarding the supernatant, the bacterial pellet was frozen using liquid nitrogen and stored at −80 °C until further analysis. Polar metabolites were extracted from cells using three freeze-thaw cycles in the presence of 2 mL of cold 80% aqueous methanol. After centrifugation, the supernatant was transferred to a new EP tube and 500 µL of chloroform and 1 mL of distilled water were added to the remaining pellet. Next, the solution was vortexed and centrifuged, and then, the supernatant was combined with the supernatant obtained from the previous step. The supernatant containing polar metabolites was vacuum dried and resuspended in 600 µL of 0.1 M sodium phosphate-buffered deuterium oxide (pH 7.0) containing 0.1 mM 3-(trimethylsilyl) propionic-2,2,3,3-d$_4$ acid (TSP-d$_4$) (Sigma Aldrich). $^1$H-NMR spectra were measured using an 800-MHz NMR instrument. Briefly, One-dimensional (1D) nuclear Overhauser enhancement spectroscopy (NOESY)-PRESAT pulse sequence was applied to suppress the residual signal in water. Total 256 transients were collected into 64,000 data points using a spectral width of 16393.4 Hz with a relaxation delay of 4.0 s and an acquisition time of 2.00 s. NMR spectra were phased and baseline corrected using Chenomx NMR suite version 7.1 (Chenomx Inc., Edmonton, Alberta, Canada). Identification of the metabolites was performed using 2D NMR TOCSY and HSQC and spiking experiments. Quantification of metabolite was accomplished using an 800MNz library to determine the concentration of individual compounds using sodium 3- (trimethylsilyl) propionate- 2,2,3,3- d4 (TSP) as an internal standard. Data file was transferred to MATLAB (R2006a; Mathworks, Inc., Natick, MA, USA) and all spectra were aligned by correlation optimized warping (COW) method[112].

**Lipidomic analysis.** Lipids were extracted from samples using three freeze-thaw cycles in the presence of 1.6 mL of cold methanol/chloroform mixture (1/1, v/v). After centrifugation, the supernatant was transferred to a new EP tube. Next, the supernatant containing the lipids was nitrogen dried and resuspended in 400 µL (for *C. elegans*) and 4 mL (for *E. coli*) of 80% aqueous isopropanol. Lipids were measured using a Xevo G2-XS Q/TOF mass spectrometry (MS) system coupled with ultra-performance liquid chromatography (UPLC) system (Waters, Milford, MA, USA). An acquity UPLC CSH C18 column (1.7 µm × 2.1 × 100 mm) was used for lipid separation. The mobile phase in positive mode was consisted of 10 mM ammonium formate and 0.1% formic acid in water:acetonitrile (4:6 v/v, solvent A) and 0.1% formic acid in isopropyl alcohol:acetonitrile (9:1 v/v, solvent B). The mobile phase in negative mode was consisted of 10 mM ammonium

formate in water:acetonitrile (6:4 v/v, solvent A) and isopropyl alcohol:acetonitrile (9:1 v/v, solvent B). Samples were eluted at 0.40 ml/min for 20 min. The eluate was analyzed with electrospray ionization (ESI) in positive and negative modes. Mass range was 50–1200 m/z. Leucine-enkephalin ([M + H] +: m/z 556.2771 and [M-H] −: m/z 554.2615) was used as a reference in the lock-spray and introduced by a lock-spray at 5 µL/min for accurate mass acquisition. Raw UPLC/QTOF MS spectral data were preprocessed using Progenesis QI software (Waters, Milford, MA, USA) and normalized to the total ion sum. Lipids were identified using Lipid Maps (www.lipidmaps.org), Human Metabolome (www.hmdb.ca), and Metlin (metlin.scripps.edu) databases. Identification was confirmed using MS/MS pattern and retention time of lipid standards (Avanti Polar Lipids, Alabaster, USA and Sigma-Aldrich, St Louis, USA).

**Fat staining.** For Nile red staining, the worms were synchronized by seeding the eggs on to fresh plates and grown until the L4 stage. The worms were then harvested using the M9 buffer, washed three times with the M9 buffer, and fixed in 1% paraformaldehyde for 10 min. The worms were then freeze–thawed three times using cold ethanol on dry ice. After three washes with the M9 buffer, the worms were dehydrated using 60% isopropanol for 2 min. The fixed worms were incubated in Nile red solution (3 µg/mL Nile red in 60% isopropanol) for 30 min and rehydrated using three washes of the M9 buffer. For Oil Red O staining, the worms were synchronized by seeding the eggs onto fresh plates and grown until the day 1 adult stage. The worms were then harvested using the M9 buffer, washed three times with the M9 buffer, and fixed in 1% paraformaldehyde for 10 min. After three washes with the M9 buffer, the worms were dehydrated using 60% isopropanol for 2 min. The worms were stained in 400µl of saturated Oil Red O solution overnight on a shaker at RT. A minimum of 20 animals was quantified for each conditions. Fat was visualized using the Nikon ECLIPSE Ni-U microscope. Fat stores were quantified using the ImageJ software.

**Triglycerides assay.** Day 1 synchronized adult worms were harvested using the M9 buffer, washed three times with the M9 buffer and suspended in NP40/ddH$_2$O (5% NP40). Samples were sonicated at 20% power, for 30 s. To solubilize triglyceride, lysates were incubated at 90 °C for 5 min, then cooled down at room temperature for 5 min, followed by centrifugation at 24,000 × g for 10 min. Aqueous phases of the samples were collected. Triglycerides were measured using the Triglyceride assay Kit (ab65336, Abcam), according to manufacturer's instructions. The amount of triglyceride was normalized with protein concentration (BCA protein kit, 23228, Pierce). One of representative data was shown from two biological replicates.

**αKG assay.** Worms were grown on NGM plates supplemented with α-KG. Day 3 synchronized adult worms were harvested, followed by washing three times with M9 buffer and sonication (30 s, 20% power). After sonication, lysed worms were centrifuged at 24,000 × g for 10 min at 4 °C.

Aqueous phases of the samples were collected. αKG concentration was measured using the αKG Assay Kit (ab83431, Abcam), according to manufacturer's instructions. The colorimetric signal was measured using the Synergy HT Multi-detection Microplate Reader. One of representative data were shown from two biological replicates.

**Western blot analysis.** The mixed stage worms were collected and washed three times with M9 buffer. Worms were resuspended in RIFA buffer containing protease inhibitor (1 mM PMSF, 1 µg/mL leupeptin and aprotinin) and phosphatase inhibitors cocktail (4906837001, Roche), followed by sonication. After sonication, lysed worms were centrifuged at 24,000 × g for 10 min at 4 °C. Aqueous phases of the samples were collected. The lysates were boiled in SDS sample buffer. Worm lysates were then resolved on 8% SDS-PAGE and western-blotted

using antibody against to GFP (1:1000 diluted, ab290, Abcam), Goat anti-rabbit IgG HRP conjugated (1:5000, #31460, ThermoFisher) and *β-actin* (1:5000, ab133626, Abcam). *β-actin* served as the endogenous controls for normalization.

**Quantitative RT-PCR.** RNA preparation and cDNA synthesis were performed according to standard protocols. Total RNA was isolated using an RNeasy Plus Mini Kit (Qiagen). Total RNA (2 μo) was used for cDNA synthesis with a SuperScriptII cDNA synthesis kit (Thermo Fisher Scientific). Quantitative RT-PCR was performed using StepOnePlusTM (Applied Biosystems). in a total reaction volume of 25 μL containing cDNA, primers, and SYBR Master Mix (AppliedBiosystems). Data were normalized to *actin* mRNA levels in each reaction. The sequences of the PCR primers are listed in Supplementary Table 8

**Fluorescence recovery after photobleaching (FRAP) analysis.** FRAP assays were performed using animals expressing the membrane-associated GFP (*Pglo-1*::GFP-CAAX) in intestinal cells with a Zeiss LSM800 confocal microscope and Zen software (Zeiss). The intestinal membranes of L2-3 stage worms were photobleached over circular region (0.75 μm radius) using a 488 nm power laser with 70% laser power transmission for 5 s. The recovered fluorescence of bleached region was recorded every 2.0 s for 90 s. The $T_{half}$ value was defined as the time the fluorescence intensity reached to 50% of maximum fluorescence recovery. Images were acquired with 16 bits image depth and 1024 × 1024 resolution using -0.76 μs pixel dwell settings.

**C-Laurdan staining.** Live *C. elegans* was stained in M9 buffer containing 10mM C-Laurdan dye (N-Methyl-N-[6-(1-oxododecyl)-2-naphthalenyl]glycine) (TOCRIS) for 2 h. Images were acquired Zeiss LSM800 confocal microscope and Zen software (Zeiss). Samples were excited with a 405 nm laser and the emission recorded between 400 and 460 nm (ordered phase) and between 470 and 530 nm (disordered phase). Pictures were acquired with 16 bits image depth and 512 × 512 resolution, using a pixel dwell of -1.52 μs. generalized polarization(GP) values were analyzed using automated ImageJ macro, according to published guidelines[91].

**Reporting summary**

Further information on research design is available in the Nature Portfolio Reporting Summary linked to this article.

## Data availability

The main data supporting the results are available within this article as well as its Supplementary information. Source data including the individual *P* values, whole western blot image and lifespan raw data are provided with this paper. Source data are provided with this paper.

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

## Acknowledgements

We are grateful to Kyuhyung Kim for advice and critical comments on the manuscript. Some of the strains were kindly provided by the *Caenorhabditis* Genetics Center, which is funded by the National Institutes of Health National Center for Research Resources. This research was supported by the National Research Foundation of Korea (NRF) funded

by the Korean Government (Ministry of Science and ICT) (2021R1A2C100891912), the National Research Council of Science & Technology (NST) grant by the Korea government (MSIT) (CRC22011-200), Korea Basic Science Institute (C270200) and the Korea Research Institute of Bioscience and Biotechnology (KRIBB) Research Initiative Program (1711134076).

## Author contributions

E.-S.K. designed and supervised the project. J.-H.J., J.-S.H., S.-H.P., M.P., M.-G.S., and E.-S.K. performed lifespan assays. J.-H.J., J.-S.H., M.-G.S., and E.-S.K. generated *E. coli* mutants and transgenic animals. J.-H.J. performed fat analysis. J.-H.J. and S.M.L. performed FRAP experiments. Y.J., N.K., M.S.K., C.-A.K., and G.-S.H. performed metabolite analysis. E.-S.K., S.K., K.-P. L., K.-S.K., and Y.R.Y. analyzed data. E.-S.K. wrote the paper. All authors read and edited the paper.

## Competing interests

The authors declare no competing interests.
