## [Peer Review File · Nature Communications]

Reviewers' Comments:

Reviewer #1:

Remarks to the Author:

Nature Communications manuscript NCOMMS-21-24510-T

This is a very solid paper replete with convincing genetics experiments using both *E. coli* and *C. elegans* mutants. Several important findings are made, including: 1) Using *E. coli* mutants, a convincing case is made for glucose restriction (GR) diets having a pro-longevity effect in *C. elegans*; 2) It seems implicit from the paper that dietary glucose leads to increased membrane rigidification in *C. elegans*; and 3) Discovery that GR diets prolong longevity via a *aak-2* isoform expressed in neurons that likely signal through neuropeptides to improve systemic membrane fluidity via the *paqr-2/nhr-49/desaturase* pathway.

MAJOR COMMENTS

1. Some lipidomics analysis would really help clarify important points. In the beginning of the results section, an analysis of metabolites in *E. coli* mutants suggests that depletion of their intracellular glucose correlates with longevity in *C. elegans*. An analysis of *E. coli* lipid composition would be useful in this context since it was previously shown that glucose toxicity in *C. elegans* is abrogated (at least in the *paqr-2* mutant) when the dietary *E. coli* is unable to convert the glucose into SFAs (<https://doi.org/10.1371/journal.pgen.1007004>).

2. In the same vein, does the dietary glucose lead to increased SFA in *C. elegans* even when DPTS *E. coli* mutants (unable to uptake the glucose) are used as food? This is an important and quite testable prediction from the present work, i.e. a lipidomics analysis should be informative. Similarly, do GR diets lead to increased levels of unsaturated fatty acids in phospholipids, which would explain the increased membrane fluidity (lipidomics again)?

3. Fig. 2F: Very effective figure. Can the effects of glucose be mimicked by pyruvate? Pyruvate is an intermediate between glucose and acetyl-CoA. As per the Figure 2F, glucose, pyruvate and acetyl-CoA are at reduced levels in all the *E. coli* strains that prolong lifespan. Acetyl-CoA, which can be derived from pyruvate, is particularly interesting because it is a precursor for fatty acid synthesis (which is always synthesized first in saturated form, and the *paqr-2* mutant is sensitive to many metabolites that can be used for fatty acid synthesis; see <https://doi.org/10.1371/journal.pgen.1007004>).

MINOR COMMENTS

1. Title: It took me a couple of reads to realize that "prolong" was in the plural form because of the unusual use of the preceding backslash.

2. Abstract, line 32: Should be "improving" then "ameliorating"

3. Abstract: It would be useful in the abstract to provide some hint as to how glucose is connected to decreased membrane fluidity.

4. Well written and informative introduction. Sets the stage for the rest of the paper.

5. Line 90: Define sDR.

6. Line 381 (and again line 529): Probably should be reworded. Perhaps "... signaling is dominant over the CRTC-1-mediated pro-aging signal."

7. Lines 476-478: The provided videos of the FRAP experiments show that a very large region is bleached (much larger than the indicated circled area). See <https://doi.org/10.21769/BioProtoc.2913> for comparison and a detailed explanation of the FRAP protocol. The excessive area being bleached complicates the interpretation (many membranes besides the target membrane were bleached) but probably does not invalidate the main finding that the GR diets improved membrane fluidity.

8. I think the discussion should bring up the concept of direct vs indirect effects of glucose on *C. elegans*. Indirect: the dietary *E. coli* can convert the glucose into SFAs as shown in (<https://doi.org/10.1371/journal.pgen.1007004>), where *E. coli* mutants unable to convert glucose into SFAs abolished the glucose toxicity in the *paqr-2* mutant (which is extremely SFA-sensitive). Direct: the paper under review implies (though does not demonstrate) an important novel point: glucose can also act directly within *C. elegans* to impact on membrane fluidity, likely because in worms too the glucose can be converted into SFAs. This would provide an interesting explanation for how glucose affects *C. elegans* lifespan even when fed DPTS *E. coli* mutants, as previously shown in <https://doi.org/10.1016/j.cmet.2009.10.003>. Another great point that could be discussed is that the PAQR-2/IGLR-2 pathway is the only essential pathway for SFA tolerance in *C. elegans* (<https://doi.org/10.1016/j.bbalip.2021.158883>; (recently reviewed in <https://doi.org/10.1186/s12944-021-01468-y>) but the present paper shows that it is beneficial to boost this pathway (via GR diet) even in the absence of a membrane-rigidifying challenge.

9. Also, in the future it will of course be very interesting to identify the pro-longevity neuropeptide produced by the *aak-2*-expressing neurons.

10. Also, has *sbp-1* been tested as another possible downstream effector of *aak-2*, perhaps acting in parallel with *nhr-49* to promote desaturase expression? Both *sbp-1* and *nhr-49* seem to act downstream of *paqr-2* according to published studies.

11. Figure 1. I would suggest adopting the conventional color scheme where the color red stands for increased and blue for decreased levels of metabolites. It is the opposite in the current Fig. 1C). Also regarding Fig 1C: Lysine is upregulated in the GR diet. A literature search lead to this paper (doi: 10.1186/s12863-015-0167-2) which says that supplementation of lysine prolongs lifespan and is dependent on AAK-2. Perhaps this could be mentioned.

12. Fig. 3A: Explain what AB42 is (this will not be obvious to many readers).

13. Fig. 6h and j: Not only was the Thalf shorter when the worms were fed the *DsucA* mutant *E. coli*, but also the mobile fraction was increased, which indicates that not only the fluidity is improved but also that actually more (roughly speaking, a greater % area) of the membrane is fluid. Perhaps this should be mentioned in the text and/or figure.

14. Fig. 6l: It should be "fatty acid desaturase", not "fat desaturase".

M&M

15. For many assays (e.g. paralysis assay, brood size, thermotolerance), there is no indication of whether the experiments were repeated.

16. How were supplements (e.g. glucose, alpha-KG) added to the diet?

17. References should be provided for the assays that are performed as per previous publications. There is often not enough experimental details provided here for other labs to repeat the experiments unless a reference is provided that contains more information. I can see from the supplementary tables that most experiments were performed multiple times but it is probably a good idea to mention this in the M&M for each assay, and to indicate that the results from a representative experiment are shown.

18. How many metabolites were identified in the metabolite analysis? Aare all included in the Suppl. Table 2? Incidentally, the bottom-left box of Suppl. Table 2 should be labeled "et cetera" or "others" (and not "et cetra"). Note that pyruvate was just as downregulated as glucose in the relevant *E. coli* mutants, which is interesting in connection to fatty acid synthesis, as mentioned above. What of Acetyl-CoA?

Reviewer #2:

Remarks to the Author:

The manuscript by Jeong et al. is a collection of intriguing findings that describe many of the metabolic effects that result downstream of a chronic glucose restricted regimen that was cleverly engineered by the authors. The main observation revealed by this work is that the lifespan of adult *C. elegans* is extended significantly in a manner that is dependent on the reduction of glucose, providing a new model for understanding how glucose affects various processes that can ultimately shorten lifespan, all the while remaining distinct from current diet based lifespan-extending models, namely dietary restriction and some variations thereof.

Not surprisingly, this extension depends on the activity of AMP-activated protein kinase (AMPK), like some of the previously described models of lifespan extension, however what is surprising is that it requires only a specific isoform of one catalytic subunit (AAK-2) that is expressed in the neurons. Using genetic analysis the authors conclude that the AMPK-mediated effect requires neuropeptide secretion, the activity of a transcription factor, changes in the activity of desaturases and even modifications of membrane fluidity.

The manuscript is very well written and presented and the first section where they describe their GR bacterial strain is really beautifully executed to engineer a novel highly useful tool for future studies. Although most of the initial work is very convincing and their genetic analyses are generally sound, some conclusions are based on limited data (membrane fluidity) and others (overall model) are only loosely supported by data shown in the manuscript.

Although the major conclusions of this manuscript are indeed novel and interesting, the manner in which the data are strung together leaves out major components of the model, the characterisation of which would improve the manuscript considerably. Among these, the authors might consider accounting for many of the factors function downstream of AMPK; how the neuropeptides affect NHR-49 signalling; What tissues require NHR-49 function; why/how would membrane fluidity affect lifespan; how do all these processes interact based on genetic analysis; How do the authors implicate PAQR in this pathway-is it regulated in parallel by AMPK or is its expression enhanced by NHR-49. Is it the target of the purported neuropeptide? None of these findings are coherently linked by experiments provided in this manuscript and thus, although independently are interesting, they are strewn together into a linear genetic pathway leaving major logical gaps between the various observations.

Significant Concerns:

First, in all the cases that different isoforms were indeed ineffective in mediating the lifespan extension, can the authors be sure that each isoform was expressed at levels that are comparable (if they are being compared) in the transgenic animals. Similarly, in RNAi experiments, can you be sure that RNAi targets that are being compared were indeed reduced significantly and in a comparable level in each case.

Expression in the neurons was SUFFICIENT (maybe not required!), for extension, but is there any role for the excretory cell? A similar implication of the excretory cell was also demonstrated in AMPK-dependent lipid rationing via ATGL-1 and both neurons and the excretory system were more recently implicated in establishing germ cell cycle arrest.

The results with *nhr-49* are somewhat confusing. The model would suggest that neuropeptide secretion is essential downstream of AMPK (*aak-2a*) to extend lifespan, however the authors allude to a potential direct target of AMPK (assuming *nhr-49* acts similarly to *ppar-a*) and conclude that everything must function downstream of this transcription factor. But this factor can work either in the neurons (by the way, the *unc-119p* is not the best for strict neuronal expression-it is a bit leaky!) or in the gut. How do the neuropeptides fit in? Is the expression of *nhr-49* sufficient in every or all tissues? How does the autonomous expression of the transcription factor implicated in the cell non-autonomous effect of GR on organismal lifespan, particularly if it is restricted to a single cell type (gut (probably not due to neuroendocrine release of neuropeptides) or neurons? Is there a possibility that there is a parallel pathway that may be functional here that is difficult to assess because the readout (lifespan extension) is not sensitive enough.

The description of membrane fluidity is dependent on one single means of quantification that is not at all a direct readout of lipid stacking or fluidity per se. FRAP is an indirect measure of protein exchange in a given environment that can be governed by many different variables, only one of which is membrane fluidity. Although the authors do reference this approach, so far I am not aware of a study that directly compares biophysical quantification of stacking vs FRAP. FRAP may be a proxy for fluidity, but it really needs to be demonstrated. If the authors believe that all the membranes are affected in the animals based on the "organismal" effect of NHR-49 (assuming the fluidity arises due to NHR-49), then using a more biophysical means like C-Laurdan emission of

determining lipid stacking in membranes would be more convincing, particularly in combination with the FRAP data. This addition would firstly be robust, and finally provide the appropriate support to link FRAP with bona fide biophysical change in the membrane lipid stacking. The link to PAQR again is intriguing but it is just not clear how this receptor fits into the linear genetic framework that the authors are invoking. In mammalian systems, AdipoR is an upstream activator of AMPK and PPAR α so it is hard to imagine how it would become active in this context or is it working in parallel to NHR-49 and the desaturases that could indeed have a direct effect on lipid composition in the membrane(s). The placement of PAQR upstream of the NHR-49/MDT-15 transcription factors seems speculative at best and should be substantiated. If the model the authors propose is correct then any membrane FRAP changes observed would not only be dependent on PAQR but also on NHR-49, MDT-15 and the fat-6 desaturase. Furthermore, if this is true, then a misregulated hyperactive form of PAQR should also extend lifespan independently of AMPK, while the same should be true of overexpressed/liganded NHR-49/MDT-15 or even better extra copies of the FAT-6 desaturase. This should all be independent of AMPK and the GR if indeed sole role of GR is to activate AMPK to set this cascade in motion. Epistasis analysis would help the authors to order their gene activities or place them in linear vs parallel pathways and either confirm or refute the model proposed in Figure 6L.

Minor Issues:

The manuscript was generally very well written, but nevertheless a number of typos and grammatical issues could be fixed in any new version. I have listed a few of these issues herein below, although this is not comprehensive and further proof-reading is encouraged to improve the quality of the manuscript.

Line 45-calorie intake

Line 59-I don't think that this is an accurate statement. I am not sure that one can conclude that AMPK is actively "downregulated" by glucose. It is simply not activated in these energy rich conditions.

Line 62-there does not exist can be replaced by "a"

Line 63-...animal model does not exist at present, that would enable...

Line 73-eukaryotes, such as yeasts

Line 79-whole body metabolism

Line 112/113-cellular physiology impinging upon

Line 151-metabolite profile

Line 158-7-fold; with lifespan fed on particular E. coli...

Line 167-under the assay...

Line 222-"It" could be re-written as "These levels of glucose"...

Line 247-...introducing a mutation in the icd gene

Line 278-suggesting a distinct...

Line 298/299-are the neurons refractory? Do you always use the enhanced RNAi strain when doing neuronal RNAi? The authors do see RNAi phenotypes using neuronal RNAi so they must be at least partially sensitive. Please reference the tissue specific RNAi strains correctly.

Line 315/316-implies...previous AMPK studies utilized the ...isoform

Line 332-I am not sure that any experiment unequivocally removed aak-2a specifically in the neurons where a true requirement aak-2a can be concluded. Neuronal aak-2 is required based on the data with TU3401, but adding back aak-2a by transgenics addresses sufficiency. In fact, aak-2c is also sufficient if driven in the right cells.

Line 361-"endocrinal" should be endocrine

Line 405-nhr-49 has already been shown to be required in neurons to extend lifespan by altering the lipidome.

Line 410-unc-119p is leaky. It is no longer the go to pan-neural promoter for neuron-specific expression

Line 421-Nile red is not conventionally used for lipid quantification-it is less specific than Oil Red O or CARS

Line 486-whether the AAK-2a

Line 511-to mimic GR diets, the inhibitor of glycolysis 2-DG, or knocking down/blocking glycolytic enzymes have been utilized...

Line 527-prolongs longevity.

Some of the findings corroborate work performed by other investigators, particularly with reference to neuronal NHR-49 and its role in lifespan extension, although the authors appear to either be unaware of this or simply overlooked referencing it.

Reviewer #3:

Remarks to the Author:

The work presented in this article presents significant insights into the mechanisms of *C.elegans* longevity associated with glucose restricted diet. By feeding worms with *E.coli* mutants that mimics glucose restriction and therefore increase *C.elegans* lifespan, the authors unravel the specific role in this process of a new AMPK isoform, AAK-2a, in neurons. Involvement of neuropeptide signaling is also shown together with modulation of GR-induced longevity by NHR-49, PAQR-2 and $\Delta 9$ desaturases that promote membrane fluidity. While a large part of the discussion is well supported by the data included in the manuscript, a number of meaningful revisions should be considered before this manuscript is suitable for publication in *Nature Communications*.

- 1) The syntax of the title should be revised to provide a clearly articulated message. The use of symbol "/" is too ambiguous and does not serve well this purpose.
- 2) Overall, the description of methods is far too succinct. When referring to "standard methods" (such as in paragraph "C. elegans"), please at least include references. Some more detailed methods could also be provided in the supplementary material. This concerns most of the methods sections.
- 3) l.171-172: is the actual uptake of α -KG by the worms evaluated? (consumption in the medium or accumulation in the worms)
- 4) l.202-203: At this point in the text, the equivalence to glucose restriction is not demonstrated. This conclusion is premature, and better supported by data presented later in Fig. 2f. The data of Fig 1 only shows that high-glucose diet suppressed the lifespan extension provided by *DsucA*.
- 5) l.248-249: This result is actually not remarkable, but looks rather trivial. In the absence of *icd*, there is no TCA production of α -KG. Without primary substrate for *sucA*, no additional effect should be expected when mutating both *icd* and *sucA* as compared to mutating *icd* alone.
- 6) The actual conclusion of the first part of the paper seems well exposed in lines 255-256: mutants impaired in both TCA and glycolate cycle can possibly serve as low glucose diet and extend worms lifespan. The authors show that there can be 3 ways to do this, by either mutating a) both *acnA* & *acnB*, or b) *aceBA* and *icd*, or c) perhaps more surprisingly, *sucA* alone by draining metabolic fluxes from the 2 pathways into α -KG accumulation. The whole narrative of the first part of the paper is constructed around this later case. While the use of single *DsucA* mutation may stand at the simplest way to mimic GR, the entire focus on *DsucA* in this part is not a straightforward way to deliver the more global conclusion. The metabolites analysis presented in Fig1 supports this narrative, but does not include data for *DacnA-DacnB* or *Dicd -DaceBA* mutants. Such data would provide interesting clues regarding common metabolic features of these long-lived mutants and better support the "estimated metabolic flux" schemes presented in Fig2f. Rather than "estimated", these schemes are somewhat hypotetic as concerns *DacnA-DacnB* and *Dicd -DaceBA* mutants, with no information about metabolites that accumulate in this case.
- 7) Metabolites analyses:
 - l.983: after quenching, the supernatant was discarded. Was this solution checked for metabolite content? Ethanol is not a fixator, but permeabilizes cell membranes. The discarded supernatant is likely to contain significant amounts of extracted metabolites.
 - Precise extraction protocol (or at least reference) should be specified: name of solvents without indication of proportions and total volume is little informative. How were these extracts then dried, and what was the NMR buffer used?
 - what was the quantity of material (number of bacterial cells) used?
 - The detection of tartrate is somewhat surprising in this type of sample. Please specify how annotation was confirmed, and if it relies on ^1H or both $^1\text{H}/^{13}\text{C}$ chemical shifts. Considering the low concentration detected, the corresponding signal must have very low S/N. As this

measurement does not bring anything to the discussion, it is safer not to bring it forward if any ambiguity remains in the annotation.

-PCA analysis of Fig 1b: how many variables were used? was is carried out on the full NMR profiles, or on the set of quantified metabolites?

8)The data of Supp Fig 2b-o shows that effect of some of the lifespan modulators can be either cumulative or counteractive with GR-induced longevity, while in a number of other cases the selected "DR effectors" does not seem to have any impact on worms longevity (regardless of GR condition), such as in the case of *eat-2*, *rsks-1*, *sir-2.1*, *skn-1*, *pdr-1* and *pink-1*. These observations could be further described and commented in the text.

9) In the data of Fig 4i, control N2 animals live considerably longer than in most of the other experiments presented in the paper (including the ones of fig 4h and 4j). Consequently, the sentence l.325-327 stating that *aak2;aak2a* animals lived longer compared to *aak-2;aak2c* on AL diets are not well supported by reliable controls, and a direct comparison would better support this observation.

10) Data in figure 6g show that *fat-6* RNAi on the double mutant *fat-5;fat-7* displays a clearly reduced lifespan as compared to the double mutation alone. The data also show that this effect is well attenuated under GR. This means GR does impact the worms longevity, even in the absence of fatty acid desaturation. This somewhat contradicts the interpretation given by the authors l. 457-459, suggesting that fatty acid desaturation was required for GR-mediated longevity.

11) In figure 6k, the lifespan assay data for *paqr-2* GR worms is not at all visible.

Reviewer #4:

Remarks to the Author:

In this manuscript, the authors investigate the effects of *sucA* deficient *E. coli* deficient on *C. elegans* longevity. They show that *C. elegans* grown on *sucA* mutants are long live relative to those grown on control *E. coli*. Based on metabolomic measurements of *sucA* mutants, examinations of several other mutant combinations, and metabolite add-back experiments, they conclude that this longevity is caused to reduced levels of glucose in *sucA* mutants. In turn, they find that the noted *C. elegans* longevity requires the presence of AA2-a, an isoform of the catalytic subunit of AMPK, in the nervous system. Based on genetic inactivations and reconstitution experiments, they posit that neural AAK-2 regulates the release of an unidentified neuropeptide from the nervous system. Their model implies that this neuropeptide acts on PAQR-2 receptor in peripheral tissues to, in turn, signal to the MDT-15/NHR-49 transcriptional complex. The consequence of this is increased transcription of genes that encode for fat desaturases, which confer longevity to animals by ultimately promoting membrane fluidity.

The appealing aspects of the manuscript include i) use of a bacterial mutant as a way of modulating the *C. elegans* diet, ii) use of 1H-NMR to characterize the metabolite profiles in the bacterial mutants iii) examination of multiple mutants in various longevity pathways, iv) a relatively straightforward and easy to follow writing and data presentation style.

There are, however, a number of serious concerns about the conclusions presented that prevent support for the current publication. These concerns are listed below.

Major concerns:

1) The model shown in figure 6l is a reasonable hypothesis based on mostly genetic data. The genetic data definitely indicate that each of the components (e.g. neural AAK-2a, peripheral NHR-49/MDT-15, PAQR-2) are required for the lifespan increasing effects of *sucA* mutants. While it is reasonable to speculate that these components form the proposed signaling cascade, there is virtually no data that indicates that these genes indeed form such a signaling cascade! Thus, either the writing has to be dramatically changed or additional data provided to support the notion that the genetic requirements define a unified, signaling pathway.

2) As one example of why it matters to distinguish between genetic requirement and a bona fide signaling cascade, considering the role of aak-2 is illustrative: loss of neural aak-2 has previously been reported to mimic the effects of enhanced serotonin signaling. For certain outcomes, loss of aak-2 promotes neuropeptide release. The authors reported that loss of each of AAK-2 and UNC-31, a component of dense core vesicles through which neuropeptides are released, is required for the noted sucA mediated longevity. Thus, it is equally, if not more, likely that what they are observing is the effects of elevated serotonin signaling simply countering the effects of sucA deficiency. As such, without additional evidence, it is hard to distinguish whether neural AAK-2a senses and signals glucose deficiency or that increased serotonin signaling functions in a parallel pathway to reduce lifespan. Similar examples can be given for the other components (e.g. NHR-49, desaturases, PAQR).

3) Concerns about novelty and advance: each of the components of the signaling cascade have already been shown to be important for mediating the lifespan extensions caused by either glucose deprivation or DR. Thus, it is unclear precisely what constitutes the advance of the manuscript. As noted above, there is very little data supporting the notion that the components form a signaling cascade.

4) The notion that sucA mutants confer longevity due to reduced glucose levels is a very reasonable conclusion but not definitive. Definitive results require inhibiting gluconeogenesis at a step, such as PEPCK, that would primarily effect glucose production and not necessarily numerous other metabolites.

Minor Concerns:

1) It is very interesting that the authors cannot attribute the noted longevity to accumulation of alpha-ketoglutarate. They should expand the discussion of this discrepancy. Are there obvious differences in the experimental set up that can account for this?

2) It is good that the authors show that growth on sucA deficient bacteria does not affect C. elegans fecundity. Nevertheless, it is possible that the noted effects are still related to the hermaphroditic nature of C. elegans. To claim that the longevity effects are broadly applicable, they should examine lifespan of males on these bacteria. This will be good to show but not required for the publication. If the experiments are not provided, the writing should, however, be modified to acknowledge this.

Dear Editor:

We appreciate the Reviewers' helpful comments and suggestions. To address the issues raised, we have performed additional experiments and provided further explanations. Responses to the reviewer's comments are presented below.

Responses to Reviewers' comments

We made point-by-point responses to reviewers' concerns. The reviewers' comments (in blue) and our responses (in black, **Bold**) are listed below.

Point-by-point responses.

Reviewer #1 (Remarks to the Author):

Nature Communications manuscript NCOMMS-21-24510-T

This is a very solid paper replete with convincing genetics experiments using both E. coli and C. elegans mutants. Several important findings are made, including: 1) Using E. coli mutants, a convincing case is made for glucose restriction (GR) diets having a pro-longevity effect in C. elegans; 2) It seems implicit from the paper that dietary glucose leads to increased membrane rigidification in C. elegans; and 3) Discovery that GR diets prolong longevity via a aak-2 isoform expressed in neurons that likely signal through neuropeptides to improve systemic membrane fluidity via the paqr-2/nhr-49/desaturase pathway.

MAJOR COMMENTS

1. Some lipidomics analysis would really help clarify important points. In the beginning of the results section, an analysis of metabolites in E. coli mutants suggests that depletion of their intracellular glucose correlates with longevity in C. elegans. An analysis of E. coli lipid composition would be useful in this context since it was previously showed that glucose toxicity in C. elegans is abrogated (at least in the paqr-2 mutant) when the dietary E. coli is unable to convert the glucose into SFAs (<https://doi.org/10.1371/journal.pgen.1007004>).

We welcome the reviewer's suggestions for improving the manuscript and better understanding the mechanisms underlying prolonged longevity by glucose-depleted Δ sucA *E. coli* mutants. As reviewer mentioned, glucose toxicity in *paqr-2* *C. elegans* is dependent on the conversion of glucose into saturated fatty acids (SFA) by *E. coli*. Thus, glucose is not toxic to *paqr-2* mutants grown on Δ P_{PTS} *E. coli* mutants that is deficient in glucose uptake. On the contrary, it has been reported that glucose shortens the lifespan of wild-type *C. elegans* by acting directly on *C. elegans*¹. Therefore, these demonstrated that glucose could affect *C. elegans* physiologies by acting directly on *C. elegans* as well as indirectly through *E. coli*. Although we showed that glucose abolished the lifespan extension of *C. elegans* when grown on Δ P_{PTS} Δ sucA *E. coli* mutants and metabolically inactivated *E. coli* (Fig. 1f, and 1g), glucose-depleted *E. coli* may contain less SFA and more unsaturated FA, which can extend the lifespan of *C. elegans*. To test this idea, we measured the lipid composition of glucose-depleted Δ sucA *E. coli* using UPLC/QTOF mass spectrometry. We found that the SFA/MUFA ratio was not lower in Δ sucA *E. coli*. It was indeed higher, compared to the control (Supplementary Fig. 1i). This data implies that the fatty acid composition in Δ sucA *E. coli* does not determine the lifespan of *C. elegans*.

Line 208-220;

Figure 1

Supplementary Figure 1

2. In the same vein, does the dietary glucose lead to increased SFA in *C. elegans* even when DPTS *E. coli* mutants (unable to uptake the glucose) are used as food? This is an important and quite testable prediction from the present work, i.e. a lipidomics analysis should be informative. Similarly, do GR diets lead to increased levels of unsaturated fatty acids in phospholipids, which would explain the increased membrane fluidity (lipidomics again)?

We welcome the reviewer's suggestions for improving the manuscript. As reviewer suggests,

we performed UPLC/QTOF mass spectrometry using *C. elegans* fed Δ PPTS *E. coli* with or without glucose supplementation. We found that glucose significantly increased the saturation of free fatty acids of *C. elegans* fed Δ PPTS *E. coli* (Supplementary Fig. 6e), while having a minor effect on the saturation of phosphatidylethanolamine (PE) and phosphatidylcholine (PC) of *C. elegans* (Supplementary Table 7). The level of polyunsaturated fatty acids (PUFA) in PC was slightly decreased ($P=0.034$) (Figure below). These data demonstrate that glucose affects the saturation of FA by directly acting on *C. elegans*, depending on the lipid species.

Line 627-633;

Supplementary Figure 6

Similarly, do GR diets lead to increased levels of unsaturated fatty acids in phospholipids, which would explain the increased membrane fluidity (lipidomics again)?

We welcome the reviewer's suggestions for improving the manuscript. As reviewer suggests, we performed UPLC/QTOF mass spectrometry using *C. elegans* fed AL vs. GR diets. Given that GR diets promoted membrane fluidity and fatty acid desaturases were required for GR induced longevity, we reasoned that GR diets might increase the levels of unsaturated FA, compared to AL diets. Lipidomic analysis revealed that GR diets slightly increased PUFA in free fatty acids, although it was not significant (Supplementary Fig. 6a; Supplementary Table 7). Although GR diets have no statistically significant change in the level of individual fatty acids in PE and PC, some PCs in which the acyl groups contained multiple double bonds were increased by GR diets, such as PC (34:2), (36:5), (38:3), (38:5) and (40:5) (Supplementary Fig. 6; Supplementary Table 7). Given that the SFA content of cellular membranes is extremely resistant to dietary challenges², these small changes might collectively contribute to GR induced membrane fluidity.

Line 578-588;

Supplementary Figure 6

3. Fig. 2F: Very effective figure. Can the effects of glucose be mimicked by pyruvate? Pyruvate is an intermediate between glucose and acetyl-CoA. As per the Figure 2F, glucose, pyruvate and acetyl-CoA are at reduced levels in all the *E. coli* strains that prolong lifespan. Acetyl-CoA, which can be derived from pyruvate, is particularly interesting because it is a precursor for fatty acid synthesis (which is always synthesized first in saturated form, and the *paqr-2* mutant is sensitive to many metabolites that can be used for fatty acid synthesis; see <https://doi.org/10.1371/journal.pgen.1007004>).

We welcome the reviewer's suggestions for improving the manuscript. As suggested, we asked if pyruvate could suppress the lifespan extension by $\Delta sucA$ *E. coli*. While glucose restored the growth retardation of $\Delta sucA$ *E. coli* mutants to normal, pyruvate did not restore that of $\Delta sucA$ *E. coli* (Supplementary Fig. 1j). This implies that supplemented pyruvate might not be actively converted to glucose in $\Delta sucA$ *E. coli*, possibly due to consumption by glycolysis/TCA cycle. Concordantly pyruvate did not abolish the lifespan extension by $\Delta sucA$ *E. coli* (Supplementary Fig. 1k). Externally supplemented glycerol can enter gluconeogenesis through dihydroxyacetone phosphate and the upper part of glycolysis. Recently it has been reported that glycerol is a better substrate for gluconeogenesis than pyruvate in fasting mice³. Therefore, we asked whether glycerol could suppress lifespan extension by $\Delta sucA$ *E. coli*. Unlike pyruvate, we found that glycerol enhanced the growth of $\Delta sucA$ *E. coli* (Supplementary Fig. 1j), and the lifespan extension by $\Delta sucA$ was completely suppressed by glycerol (Supplementary Fig. 1l).

Line 274 -289;

Supplementary Figure 1

MINOR COMMENTS

1. Title: It took me a couple of reads to realize that “prolong” was in the plural form because of the unusual use of the preceding backslash.

We appreciate this request for clarification. As suggested, we have changed the title to “A new AMPK isoform mediates glucose-restriction induced longevity non-cell autonomously by promoting membrane fluidity.”

2. Abstract, line 32: Should be “improving” then “ameliorating”

We thank the reviewer for carefully reading the manuscript. As reviewer suggested, we revised the manuscript.

Line 34;

3. Abstract: It would be useful in the abstract to provide some hint as to how glucose is connected to decreased membrane fluidity.

We welcome the reviewer’s suggestions for improving the manuscript. Due to the word limits of the abstract, we provide explanations in the introduction section on how glucose is connected to membrane fluidity.

Line 112-113;

Interestingly glucose rich diets promote the saturation of fatty acids ⁴, hence reducing membrane fluidity, which may contribute to glucose toxicity ⁵.

4. Well written and informative introduction. Sets the stage for the rest of the paper.

We are grateful for the reviewer's comment.

5. Line 90: Define sDR.

We appreciate this comment and apologize for the unclear description. In the revised manuscript, we defined sDR as follows:

Line 89-91;

Of note, *aak-2* is required for only a subset of the DR regimen, in which diluted bacteria were provided on agar plates in *C. elegans* (sDR) ⁶.

6. Line 381 (and again line 529): Probably should be reworded. Perhaps "... signaling is dominant over the CRTC-1-mediated pro-aging signal."

We welcome the reviewer's suggestions for improving the manuscript. As suggested, we revised the manuscript.

Line 454 and 653;

7. Lines 476-478: The provided videos of the FRAP experiments show that a very large region is bleached (much larger than the indicated circled area). See <https://doi.org/10.21769/BioProtoc.2913> for comparison and a detailed explanation of the FRAP protocol. The excessive area being bleached complicates the interpretation (many membranes besides the target membrane were bleached) but probably does not invalidate the main finding that the GR diets improved membrane fluidity.

We appreciate this request for clarification. FRAP assays were performed with a Zeiss LSM800 confocal microscope. The intestinal membranes of worms were photo-bleached over a circular region (0.75 μ m radius ⁷) using a 488nm power laser with 70% laser power transmission for 5 seconds. Since we could clearly observe the fluorescence recovery under the indicated condition, we simply continued to use these experimental conditions.

8. I think the discussion should bring up the concept of direct vs indirect effects of glucose on *C. elegans*. Indirect: the dietary *E. coli* can convert the glucose into SFAs as shown in (<https://doi.org/10.1371/journal.pgen.1007004>), where *E. coli* mutants unable to convert glucose into SFAs abolished the glucose toxicity in the *paqr-2* mutant (which is extremely SFA-sensitive). Direct: the paper under review implies (though does not demonstrate) an important novel point: glucose can also act directly within *C. elegans* to impact on membrane fluidity, likely because in worms too the glucose can be converted into SFAs. This would provide an interesting explanation for how glucose affects *C. elegans* lifespan even when fed DPTS *E. coli* mutants, as previously shown in <https://doi.org/10.1016/j.cmet.2009.10.003>. Another great point that could be discussed is that the PAQR-2/IGLR-2 pathway is the only essential pathway for SFA tolerance in *C. elegans*

<https://doi.org/10.1016/j.bbalip.2021.158883>; (recently reviewed in <https://doi.org/10.1186/s12944-021-01468-y>) but the present paper shows that it is beneficial to boost this pathway (via GR diet) even in the absence of a membrane-rigidifying challenge.

We welcome the reviewer's suggestions for improving the manuscript. As suggested, we mentioned the direct and indirect effects of glucose on *C. elegans* and highlight

PAQR-2 as the potential target for DR-mediated benefit as follows.

Line 626-633;

Glucose is known to reduce the lifespan in *C. elegans*. Glucose likely functions directly on worms, as most of its effects on wild-type animals still remain when supplied to *E. coli* deficient in glucose uptake ¹. However, glucose toxicity is abrogated in *paqr-2* mutant *C. elegans* when the dietary *E. coli* is unable to convert the glucose into SFAs ⁵. Therefore, at least in certain circumstance, glucose toxicity is caused by gut microbes. In this study, glucose supplementation significantly increased the saturation of free fatty acids in *C. elegans* when fed Δ PPTS *E. coli* (Supplementary Fig. 6e), demonstrating that glucose could directly affect the lipid composition of *C. elegans*.

Line 678-684;

The PAQR-2 pathway functions not only for SFA tolerance ⁸ but also for GR induced longevity. Our studies identify the PAQR-2 pathway as a therapeutic target to promote health and lifespan.

9. Also, in the future it will of course be very interesting to identify the pro-longevity neuropeptide

produced by the *aak-2*-expressing neurons.

We are grateful for reviewer's invaluable comments. It will be interesting which neuropeptide is implicated in GR induced longevity. There exist more than 140 predicted neuropeptides in *C. elegans*, including insulin-like peptides, FMRFamide-Like Peptides (FLPs) and Neuropeptide-Like Proteins (NLPs). Despite the well-known function of insulin-like peptides in longevity^{9, 10}, it is relatively unknown whether other neuropeptides play roles in regulating aging. Since AdipoR/PAQR-2 is involved in GR-induced longevity and no adiponectin homolog is identified in *C. elegans*, it will be interesting to find the pro-longevity neuropeptide likely acting through PAQR-2. However, we believe this is out of the scope of this manuscript.

10. Also, has *sbp-1* been tested as another possible downstream effector of *aak-2*, perhaps acting in parallel with *nhr-49* to promote desaturase expression? Both *sbp-1* and *nhr-49* seem to act downstream of *paqr-2* according to published studies.

We welcome the reviewer's suggestions for improving the manuscript and better understanding of the mechanisms underlying prolonged longevity by Δ *sucA* *E. coli* mutants. Since *sbp-1* is an essential gene for a worm's survival, knocking down *sbp-1* from L1 stage leads to sick animals. When *sbp-1* is knocked down post-developmentally, Δ *sucA* *E. coli* reproducibly extend the lifespan. This data implies that *sbp-1* might not be involved in GR induced longevity. Due to insufficient RNAi knock-down, we cannot exclude the possibility that *sbp-1* is still involved in GR longevity.

Line 655-662;

Supplementary Figure 5

11. Figure 1. I would suggest adopting the conventional color scheme where the color red stands for increased and blue for decreased levels of metabolites. It is the opposite in the current Fig. 1C).

We welcome the reviewer's suggestions for improving the manuscript. As suggested, we revised the manuscript.

Figure 1

Also regarding Fig 1C: Lysine is upregulated in the GR diet. A literature search lead to this paper (doi: 10.1186/s12863-015-0167-2) which says that supplementation of lysine prolongs lifespan and is dependent on AAK-2. Perhaps this could be mentioned.

We welcome the reviewer's suggestions for improving the manuscript and better understanding of the mechanisms underlying prolonged longevity by GR diets. As reviewer mentioned, lysine has been shown to extend the lifespan in liquid medium ¹¹. To validate the effect of lysine on longevity, we tested whether supplemented lysine could extend the lifespan on a standard solid agar medium in wild type *C. elegans*. Lysine did not extend the lifespan on agar medium at the concentrations previously known to increase lifespan (Supplementary Fig. 1g, and 1h; Supplementary Table 1g). Further, the lifespan extension by lysine in liquid media is dependent on not only *aak-2* but also *daf-16*. We showed that ΔsucA *E. coli* extend the lifespan in *daf-16* mutants (Supplementary Fig. 2c), indicating that ΔsucA *E. coli* promoted longevity through a distinct mechanism.

Line 182-187;

Supplementary Figure 1

Supplementary Figure 2

12. Fig. 3A: Explain what AB42 is (this will not be obvious to many readers).

We appreciate this request for clarification. As suggested, we add the explanation on A β ₄₂.

Line 309-310;

The 42 amino acids of beta amyloid (A β ₄₂) play a pivotal role in the pathogenesis of AD ¹².

13. Fig. 6h and j: Not only was the Thalf shorter when the worms were fed the *DsucA* mutant *E. coli*, but also the mobile fraction was increased, which indicates that not only the fluidity is improved but also that actually more (roughly speaking, a greater % area) of the membrane is fluid. Perhaps this should be mentioned in the text and/or figure.

We appreciate this request for clarification. As suggested, we revised the manuscript

Line 571-572;

Moreover, the maximum recovered fluorescence was greater on GR diets. This indicates that not only fluidity is improved but also mobile membrane fraction increases by GR diets.

14. Fig. 6I: It should be “fatty acid desaturase”, not “fat desaturase”.

We appreciate this request for clarification. As suggested, we revised the manuscript line 548;

M&M

15. For many assays (e.g. paralysis assay, brood size, thermotolerance), there is no indication of whether the experiments were repeated.

We appreciate this request for clarification. As suggested, we revised the manuscript.

16. How were supplements (e.g. glucose, alpha-KG) added to the diet?

We appreciate this request for clarification. Glucose and α KG were dissolved in distilled water, then sterilized by filtering. Then they were added to NGM plates one day before seeding *E. coli*. Other supplements such as pyruvate, glycerol, and lysine were treated in the same manner. We describe this in the materials and method section.

17. References should be provided for the assays that are performed as per previous publications. There is often not enough experimental details provided here for other labs to repeat the experiments unless a reference is provided that contains more information. I can see from the supplementary tables that most experiments were performed multiple times but it is probably a good idea to mention this in the M&M for each assay, and to indicate that the results from a representative experiment are shown.

We welcome the reviewer’s suggestions for improving the manuscript. As suggested, we provided more experimental details in the materials and method section and indicated the number of times the experiments were repeated.

18. How many metabolites were identified in the metabolite analysis? Are all included in the Suppl. Table 2? Incidentally, the bottom-left box of Suppl. Table 2 should be labeled “et cetera” or “others” (and not “et cetra”). Note that pyruvate was just as downregulated as glucose in the relevant *E. coli* mutants, which is interesting in connection to fatty acid synthesis, as mentioned above. What of Acetyl-CoA?

We appreciate this request for clarification. We provide all metabolites identified in $^1\text{H-NMR}$. Unfortunately, acetyl-CoA was not detected in our NMR analysis. Although acetyl-CoA is a ubiquitous cellular molecule that mediates various anabolic and catabolic reactions, it exists

near sub-micromolar levels, which is out of range for analysis using NMR ¹³. NMR has been restricted in the analysis of biological mixtures to relatively high concentration metabolites (>1 μM). Therefore, NMR spectroscopy has rarely been used to analyze acetyl-CoA. Highly sensitive methods such as absorption spectroscopy and mass spectrometry are used to analyze acetyl-CoA.

Reviewer #2 (Remarks to the Author):

The manuscript by Jeong et al. is a collection of intriguing findings that describe many of the metabolic effects that result downstream of a chronic glucose restricted regimen that was cleverly engineered by the authors. The main observation revealed by this work is that the lifespan of adult *C. elegans* is extended significantly in a manner that is dependent on the reduction of glucose, providing a new model for understanding how glucose affects various processes that can ultimately shorten lifespan, all the while remaining distinct from current diet based lifespan-extending models, namely dietary restriction and some variations thereof.

Not surprisingly, this extension depends on the activity of AMP-activated protein kinase (AMPK), like some of the previously described models of lifespan extension, however what is surprising is that it requires only a specific isoform of one catalytic subunit (AAK-2) that is expressed in the neurons. Using genetic analysis the authors conclude that the AMPK-mediated effect requires neuropeptide secretion, the activity of a transcription factor, changes in the activity of desaturases and even modifications of membrane fluidity.

The manuscript is very well written and presented and the first section where they describe their GR bacterial strain is really beautifully executed to engineer a novel highly useful tool for future studies. Although most of the initial work is very convincing and their genetic analyses are generally sound, some conclusions are based on limited data (membrane fluidity) and others (overall model) are only loosely supported by data shown in the manuscript.

Although the major conclusions of this manuscript are indeed novel and interesting, the manner in which the data are strung together leaves out major components of the model, the characterisation of which would improve the manuscript considerably. Among these, the authors might consider accounting for many of the factors function downstream of AMPK; how the neuropeptides affect NHR-49 signalling; What tissues require NHR-49 function; why/how would membrane fluidity affect lifespan; how do all these processes interact based on genetic analysis; How do the authors implicate PAQR in this pathway-is it regulated in parallel by AMPK or is its expression enhanced by NHR-49. Is is the target of the purported neuropeptide? None of these findings are coherently linked by experiments provided in this manuscript and thus, although independently are interesting, they are strewn together into a linear genetic pathway leaving major logical gaps between the various observations.

Significant Concerns:

First, in all the cases that different isoforms were indeed ineffective in mediating the lifespan extension, can the authors be sure that each isoform was expressed at levels that are comparable (if they are

being compared) in the transgenic animals.

We appreciate this request for clarification. According to the GFP images of *aak-2* isoform transgenic animals, the level of AAK-2c is higher and expressed in more tissues, compared to AAK-2a. Surprisingly, however, AAK-2c does not have a role in GR longevity, raising concerns about whether *aak-2c* transgenic animals indeed express AAK-2c. To address this issue, we compared the level of AAK-2 protein in isoform transgenic animals that were used for lifespan analysis. Consistent with GFP intensities, the protein level of AAK-2c is much higher than that of AAK-2a (Fig. 4h). Of note, AAK-2a (upper band) is 70.17KDa with additional 62 amino acids at its N-terminus, and AAK-2c (lower band) is 63.38KDa. Importantly, we could detect both AAK-2a and AAK-2c isoforms in AAK-2a/c transgenic animals, demonstrating that AAK-2a is an endogenously expressed AAK-2 isoform in *C. elegans*.

Line 369-375;

Figure 4

h

Similarly, in RNAi experiments, can you be sure that RNAi targets that are being compared were indeed reduced significantly and in a comparable level in each case.

We appreciate this request for clarification. To address this concern, we asked if *aak-2* RNAi could reduce the GFP intensity in AAK-2::GFP transgenic animals. As shown in Supplementary Fig. 2q, and 2r, GFP is significantly reduced by *aak-2* RNAi. Furthermore, we directly measured

the level of *aak-2* transcripts in worms fed control diets vs *aak-2* RNAi diets. The qRT-PCR revealed that *aak-2* RNAi significantly reduced endogenous *aak-2* transcripts (Supplementary Fig. 2p).

Line 346-348;

Supplementary Figure 2

Expression in the neurons was SUFFICIENT (maybe not required!), for extension, but is there any role for the excretory cell? A similar implication of the excretory cell was also demonstrated in AMPK-dependent lipid rationing via ATGL-1 and both neurons and the excretory system were more recently implicated in establishing germ cell cycle arrest.

We welcome the reviewer's suggestions for improving the manuscript. As mentioned, it was shown that *aak-2* in excretory cells plays roles in longevity¹⁴ and germ cell cycle¹⁵. In those studies, the *sulp-5* promoter was used to drive *aak-2* expression in the excretory cells. However, *sulp-5* is expressed not only in the excretory cells but also neurons¹⁶. To address the reviewer's comment, we generated *aak-2(ok524);Psulp-5::aak-2a::gfp* animals and found that AAK-2 was indeed in excretory cells and in neurons (Supplementary Fig. 3h). GR diets extended the lifespan of these transgenic animals (Supplementary Fig. 3k). Of note, this is in contrast to the lack of longevity restoration by the *Pmca-1::aak-2a* transgene which is expressed only in excretory cells (Fig. 4n; Supplementary Fig. 3g). To knock down AAK-2a in excretory cells, we took advantage of RNAi feeding, which is refractory in neurons. As expected, GFP RNAi abolished excretory cell expression of AAK-2 and had no effect on neuronal expression (Supplementary Fig. 3l). Importantly, this RNAi did not suppress GR-mediated longevity, demonstrating that AAK-2 functions not in excretory cell but in neurons to mediate GR induced longevity. Together our data demonstrate that AAK-2a functions exclusively in neurons for GR-mediated longevity.

Line 402-423;

Supplementary Figure 3

Figure 4

Supplementary Figure 3

The results with *nhr-49* are somewhat confusing. The model would suggest that neuropeptide secretion is essential downstream of AMPK (*aak-2a*) to extend lifespan, however the authors allude to a potential direct target of AMPK (assuming *nhr-49* acts similarly to *ppar-a*) and conclude that everything must function downstream of this transcription factor. But this factor can work either in the neurons (by the way, the *unc-119p* is not the best for strict neuronal expression-it is a bit leaky!) or in the gut. How do the neuropeptides fit in? Is the expression of *nhr-49* sufficient in every or all tissues? How does the autonomous expression of the transcription factor implicated in the cell non-autonomous effect of GR on organismal lifespan, particularly if it is restricted to a single cell type (gut (probably not due to neuroendocrine release of neuropeptides) or neurons)?

We appreciate this request for clarification. The mechanism by which NHR-46 regulates lifespan is complex. There are multiple publications that have reported that NHR-49 in a single cell type is sufficient for a non-cell autonomous effect on lifespan. Similar to our data, NHR-49 modulates neuronal CRTC-1 mediated longevity non-cell autonomously either in neurons or the intestine¹⁷. NHR-49 is essential for a long lifespan and enhance immunity of germline depleted *glp-1* mutants^{18, 19}. While neuronal NHR-49 is sufficient for pathogen resistance, NHR-49 expression in any somatic tissue rescues longevity¹⁹. These imply that some of downstream effects of NHR-49 are non-cell autonomous. Interestingly, similar to NHR-49, the expression of *paqr-2* in one tissue is sufficient to suppress systemic *paqr-2* mutant phenotypes²⁰. The authors show that cell membrane homeostasis is maintained non-cell autonomously via lipid exchange between distant cells (as shown in Figure 6 from Bodhicharla et al., 2018²⁰). It is possible that NHR-49 in one tissue modulates the systemic effect of GR-induced longevity in a similar manner.

These data were mentioned in the result and the discussion section

Line 498-502; and Line 664-671;

Figure 6 from Bodhicharla et al., 2018²⁰

> by the way, the *unc-119p* is not the best for strict neuronal expression-it is a bit leaky!

We welcome the reviewer's suggestions for improving the manuscript. As reviewer mentioned, the *unc-119* promoter might be a bit leaky. To confirm the neuronal function of NHR-49, we conducted lifespan analysis using other neuronal NHR-49 transgenic animals (*nhr-49(nr2041);Prab-3::nhr-49*) on AL vs. GR diets (Supplementary Fig. 5h). Similarly to the *Punc-119::nhr-49* transgene, *Prab-3::nhr-49* rescued GR induced longevity to *nhr-49* mutants.

Line 493-498;

Supplementary Figure 5

Is there a possibility that there is a parallel pathway that may be functional here that is difficult to assess because the readout (lifespan extension) is not sensitive enough.

We welcome the reviewer's suggestions for improving the manuscript. Both AMPK and NHR-49 control energy homeostasis, lipid metabolism, and longevity^{17, 21}. However, their relative interactions remain elusive. Since either *aak-2* or *nhr-49* mutation shortened normal lifespan on AL diets and completely abolished GR induced longevity (Fig. 4a, and 6a), we asked if the effects on lifespan by either mutation are additive or not. We found that the *aak-2* mutation in *nhr-49(nr2041)* mutants did not further reduce the lifespan of *nhr-49(nr2041)* animals (Fig. 6b; Supplementary Table 6d), suggesting that AAK-2 and NHR-49 might function in the same pathway. For genetic epistasis analysis, we introduced *nhr-49* gain-of-function(*gof*) mutations, *et7* and *et13*, in *aak-2(ok524)* animals. Despite the unresponsiveness of *aak-2* mutants to GR diets, *nhr-49^{gof};aak-2* animals live longer on GR diets compared to AL diets (Fig. 6c, and 6d;

Supplementary Table 6e). Furthermore, the *nhr-49^{gof}(et7 and et13)* transgene confers the GR-induced longevity to *aak-2* mutant animals (Fig. 6e, and 6f; Supplementary Table 6f). This data suggests that *nhr-49* functions at least genetically downstream of *aak-2* for GR-induced longevity. Due to lack of direct evidence, we mentioned the possibility that AAK-2 and NHR-49 function in parallel to promote metabolic remodeling for GR-induced longevity.

Line 473-487;

Figure 4

Figure 6

The description of membrane fluidity is dependent on one single means of quantification that is not at all a direct readout of lipid stacking or fluidity per se. FRAP is an indirect measure of protein exchange in a given environment that can be governed by many different variables, only one of which is membrane fluidity. Although the authors do reference this approach, so far I am not aware of a study that directly compares biophysical quantification of stacking vs FRAP. FRAP may be a proxy for fluidity, but it really needs to be demonstrated. If the authors believe that all the membranes are affected in the animals based on the "organismal" effect of NHR-49 (assuming the fluidity arises due to NHR-49), then using a more biophysical means like C-Laurdan emission of determining lipid stacking in membranes would be more convincing, particularly in combination with the FRAP data. This addition would firstly be robust, and finally provide the appropriate support to link FRAP with bona fide biophysical change in the membrane lipid stacking.

We welcome the reviewer's suggestions for improving the manuscript. As reviewer mentioned, C-Laurdan dye is commonly used to measure membrane fluidity^{22, 23, 24}. It exhibits the spectral shift in emission spectrum according to the levels of membrane order, enabling a straightforward method to monitor the membrane fluidity. We found that the spectrum of C-Laurdan was shifted from solid ordered phases to liquid disordered phases by GR diets, as shown by generalized polarization (GP) index²⁵ (Fig. 7b, and 7c). Taken with FRAP data, C-Laurdan analyses reveal that GR diets promote membrane fluidity in *C. elegans*.

Line 573-578;

Figure 7

The link to PAQR again is intriguing but it is just not clear how this receptor fits into the linear genetic framework that the authors are invoking. In mammalian systems, AdipoR is an upstream activator of AMPK and PPAR α so it is hard to imagine how it would become active in this context or is it working in parallel to NHR-49 and the desaturases that could indeed have a direct effect on lipid composition in the membrane(s). The placement of PAQR upstream of the NHR-49/MDT-15 transcription factors seems speculative at best and should be substantiated.

We welcome the reviewer's suggestions for improving the manuscript. As reviewer mentioned, AdipoR is an upstream activator of AMPK and PPAR α . More specifically, AdipoR1 is tightly linked to the activation of AMPK pathways, whereas AdipoR2 seems to be associated with the activation of PPAR α pathways ²⁶. To investigate whether NHR-49^{gof}/PPAR α could bypass the requirement of PAQR-2/AdipoR2 in GR-induced longevity, we introduced the *nhr-49*^{gof} mutations, *et7* and *et13*, in *paqr-2(tm3410)* animals. Similarly to *nhr-49*^{gof};*aak-2* animals, *nhr-49*^{gof};*paqr-2* animals exhibited lifespan extension by GR diets (Fig. 7g, and 7h; Supplementary Table 6n). This data suggests that NHR-49 may function at least genetically downstream of PAQR-2 for GR-induced longevity. Alternatively, it is also possible that PAQR-2 signaling activates metabolic programs, which requires NHR-49. Of note, *nhr-49* mutations in *paqr-2* mutants severely impaired the general fitness, resulting in the inability to conduct lifespan assays. This implies that NHR-49 and PAQR-2 could play non-redundant roles in general worm health.

Line 603-611;

Figure 7

If the model the authors propose is correct then any membrane FRAP changes observed would not only be dependent on PAQR but also on NHR-49, MDT-15 and the fat-6 desaturase.

We appreciate the reviewer's suggestions for improving the manuscript. As suggested, we generated FRAP transgenic animals in *nhr-49* mutants. Transgenes in *paqr-2* mutants are too unstable to make stable transgenic lines. We did not generate *fat-5;fat-6;fat-7* FRAP transgenic animals because they are lethal. We are sorry that we were unable to conduct all of the experiments the reviewer suggested to do. Similar to *aak-2* mutants, GR-enhanced membrane fluidity was completely abrogated in the *nhr-49(nr2041)* mutants (Figure below).

Furthermore, if this is true, then a misregulated hyperactive form of PAQR should also extend lifespan independently of AMPK, while the same should be true of overexpressed/liganded NHR-49/MDT-15 or even better extra copies of the FAT-6 desaturase. This should all be independent of AMPK and the GR if indeed sole role of GR is to activate AMPK to set this cascade in motion.

We appreciate this request for clarification. As in many situations in biology, homeostasis seems to be critical for players in GR-induced longevity such as AMPK, NHR-49 and PAQR-2. For instance, constitutive active AAK-2 increases *C. elegans* lifespan, yet it induces detrimental pleiotropic side effects including small body size and reduced reproductive capacity ¹⁷. Furthermore, the *nhr-49^{gof}* mutations differentially affect lifespan; *nhr-49(et8)* was short-lived, *nhr-49(et7)* was long-lived, and *nhr-49(et13)* displayed a wild-type life span ²⁷. The high dose injection of *paqr-2* transgene leads to phenotypically sick animals that do not produce stable F2. It might be critical to maintain an appropriate level of membrane fluidity, that is neither too fluidic nor too rigid for the beneficial effects.

Epistasis analysis would help the authors to order their gene activities or place them in linear vs parallel pathways and either confirm or refute the model proposed in Figure 6L.

We appreciate the reviewer's suggestions for improving the manuscript. In response to the previous comments, we conducted genetic epistasis analysis (that is, lifespan on AL vs. GR diets) using *NHR-49^{9of}* in *aak-2* or *paqr-2* mutants. Our data suggest that *NHR-49* functions at least genetically downstream of *AMPK* and *PAQR-2*. And we also modified the text and the model (Fig. 7i) to be better supported by the presented data.

Line 473-487; and 603-611;

Figure 6

Figure 7

Minor Issues:

The manuscript was generally very well written, but nevertheless a number of typos and grammatical issues could be fixed in any new version. I have listed a few of these issues herein below, although this is not comprehensive and further proof-reading is encouraged to improve the quality of the manuscript.

Line 45-calorie intake

Line 59-I don't think that this is an accurate statement. I am not sure that one can conclude that AMPK is actively "downregulated" by glucose. It is simply not activated in these energy rich conditions.

Line 62-there does not exist can be replaced by "a"

Line 63-...animal model does not exist at present, that would enable...

Line 73-eukaryotes, such as yeasts

Line 79-whole body metabolism

Line 112/113-cellular physiology impinging upon

Line 151-metabolite profile

Line 158-7-fold; with lifespan fed on particular E. coli...

Line 167-under the assay...

Line 222-"It" could be re-written as "These levels of glucose"...

Line 247-...introducing a mutation in the *icd* gene

Line 278-suggesting a distinct...

We appreciate all these requests for clarification. As suggested, we revised the manuscript

Line 298/299-are the neurons refractory? Do you always use the enhanced RNAi strain when doing neuronal RNAi? The authors do see RNAi phenotypes using neuronal RNAi so they must be at least partially sensitive. Please reference the tissue specific RNAi strains correctly.

We appreciate this request for clarification. Yes, *C. elegans* neuron is refractory to RNAi due to the lack of the RNA transporter in neurons^{28, 29}. Therefore, to knock down genes of interest in neurons, it is necessary to use the *rrf-3* mutant which is hypersensitive RNAi or TU3401 transgenic animals which express the RNA transporter in neurons³⁰. As suggested, we reference tissue-specific RNAi strains as well as RNAi hyper-sensitive strains.

Line 348-353;

Line 315/316-implies...previous AMPK studies utilized the ...isoform

Line 332-I am not sure that any experiment unequivocally removed aak-2a specifically in the neurons where a true requirement aak-2a can be concluded. Neuronal aak-2 is required based on the data with TU3401, but adding back aak-2a by transgenics addresses sufficiency. In fact, aak-2c is also sufficient if driven in the right cells.

We appreciate the reviewer's suggestions for improving the manuscript. As suggested, we removed 'necessary' from 'necessary and sufficient' in the revised manuscript.

Line 389-391;

We found that their lifespans were comparable on either AL or GR diets (Supplementary Fig. 3b; Supplementary Table 4k), indicating that the AAK-2a isoform was sufficient for endogenous AMPK-mediated lifespan regulation.

Supplementary Figure 3

Line 361-"endocrinal" should be endocrine

We appreciate the reviewer's suggestions for improving the manuscript. As suggested, we revised the manuscript.

Line 405-nhr-49 has already been shown to be required in neurons to extend lifespan by altering the lipidome.

We appreciate this request for clarification. As reviewer mentioned, we reference the role of NHR-49 in lifespan and lipid metabolism.

Line 473-474; 498-499;

Both AMPK and NHR-49 control energy homeostasis, lipid metabolism and longevity ^{17, 21}

Interestingly, there are multiple publications that NHR-49 in a single cell type is sufficient for a non-cell autonomous effect on lifespan ^{17, 18, 19}.

Line 410-*unc-119p* is leaky. It is no longer the go to pan-neural promoter for neuron-specific expression

We welcome the reviewer's suggestions for improving the manuscript. In addition to the *unc-119* promoter, we expressed NHR-49 using another pan-neuronal promoter, *Prab-3* in *nhr-49* background. As expected, the *Prab-3::nhr-49* transgene restored GR-induced longevity (Supplementary Fig. 5h).

Line 493-498;

Supplementary Figure 5

Line 421-Nile red is not conventionally used for lipid quantification-it is less specific than Oil Red O or CARS

We appreciate this request for clarification. As reviewer suggests, we conducted Oil Red O staining using control, *aak-2* and *aak-2;aak-2a* transgenic animals on AL vs. GR diets (Supplementary Fig. 5f). Biochemical analysis of TAG is also provided in Supplementary Fig. 5g, showing the same trends in fat contents.

Line 509-518;

Supplementary Figure 5

f

Line 486-whether the AAK-2a

Line 511-to mimic GR diets, the inhibitor of glycolysis 2-DG, or knocking down/blocking glycolytic enzymes have been utilized...

We appreciate these requests for clarification. As reviewer suggested, we revised the manuscript.

Line 527-prolongs longevity.

Some of the findings corroborate work performed by other investigators, particularly with reference to neuronal NHR-49 and its role in lifespan extension, although the authors appear to either be unaware of this or simply overlooked referencing it.

We appreciate this request for clarification. The mechanism by which NHR-49 regulates lifespan is complex. There have been multiple publications that NHR-49 in a single cell type is sufficient for a non-cell autonomous effect on lifespan. Similar to our data, NHR-49 modulates neuronal CRT-1 mediated longevity non-cell autonomously either in neuron or intestine¹⁷. NHR-49 is essential for a long lifespan and enhances immunity of germline depleted *glp-1* mutants^{18, 19}. While neuronal NHR-49 is sufficient for pathogen resistance, NHR-49 expression in any somatic tissue rescued longevity¹⁹. We provide the indicated references in the revised manuscript.

Line 498-502;

Reviewer #3 (Remarks to the Author):

The work presented in this article presents significant insights into the mechanisms of *C.elegans* longevity associated with glucose restricted diet. By feeding worms with *E.coli* mutants that mimics glucose restriction and therefore increase *C.elegans* lifespan, the authors unravel the specific role in this process of a new AMPK isoform, AAK-2a, in neurons. Involvement of neuropeptide signaling is also shown together with modulation of GR-induced longevity by NHR-49, PAQR-2 and $\Delta 9$ desaturases that promote membrane fluidity. While a large part of the discussion is well supported by the data included in the manuscript, a number of meaningful revisions should be considered before this manuscript is suitable for publication in Nature Communications.

1) The syntax of the title should be revised to provide a clearly articulated message. The use of symbol “/” is too ambiguous and does not serve well this purpose.

We welcome the reviewer’s suggestions for improving the manuscript. As suggested, we revised the title as follows.

“A new AMPK isoform mediates glucose-restriction induced longevity non-cell autonomously by promoting membrane fluidity”

2) Overall, the description of methods is far too succinct. When referring to “standard methods” (such as in paragraph “*C. elegans*”), please at least include references. Some more detailed methods could also be provided in the supplementary material. This concerns most of the methods sections.

We welcome the reviewer’s suggestions for improving the manuscript. As suggested, we completely revised the methods to provide detailed information.

3) l.171-172: is the actual uptake of α -KG by the worms evaluated? (consumption in the medium or accumulation in the worms)

We welcome the reviewer’s suggestions for improving the manuscript. As suggested, we asked if *C. elegans* could uptake α KG from media. When supplemented in various concentrations (0, 2, 4, 10mM), α KG levels in *C. elegans* increased in a dose-dependent manner (Supplementary Fig. 1d).

Line 173-176;

Supplementary Figure 1

4) I.202-203: At this point in the text, the equivalence to glucose restriction is not demonstrated. This conclusion is premature, and better supported by data presented later in Fig. 2f. The data of Fig 1 only shows that high-glucose diet suppressed the lifespan extension provided by *DsucA*.

We appreciate this request for clarification. As suggested, we carefully revised the initial part of the manuscript and referred to $\Delta sucA$ *E. coli* as GR diets Fig. 2h (previous Fig. 2f) hereafter.

5) I.248-249: This result is actually not remarkable, but looks rather trivial. In the absence of *icd*, there is no TCA production of α -KG. Without primary substrate for *sucA*, no additional effect should be expected when mutating both *icd* and *sucA* as compared to mutating *icd* alone.

We appreciate this request for clarification. As suggested, we revised the manuscript as follows.

Line 291-299;

If that is the case, the inhibition of TCA cycle flux in $\Delta sucA$ *E. coli* by adding Δicd mutation can facilitate the glyoxylate cycle (Fig. 2h) and, in turn, abolish the lifespan extension by $\Delta sucA$ mutation. As hypothesized, the metabolite profile of the $\Delta icd\Delta sucA$ double mutants became similar to those of the control and Δicd mutants (Fig. 1b; Supplementary Table 2), recovering the levels of TCA intermediates and glucose (Fig. 1c).

Figure 2

h

E. coli genotype	WT	Δ sucA	Δ icd	Δ sacEBA	Δ icd Δ sacEBA	Δ sacNA Δ sacNB	Δ icd Δ sucA
Estimated Metabolic Flux							
TCA cycle/ glyoxylate cycle	O/O	X/-	X/O	O/X	X/X	X/X	X/O
gluconeogenesis substrates	-	↓	-	-	↓	↓	-
glucose	-	↓	-	-	↓	↓	-
lifespan	normal	long	normal	normal	long	long	normal

Figure 1

b

c

6) The actual conclusion of the first part of the paper seems well exposed in lines 255-256: mutants impaired in both TCA and glycolate cycle can possibly serve as low glucose diet and extend worms lifespan. The authors show that there can be 3 ways to do this, by either mutating a) both *acnA* & *acnB*, or b) *aceBA* and *icd*, or c) perhaps more surprisingly, *sucA* alone by draining metabolic fluxes from the 2 pathways into α -KG accumulation. The whole narrative of the first part of the paper is constructed around this later case. While the use of single *DsucA* mutation may stand at the simplest way to mimick GR, the entire focus on *DsucA* in this part is not a straightforward way to deliver the more global conclusion. The metabolites analysis presented in Fig1 supports this narrative, but does not include data for *DacnA-DacnB* or *Dicd -DaceBA* mutants. Such data would provide interesting clues regarding common metabolic features of these long-lived mutants and better support the "estimated metabolic flux" schemes presented in Fig2f. Rather than "estimated", these schemes are somewhat hypotetic as concerns *DacnA-DacnB* and *Dicd -DaceBA* mutants, with no information about metabolites that accumulate in this case.

We welcome the reviewer's suggestions for improving the manuscript. As suggested, we analyzed the intracellular metabolites of $\Delta aceBA$, $\Delta acnA\Delta acnB$ and $\Delta icd\Delta aceBA$ as well as control. As presented in Fig. 2f, and 2g; Supplementary Table 3, all of *E. coli* mutants that extend the lifespan of *C. elegans* exhibit the similar metabolite profile to $\Delta sucA$ *E. coli*, with low level of glucose and TCA intermediates. Importantly, the metabolite profile of glyoxylate cycle deficient $\Delta aceBA$ mutants resembles that of control.

Line 266-273;

Figure 2

7) Metabolites analyses:

-l.983: after quenching, the supernatant was discarded. Was this solution checked for metabolite content? Ethanol is not a fixator, but permeabilizes cell membranes. The discarded supernatant is likely to contain significant amounts of extracted metabolites.

We appreciate this comment for clarification. We did not check the metabolites in the supernatant. Due to their rapid turnover, metabolite levels can change significantly during sampling. To prevent this, the cellular metabolism should be stopped, referred to as quenching. Quenching with 60% methanol at -20°C has been widely used for bacteria. This approach was originally developed for yeast, but has been used in studies of various bacteria. However, when this procedure is applied to prokaryotic microbes, drastic loss (60 ~ 80%) of all metabolites was observed in some cases, due to unspecific leakage ³¹. To prevent damage of the cell membrane and thereby prevent metabolite leakage, 40% ethanol, 0.8% NaCl solution was used instead of 60% methanol. This quenching solution leads to increased metabolite levels up to 80% recovery and decrease standard error, compared with methanol quenching ³². Therefore, we followed the protocol developed by Spura et al. ³².

-Precise extraction protocol (or at least reference) should be specified: name of solvents without indication of proportions and total volume is little informative. How were these extracts then dried, and what was the NMR buffer used?

We welcome the reviewer's suggestions for improving the manuscript. As suggested, we have modified the Method section and included a detailed extraction procedure, as shown below.

"Polar metabolites were extracted from cells using three freeze-thaw cycles in the presence of 2 mL of cold 80% aqueous methanol. After centrifugation, the supernatant was transferred to a new EP tube and 500 µL of chloroform and 1 mL of distilled water were added to the remaining pellet. Next, the solution was vortexed and centrifuged, and then, the supernatant was combined with the supernatant obtained from the previous step. The aqueous supernatant containing the water-soluble metabolites was vacuum dried and resuspended in 600 µL of 0.1 M sodium phosphate-buffered deuterium oxide (pH 7.0) containing 0.1 mM 3-(trimethylsilyl) propionic-2,2,3,3-d₄ acid (TSP-d₄) (Sigma Aldrich)."

-what was the quantity of material (number of bacterial cells) used?

We appreciate this comment for clarification. The number of bacterial cells for metabolite analysis is as follows

related to Fig. 1b, and 1c; Supplementary Table 2

	Used for	E. coli	Cell number
SET 1	E. coli metabolomics	ctrl	0.295 x 10 ¹¹
		Δ sucA	0.606 x 10 ¹¹
		Δ icd	0.494 x 10 ¹¹
		Δ icd Δ sucA	0.602 x 10 ¹¹
SET 2	E. coli metabolomics	ctrl	0.295 x 10 ¹¹
		Δ sucA	0.468 x 10 ¹¹
		Δ icd	0.408 x 10 ¹¹
		Δ icd Δ sucA	0.716 x 10 ¹¹
SET 3	E. coli metabolomics	ctrl	0.408 x 10 ¹¹
		Δ sucA	0.501 x 10 ¹¹
		Δ icd	0.504 x 10 ¹¹
		Δ icd Δ sucA	0.632 x 10 ¹¹

related to Fig. 2f, and 2g; Supplementary Table 3

	Used for	E. coli	Cell number
SET 1	E. coli metabolomics	ctrl	0.715 x 10 ¹¹
		Δ aceBA	0.914 x 10 ¹¹
		Δ acnAB	0.401 x 10 ¹¹
		Δ icd Δ aceBA	0.811 x 10 ¹¹
SET 2	E. coli metabolomics	ctrl	0.946 x 10 ¹¹
		Δ aceBA	0.119 x 10 ¹²
		Δ acnAB	0.693 x 10 ¹¹
		Δ icd Δ aceBA	0.863 x 10 ¹¹
SET 3	E. coli metabolomics	ctrl	0.141 x 10 ¹²
		Δ aceBA	0.132 x 10 ¹²
		Δ acnAB	0.597 x 10 ¹¹
		Δ icd Δ aceBA	0.986 x 10 ¹¹

-The detection of tartrate is somewhat surprising in this type of sample. Please specify how annotation was confirmed, and if it relies on 1H or both 1H/13C chemical shifts. Considering the low concentration detected, the corresponding signal must have very low S/N. As this measurement does not bring anything to the discussion, it is safer not to bring it forward if any ambiguity remains in the annotation.

We appreciate this comment for clarification. We have annotated the NMR peak of tartrate using both 1H and 13C chemical shifts in the 2D NMR HSQC experiment. However, the NMR peak of tartrate was very low and it may affect the quantitation accuracy. Therefore, as per the reviewer's comment, we excluded tartrate in our results (Supplementary Table 2) and re-analyzed data for

PCA analysis (Fig. 1b).

-PCA analysis of Fig 1b: how many variables were used? was is carried out on the full NMR profiles, or on the set of quantified metabolites?

We appreciate this comment for clarification. The PCA analysis of Fig. 1b and Fig. 2f was carried out on the set of quantified metabolites, which are presented in Supplementary Tables 2 and 3, respectively. 50 metabolites were used for Fig. 1b and 19 metabolites were used for Fig. 2f.

8)The data of Supp Fig 2b-o shows that effect of some of the lifespan modulators can be either cumulative or counteractive with GR-induced longevity, while in a number of other cases the selected “DR effectors” does not seem to have any impact on worms longevity (regardless of GR condition), such as in the case of *eat-2*, *rsks-1*, *sir-2.1*, *skn-1*, *pdr-1* and *pink-1*. These observations could be further described and commented in the text.

We welcome the reviewer’s suggestions for improving the manuscript. It is hard to discrete DR effectors from other longevity pathways, due to their overlapping functions. We wanted to test as many mutants as possible. In the manuscript, we selected lifespan regulating genes such as DR effectors (genetic DR, FOXA, FOXO, TOR, HIF, AMPK, Sirtuin, HSF, SEK-1, and NRF2) and other longevity pathways (PDR-1, PINK-1, ISP-1). As shown in Supplementary Fig. 2 (submitted), GR extended the lifespan of most mutants. As for the mutants the reviewer mentioned, *eat-2* and *rsks-1* mutants were not long-lived on our AL diets, which is contradictory to previous reports. Therefore, as per the reviewer’s comment, we excluded lifespan data of *eat-2* and *rsks-1* mutants. There exist multiple publications showing that *sir-2.1* and *skn-1* mutants have either shorted or similar lifespans compared to control. Therefore, we did not make changes to *sir-2.1* and *skn-1* data. In *pdr-1* and *pink-1* mutants, there are multiple reports showing that their lifespans are similar to control. Therefore, we did not make changes to *pdr-1* and *pink-1* data. Of note, regardless of their lifespans on AL diets, lifespans are further extended by GR diets.

These are summarized in the table below.

Lifespan on AL diets

	Reported as	In this study	Revised as
eat-2(ad1116)	long-lived	similar to control	deleted
rsks-1(ok1255)	long-lived	similar to control	deleted
sir-2.1(ok434)	short-lived ^{33, 34} similar to control ³⁵	similar to control	not changed

skn-1(zu135)	short-lived ^{36, 37} similar to control ^{38, 39, 40}	similar to control	not changed
pdr-1(gk448)	similar to control ^{41, 42, 43}	similar to control	not changed
pink-1(ok3538)		similar to control	not changed

9) In the data of Fig 4i, control N2 animals live considerably longer than in most of the other experiments presented in the paper (including the ones of fig 4h and 4j).

We appreciate this comment for clarification. We replaced the lifespan graph (Fig. 4j) from 3 biological repeats. The raw data and statistics of all 3 lifespan repeats are presented in Supplementary Table 4h.

Figure 4

Consequently, the sentence l.325-327 stating that aak2;aak2a animals lived longer compared to aak-2;aak2c on AL diets are not well supported by reliable controls, and a direct comparison would better support this observation.

We appreciate this comment for clarification. A direct comparison between AAK-2 isoforms was presented in Supplementary Fig. 3a and Supplementary Table 4j. These data directly support the critical role of AAK-2a in longevity control, compared with AAK-2c.

Line 383-386;

Supplementary Figure 3

10) Data in figure 6g show that *fat-6* RNAi on the double mutant *fat-5;fat-7* displays a clearly reduced lifespan as compared to the double mutation alone. The data also show that this effect is well attenuated under GR. This means GR does impact the worms longevity, even in the absence of fatty acid desaturation. This somewhat contradicts the interpretation given by the authors l. 457-459, suggesting that fatty acid desaturation was required for GR-mediated longevity.

We appreciate this comment for clarification. As reviewer mentioned, *fat-6* RNAi did not fully suppress the lifespan extension of *fat-5;fat-7* mutants. It reduced GR induced longevity by ~60%. Considering incomplete RNAi knocking-down, we concluded that fatty acid desaturation was required for GR-mediated longevity. Based on the reviewer's comment, we revised the manuscript as follows.

Line 546-549;

However, we found that *fat-6* RNAi reduced the GR-mediated longevity in *fat-5(tm420);fat-7(wa36)* animals by 61.03 % (Fig. 6g; Supplementary Table 6l), demonstrating that fatty acid desaturation was implicated in GR-mediated longevity.

Figure 6

11) In figure 6k, the lifespan assay data for *paqr-2* GR worms is not at all visible.

We appreciate this comment for clarification. We revised the lifespan graph for better visualization.

Figure 7

Reviewer #4 (Remarks to the Author)::

In this manuscript, the authors investigate the effects of sucA deficient E. coli deficient on C. elegans longevity. They show that C. elegans grown on sucA mutants are long live relative to those grown on control E. coli. Based on metabolomic measurements of sucA mutants, examinations of several other mutant combinations, and metabolite add-back experiments, they conclude that that this longevity is caused to reduced levels of glucose in sucA mutants. In turn, they find that the noted C. elegans longevity requires the presence of AA2-a, an isoform of the catalytic subunit of AMPK, in the nervous system. Based on genetic inactivations and reconstitution experiments, they posit that neural AAK-2 regulates the release of an unidentified neuropeptide from the nervous system. Their model implies that this neuropeptide acts on PAQR-2 receptor in peripheral tissues to, in turn, signal to the MDT-15/NHR-49 transcriptional complex. The consequence of this is increased transcription of genes that encode for fat desaturases, which confer longevity to animals by ultimately promoting membrane fluidity.

The appealing aspects of the manuscript include i) use of a bacterial mutant as a way of modulating the C. elegans diet, ii) use of 1H-NMR to characterize the metabolite profiles in the bacterial mutants iii) examination of multiple mutants in various longevity pathways, iv) a relatively straightforward and easy to follow writing and data presentation style.

There are, however, a number of serious concerns about the conclusions presented that prevent support for the current publication. These concerns are listed below.

Major concerns:

1) The model shown in figure 6l is a reasonable hypothesis based on mostly genetic data. The genetic data definitely indicate that each of the components (e.g. neural AAK-2a, peripheral NHR-49/MDT-15, PAQR-2) are required for the lifespan increasing effects of sucA mutants. While it is reasonable to speculate that these components form the proposed signaling cascade, there is virtually no data that indicates that these genes indeed form such a signaling cascade! Thus, either the writing has to be dramatically changed or additional data provided to support the notion that the genetic requirements define a unified, signaling pathway.

We welcome the reviewer's suggestions for improving the manuscript. To investigate the relative interactions of AMPK, PAQR-2 and NHR-49 in GR-mediated longevity, we performed a genetic epistasis analysis. Although AMPK and NHR-49 control energy homeostasis, lipid metabolism and longevity^{17, 21}, their relative interactions remain elusive. Since either *aak-2* or *nhr-49* mutation shortened normal lifespans on AL diets and completely abolished GR-induced longevity (Fig. 4a, and 6a), firstly we asked if their effects on lifespan were additive or not. An *aak-2* mutation in the *nhr-49(nr2041)* mutants did not further reduce the lifespan of *nhr-49(nr2041)* animals (Fig. 6b; Supplementary Table 6d), suggesting that AAK-2 and NHR-49 might function in the same pathway. For a genetic epistasis analysis, we introduced the *nhr-49 gain-of-function(gof)* mutations, *et7* and *et13*, in *aak-2(ok524)* animals. Despite the unresponsiveness of *aak-2* mutants to GR diets, *nhr-49^{gof};aak-2* animals live longer on GR diets, compared to AL diets (Fig. 6c, and 6d; Supplementary Table 6e). Further, the *nhr-49^{gof}(et7 and et13)* transgene confer the GR induced longevity to *aak-2* mutant animals as well (Fig. 6e, and 6f; Supplementary Table 6f). These data suggest that NHR-49 functions at least genetically downstream of AAK-2 for GR induced longevity.

Line 473-487;

Secondly, AdipoR2 is reported to be an upstream activator of PPAR α ²⁶. To investigate whether NHR-49^{gof}/PPAR α could bypass the requirement of PAQR-2/AdipoR2 in GR induced longevity, we introduced *nhr-49^{gof}(et7 and et13)* mutations in *paqr-2(tm3410)* animals. Similarly to *nhr-49^{gof};aak-2* animals, *nhr-49^{gof};paqr-2* animals exhibited lifespan extension by GR diets (Fig. 7g, and 7h; Supplementary Table 6n). Of note, the *nhr-49* mutations in *paqr-2* mutants lead to severe sickness of animals, resulting in an inability to conduct lifespan assays.

Line 603-611;

Figure 4

Figure 6

Figure 7

Taken altogether, genetic epistasis analysis suggests that NHR-49 may function at least genetically downstream of AMPK and PAQR-2 for GR-induced longevity. Surprisingly we could not generate *paqr-2;aak-2* double mutants likely due to their synthetic lethality. This implies that AAK-2 and PAQR-2 play non-overlapping roles in general worm health and may function in parallel for GR induced longevity. We modified the text and the model (Fig. 7i) to be better supported by the presented data.

Figure 6

Figure 7

2) As one example of why it matters to distinguish between genetic requirement and a bona fide signaling cascade, considering the role of aak-2 is illustrative: loss of neural aak-2 has previously been reported to mimic the effects of enhanced serotonin signaling. For certain outcomes, loss of aak-2 promotes neuropeptide release. The authors reported that loss of each of AAK-2 and UNC-31, a component of dense core vesicles through which neuropeptides are released, is required for the noted sucA mediated longevity. Thus, it is equally, if not more, likely that what they are observing is the effects of elevated serotonin signaling simply countering the effects of sucA deficiency. As such, without additional evidence, it is hard to distinguish whether neural AAK-2a senses and signals glucose deficiency or that increased serotonin signaling functions in a parallel pathway to reduce lifespan. Similar examples can be given for the other components (e.g. NHR-49, desaturases, PAQR).

We appreciate this comment for clarification. It is my understanding that the reviewer raised the possibility that GR induced longevity is not dependent on AAK-2 but dependent on the reduced level of pro-aging neuropeptide. GR diets reduce the secretion of pro-aging neuropeptide, leading to a long lifespan. In *aak-2* mutants, secretion of pro-aging neuropeptide is enhanced, countering the effects of GR diets, thereby leading to a short lifespan.

However, we strongly argue that for the following reasons.

1. To determine whether GR diets activate AAK-2, we examined the phosphorylation status of AAK-2. We found that GR diets enhanced the phosphorylation of Thr²⁴³ of AAK-2 (equivalent to Thr¹⁷² of the human AMPK α subunit) (Figure below). Therefore, GR diets indeed activate AAK-2 in *C. elegans*. Currently we are working on upstream regulatory signaling that activate AAK-2 in response to glucose restriction and preparing the manuscript.

2. If pro-aging neuropeptide is key for GR-induced longevity, mutation of neuropeptide processing enzymes such as UNC-31 and EGL-21 should extend the lifespan. On the contrary, *unc-31* and *egl-21* mutations completely abolished lifespan extension by GR diets (Fig. 5b, and 5c). These data demonstrate that pro-aging neuropeptide is not implicated in GR induced longevity but pro-longevity neuropeptide and its processing are required for GR-induced longevity.

Figure 5

3. GR diets extend the lifespan of *tph-1* mutants that are deficient in serotonin biosynthesis (Supplementary Fig. 4d). This data suggests that GR-induced longevity is not related to serotonin signaling.

Supplementary Figure 4

4. Similar to reviewer's comment, CA-AAK-2 inhibited CRTC-1 in the neurons, thereby inhibiting pro-aging signaling to prolong longevity¹⁷. Therefore, CRTC-1 (S76A, S179A) mutants, in which CRTC-1 is constitutively active, are refractory to the lifespan extension by CA-AAK-2⁴⁴. We found that GR diets extended the lifespan of CRTC-1(S76A, S179A) animals (Fig. 5e; Supplementary Table 5e), demonstrating that GR-mediated pro-longevity signaling is dominant over the CRTC-1-mediated pro-aging signaling.

Figure 5

Taken altogether, we believe that GR activates AAK-2a, activates pro-longevity signaling and thereby extends the lifespan.

3) Concerns about novelty and advance: each of the components of the signaling cascade have already been shown to be important for mediating the lifespan extensions caused by either glucose deprivation or DR. Thus, it is unclear precisely what constitutes the advance of the manuscript. As noted above, there is very little data supporting the notion that the components form a signaling cascade.

We appreciate this comment for clarification. As reviewer mentioned, some, not all, of the factors in our study have been shown to function in lifespan extension by DR. The novelty and the advances in our study are presented as follows.

1. There has been no bona fide glucose restricted regimen yet. To mimic GR diets, the inhibitor of glycolysis, 2-DG, or knocking down/blocking of glycolytic enzymes has been utilized ⁴⁵. We provide multiple *E. coli* mutants depleted of intracellular glucose as bona fide GR diets. Furthermore, we identify the underlying mechanisms resulting in their lower glucose level. Our GR regimen exhibits advantages over general DR such as resistance to stress and proteopathy without loss of fertility. Given the unsustainability and possible side-effects of general DR application for humans, this work deepens our understanding of the health-benefit by specific nutrient-restricted diets such as GR and provides novel molecular targets for healthspan.
2. AMPK is a highly conserved protein kinase regulating energy homeostasis. Interestingly, in *C. elegans*, AMPK is required for only a subset of DR regimens ⁶ and the exact mechanisms remain to be elucidated, including tissue specific functions and the distinct roles by AMPK isoforms. For the first time, we identified a new AAK-2 isoform, AAK-2a, that functions exclusively in neurons to extend lifespan non-cell autonomously via pro-longevity neuropeptide signal.
3. Using fluorescence recovery after photobleaching (FRAP) and C-Laurdan dye analysis, we demonstrate that GR diets promote membrane fluidity in intestinal cells in a neuronal AAK-2a dependent manner. Further, PAQR-2/AdipoR2 is required for GR-mediated longevity, providing potential targets to promote the DR-induced fitness in humans.

We believe that our findings will be of interest to readers of *Nature Communications*, particularly readers interested in how dietary restriction promotes longevity and prevents age-associated diseases.

4) The notion that *sucA* mutants confer longevity due to reduced glucose levels is a very reasonable conclusion but not definitive. Definitive results require inhibiting gluconeogenesis at a step, such as PEPCK, that would primarily effect glucose production and not necessarily numerous other metabolites.

We welcome the reviewer's suggestions for improving the manuscript. In *E. coli*, the gluconeogenesis pathway is essentially a reversal of glycolysis, sharing most of the glycolysis enzymes. During gluconeogenesis two unique enzyme activities catalyze irreversible reactions; class I fructose 1,6-bisphosphatase (*fbp*), and PEP synthetase (*ppsA*). Multiple isoenzymes have been identified as follows; *E. coli* has two class II fructose-1,6-bisphosphatases, *GlpX* and *YggF*, which have a lower catalytic efficiency compared with class I *Fbp* ⁴⁶. *YaeD* and *YbhA* are also

known to catalyze FBP hydrolysis⁴⁷. The formation of PEP is catalyzed by PpsA along with reversible PEP carboxykinase (pck)⁴⁸. The presence of multiple isoenzymes make it challenging to explore their effects on intracellular glucose levels in *E. coli* (Figure below). In contrast to the poor growth of glucose-depleted *E. coli* mutants such as $\Delta sucA$, $\Delta acnA\Delta acnB$, $\Delta icd\Delta aceBA$ on NGM media, we found that $\Delta ppsA$ and Δpck *E. coli* single mutants grow well on NGM plates. This implies that either single mutant can produce glucose using the remaining enzyme activity to produce PEP. Consequently, neither single mutants extend the lifespan (Figure below). Therefore, we tried to generate $\Delta ppsA\Delta pck$ *E. coli* double mutants several times, with no success. We are sorry that we were unable to address the issue that reviewer raised.

Minor Concerns:

1) It is very interesting that the authors cannot attribute the noted longevity to accumulation of alpha-ketoglutarate. They should expand the discussion of this discrepancy. Are there obvious differences in the experimental set up that can account for this?

We appreciate this comment for clarification. As mentioned in the manuscript, we conducted lifespan under various experimental conditions such as 2, 4, 6, 8, or 12 mM treatment from L1 larvae or adults; treatment every other day; and treatment on standard *E. coli* B strain OP50 or K12 strains, 20°C or 25°C. In every condition, α KG did not extend the lifespan (Supplementary Fig. 1b). To exclude the possibility that *E. coli* consumed supplemented α KG or *C. elegans* did not uptake α KG from media, we used $\Delta kgtP$ mutants (that are unable to uptake α KG) for lifespan analysis (Supplementary Fig. 1c) or measured α KG uptake by *C. elegans* (Supplementary Fig.

1d). With all these, we concluded that the α KG accumulated in Δ sucA *E. coli* did not cause the lifespan extension in *C. elegans*. Currently, we do not have an explanation for the difference between the previous report and our studies. Recently Edwards et al., reported that α KG extends the lifespan in *C. elegans*¹¹. However, the extent of lifespan extension seems to be much lesser, compared with previous reports⁴⁹. It is possible that the effect of α KG on *C. elegans* lifespan may not be significant. For the reviewer's convenience, two lifespan graphs from two different publications are shown below.

Supplementary Figure 1

2) It is good that the authors show that growth on sucA deficient bacteria does not affect *C. elegans* fecundity. Nevertheless, it is possible that the noted effects are still related to the hermaphroditic nature of *C. elegans*. To claim that the longevity effects are broadly applicable, they should examine lifespan of males on these bacteria. This will be good to show but not required for the publication. If the experiments are not provided, the writing should, however, be modified to acknowledge this.

We welcome the reviewer's suggestions for improving the manuscript. As suggested, we performed lifespan assays using male *C. elegans* on GR diets vs AL diets. Similarly to *C. elegans* hermaphrodite, $\Delta sucA$ *E. coli* extended the lifespan of male *C. elegans* (Fig. 1h; Supplementary Table 1m).

Line 223-226;

Figure 1

h

References

1. Lee SJ, Murphy CT, Kenyon C. Glucose shortens the life span of *C. elegans* by downregulating DAF-16/FOXO activity and aquaporin gene expression. *Cell Metab* **10**, 379-391 (2009).
2. Yamauchi T, *et al.* Cloning of adiponectin receptors that mediate antidiabetic metabolic effects. *Nature* **423**, 762-769 (2003).
3. Kalemka KM, *et al.* Glycerol induces G6pc in primary mouse hepatocytes and is the preferred substrate for gluconeogenesis both in vitro and in vivo. *J Biol Chem* **294**, 18017-18028 (2019).
4. Svensk E, *et al.* *Caenorhabditis elegans* PAQR-2 and IGLR-2 Protect against Glucose Toxicity by Modulating Membrane Lipid Composition. *PLoS Genet* **12**, e1005982 (2016).
5. Devkota R, Svensk E, Ruiz M, Stahlman M, Boren J, Pilon M. The adiponectin receptor AdipoR2 and its *Caenorhabditis elegans* homolog PAQR-2 prevent membrane rigidification by exogenous saturated fatty acids. *PLoS Genet* **13**, e1007004 (2017).
6. Greer EL, Brunet A. Different dietary restriction regimens extend lifespan by both independent and overlapping genetic pathways in *C. elegans*. *Aging Cell* **8**, 113-127 (2009).
7. Hagiwara K, *et al.* NRFL-1, the *C. elegans* NHERF orthologue, interacts with amino acid transporter 6 (AAT-6) for age-dependent maintenance of AAT-6 on the membrane. *PLoS One* **7**, e43050 (2012).
8. Devkota R, Henricsson M, Boren J, Pilon M. The *C. elegans* PAQR-2 and IGLR-2 membrane homeostasis proteins are uniquely essential for tolerating dietary saturated fats. *Biochim Biophys Acta Mol Cell Biol Lipids* **1866**, 158883 (2021).
9. Alcedo J, Kenyon C. Regulation of *C. elegans* longevity by specific gustatory and olfactory neurons. *Neuron* **41**, 45-55 (2004).
10. Apfeld J, Kenyon C. Regulation of lifespan by sensory perception in *Caenorhabditis elegans*.

Nature **402**, 804-809 (1999).

11. Edwards C, *et al.* Mechanisms of amino acid-mediated lifespan extension in *Caenorhabditis elegans*. *BMC Genet* **16**, 8 (2015).
12. Yazawa H, *et al.* Beta amyloid peptide (A β 42) is internalized via the G-protein-coupled receptor FPRL1 and forms fibrillar aggregates in macrophages. *FASEB J* **15**, 2454-2462 (2001).
13. Nagana Gowda GA, Abell L, Tian R. Extending the Scope of (1)H NMR Spectroscopy for the Analysis of Cellular Coenzyme A and Acetyl Coenzyme A. *Anal Chem* **91**, 2464-2471 (2019).
14. Narbonne P, Roy R. *Caenorhabditis elegans* dauers need LKB1/AMPK to ration lipid reserves and ensure long-term survival. *Nature* **457**, 210-214 (2009).
15. Kadekar P, Roy R. AMPK regulates germline stem cell quiescence and integrity through an endogenous small RNA pathway. *PLoS Biol* **17**, e3000309 (2019).
16. Spencer WC, *et al.* A spatial and temporal map of *C. elegans* gene expression. *Genome Res* **21**, 325-341 (2011).
17. Burkewitz K, *et al.* Neuronal CRTIC-1 governs systemic mitochondrial metabolism and lifespan via a catecholamine signal. *Cell* **160**, 842-855 (2015).
18. Ratnappan R, *et al.* Germline signals deploy NHR-49 to modulate fatty-acid beta-oxidation and desaturation in somatic tissues of *C. elegans*. *PLoS Genet* **10**, e1004829 (2014).
19. Naim N, Amrit FRG, Ratnappan R, DelBuono N, Loose JA, Ghazi A. Cell nonautonomous roles of NHR-49 in promoting longevity and innate immunity. *Aging Cell* **20**, e13413 (2021).
20. Bodhicharla R, Devkota R, Ruiz M, Pilon M. Membrane Fluidity Is Regulated Cell Nonautonomously by *Caenorhabditis elegans* PAQR-2 and Its Mammalian Homolog AdipoR2. *Genetics* **210**, 189-201 (2018).
21. Moreno-Arriola E, El Hafidi M, Ortega-Cuellar D, Carvajal K. AMP-Activated Protein Kinase

- Regulates Oxidative Metabolism in *Caenorhabditis elegans* through the NHR-49 and MDT-15 Transcriptional Regulators. *PLoS One* **11**, e0148089 (2016).
22. Owen DM, Rentero C, Magenau A, Abu-Siniyeh A, Gaus K. Quantitative imaging of membrane lipid order in cells and organisms. *Nat Protoc* **7**, 24-35 (2011).
 23. Ruiz M, Stahlman M, Boren J, Pilon M. AdipoR1 and AdipoR2 maintain membrane fluidity in most human cell types and independently of adiponectin. *J Lipid Res* **60**, 995-1004 (2019).
 24. Kim HM, *et al.* A two-photon fluorescent probe for lipid raft imaging: C-laurdan. *Chembiochem* **8**, 553-559 (2007).
 25. Amaro M, Reina F, Hof M, Eggeling C, Sezgin E. Laurdan and Di-4-ANEPPDHQ probe different properties of the membrane. *J Phys D Appl Phys* **50**, 134004 (2017).
 26. Yamauchi T, *et al.* Targeted disruption of AdipoR1 and AdipoR2 causes abrogation of adiponectin binding and metabolic actions. *Nat Med* **13**, 332-339 (2007).
 27. Lee K, Goh GY, Wong MA, Klassen TL, Taubert S. Gain-of-Function Alleles in *Caenorhabditis elegans* Nuclear Hormone Receptor nhr-49 Are Functionally Distinct. *PLoS One* **11**, e0162708 (2016).
 28. Tavernarakis N, Wang SL, Dorovkov M, Ryazanov A, Driscoll M. Heritable and inducible genetic interference by double-stranded RNA encoded by transgenes. *Nat Genet* **24**, 180-183 (2000).
 29. Kamath RS, *et al.* Systematic functional analysis of the *Caenorhabditis elegans* genome using RNAi. *Nature* **421**, 231-237 (2003).
 30. Calixto A, Chelur D, Topalidou I, Chen X, Chalfie M. Enhanced neuronal RNAi in *C. elegans* using SID-1. *Nat Methods* **7**, 554-559 (2010).
 31. Bolten CJ, Kiefer P, Letisse F, Portais JC, Wittmann C. Sampling for metabolome analysis of microorganisms. *Anal Chem* **79**, 3843-3849 (2007).

32. Spura J, Reimer LC, Wieloch P, Schreiber K, Buchinger S, Schomburg D. A method for enzyme quenching in microbial metabolome analysis successfully applied to gram-positive and gram-negative bacteria and yeast. *Anal Biochem* **394**, 192-201 (2009).
33. Berdichevsky A, Viswanathan M, Horvitz HR, Guarente L. C. elegans SIR-2.1 interacts with 14-3-3 proteins to activate DAF-16 and extend life span. *Cell* **125**, 1165-1177 (2006).
34. Chen S, *et al.* The conserved NAD(H)-dependent corepressor CTBP-1 regulates *Caenorhabditis elegans* life span. *Proc Natl Acad Sci U S A* **106**, 1496-1501 (2009).
35. Wang Y, Tissenbaum HA. Overlapping and distinct functions for a *Caenorhabditis elegans* SIR2 and DAF-16/FOXO. *Mech Ageing Dev* **127**, 48-56 (2006).
36. Tullet JMA, *et al.* The SKN-1/Nrf2 transcription factor can protect against oxidative stress and increase lifespan in *C. elegans* by distinct mechanisms. *Aging Cell* **16**, 1191-1194 (2017).
37. Ferraz RC, *et al.* IMPACT is a GCN2 inhibitor that limits lifespan in *Caenorhabditis elegans*. *BMC Biol* **14**, 87 (2016).
38. Bishop NA, Guarente L. Two neurons mediate diet-restriction-induced longevity in *C. elegans*. *Nature* **447**, 545-549 (2007).
39. Steinbaugh MJ, *et al.* Lipid-mediated regulation of SKN-1/Nrf in response to germ cell absence. *Elife* **4**, (2015).
40. Tataridas-Pallas N, *et al.* Neuronal SKN-1B modulates nutritional signalling pathways and mitochondrial networks to control satiety. *PLoS Genet* **17**, e1009358 (2021).
41. Cooper JF, Machiela E, Dues DJ, Spielbauer KK, Senchuk MM, Van Raamsdonk JM. Activation of the mitochondrial unfolded protein response promotes longevity and dopamine neuron survival in Parkinson's disease models. *Sci Rep* **7**, 16441 (2017).
42. Palikaras K, Lionaki E, Tavernarakis N. Coordination of mitophagy and mitochondrial

- biogenesis during ageing in *C. elegans*. *Nature* **521**, 525-528 (2015).
43. Ryu D, *et al.* Urolithin A induces mitophagy and prolongs lifespan in *C. elegans* and increases muscle function in rodents. *Nat Med* **22**, 879-888 (2016).
 44. Taylor RC, Dillin A. XBP-1 is a cell-nonautonomous regulator of stress resistance and longevity. *Cell* **153**, 1435-1447 (2013).
 45. Schulz TJ, Zarse K, Voigt A, Urban N, Birringer M, Ristow M. Glucose restriction extends *Caenorhabditis elegans* life span by inducing mitochondrial respiration and increasing oxidative stress. *Cell Metab* **6**, 280-293 (2007).
 46. Brown G, *et al.* Structural and biochemical characterization of the type II fructose-1,6-bisphosphatase GlpX from *Escherichia coli*. *J Biol Chem* **284**, 3784-3792 (2009).
 47. Kuznetsova E, *et al.* Genome-wide analysis of substrate specificities of the *Escherichia coli* haloacid dehalogenase-like phosphatase family. *J Biol Chem* **281**, 36149-36161 (2006).
 48. Chao YP, Patnaik R, Roof WD, Young RF, Liao JC. Control of gluconeogenic growth by pps and pck in *Escherichia coli*. *J Bacteriol* **175**, 6939-6944 (1993).
 49. Chin RM, *et al.* The metabolite alpha-ketoglutarate extends lifespan by inhibiting ATP synthase and TOR. *Nature* **510**, 397-401 (2014).

Reviewers' Comments:

Reviewer #1:

Remarks to the Author:

The authors have adequately addressed my concerns either through modifying the manuscript/adding new data or in their answers to reviewer comments. This is a very interesting paper and I support its publication.

Reviewer #2:

Remarks to the Author:

The manuscript submitted by Jeong et al. is a revised version of an initial submission that I reviewed for Nature Communications over a year ago. The work described an extensive series of experiments that ultimately allowed the investigators to conclude that a novel isoform of AMPK plays a critical role in extending lifespan in response to glucose restricted diets in *C. elegans*. It reveals multiple novel and highly interesting new findings that put AMPK at a pole position in the neurons to control several aspects of organismal ageing; two of which include the regulation of an Adiponectin linked pathway in *C. elegans* and also the activation of PPAR α to control membrane fluidity through the increase in unsaturated fatty acid composition. It is a genetic and biochemical tour de force that is both comprehensive and exciting, opening up a number of new avenues that will be interesting for lipid biochemists and investigators committed to understanding healthy ageing.

The major issue that concerned me in the initial submission was the authors' dependence on an indirect GFP-based proxy as a means of assessing membrane fluidity. I suggested that these data be replaced with a more biophysics-based assay using C-Laurdan which they did and incorporated the new data in the revised version. All my other concerns were either experimentally addressed and included in the manuscript, or defused through description or explanation of the investigators' interpretation in the response to reviewers.

As a result I am completely satisfied with this work. I believe that it represents a refreshing step forward for the ageing field, which has become stale in recent years, lacking any fresh new discoveries that might reveal novel avenues that contribute to better healthspan, beyond the current series of lifespan extending paradigms.

There are, as always, some outstanding questions that arise from these findings, namely how paqr-2 mechanistically induces all these non-cell autonomous changes in the organism, and how this pathway ultimately contributes to the observed changes in overall membrane fluidity.

However, I am sure that experiments are currently ongoing to address these questions which will undoubtedly be a focus in the group's next manuscript. Overall, these exciting new questions only attest to the overall importance of the current submitted work. I really only have a few concerns that I have listed below, most of which are editorial in nature. In addition, there are a number of minor grammatical issues that I presume could be addressed by the authors following a solid proofread by a native English speaker.

Minor concerns:

The discussion of RNAi refractoriness in neurons and then the eventual neuron-specific RNAi experiments is contradictory (Lines 341-358). If indeed the neurons are refractory to RNAi, which I agree with, how does the neuron-specific aak-2 RNAi give a different result than that obtained with whole animal aak-2 RNAi? The RNAi is still taking place in the neurons. Although I realise that the genotype enhances the sensitivity of the neurons to RNAi, the text doesn't make sense that in one situation the neurons cannot support RNAi and in another situation they are fully capable of carrying out functional RNAi. The fundamental basis for this difference should be better rationalized, and the language in the lead up discussion should be modified to iron out the apparent contradiction.

There are a number of small grammatical issues that should be attended to. I have listed many of them below:

Line 27 onsets->onset

Line 68 it appears that clk-1 was not included in this list. Is that because you assume that clk-1

(mitochondrial paradigm) is working through AMPK?
Line 73 eukaryote->eukaryotes
Line 79 whole body metabolisms->metabolism
Line 98 regulations->regulation
Line 124 through a series of actions
Line 158 glycine and lysine accumulation increased by...
Line 214 could extend lifespan
Line 219-220 Taken altogether (the rest of this sentence does not really say anything...it is awkward as it currently reads)
Line 248 enabled->enables
Line 294 by introducing an icd mutation
Line 324 a beneficial effect
Line 385 demonstrating that the AAK-2a...
Lines 391-394 It was not clear to me if the authors could conclude that expression of the individual aak-2 isoforms was sufficient to extend WT lifespan if expression in multi-copy arrays. That information might be useful.
Lines 493 the idea that nhr-49 is required in the intestine and the neurons downstream of AMPK to mediate changes in organismal lipid metabolism is quite reminiscent of work published by Schmeisser et al. in Cell Reports.
Line 502 taken altogether,
Line 508 fat metabolism
Line 577 taken altogether,
Line 600 introducing the paqr-2 mutation...
Line 611 these data imply...
Line 630 certain circumstances...
Line 634 Studies on the longevity effects of GR diets are even less common
Line 660 These data imply that sbp...
Line 665 in a single tissue or alternatively in any one tissue... In its present state it is not at all clear what this is meant to convey to the reader.
Line 670-671 From the data I have seen it seems difficult to distinguish that AMPK is not acting through NHR-49 to bring about these changes. The current way that this is expressed suggests that AMPK and NHR-49 do not work together, but their activities are non-additive suggesting the contrary. This sentence should be reworded to correctly express the major conclusions of the work without any ambiguity.

Reviewer #3:

Remarks to the Author:

All points raised in the original review report have been carefully addressed by the authors, with additional material suitably provided to support their conclusions. I am ready to recommend publication of this article as an original contribution to Nature Communications.

Reviewer #4:

Remarks to the Author:

The authors have made major revisions to the manuscript, including addition of significant, new data. In short, the authors have responded to all my concerns. All the main conclusions of the manuscript are supported by evidence. Congratulations to the authors on a very solid and extensive study.

Dear Editor:

We appreciate the Reviewers' helpful comments and support publication. To address the issues raised by Reviewer #2, we revised manuscript and provided further explanations. Responses to the reviewer's comments are presented below.

We made point-by-point responses to reviewers' concerns. The reviewers' comments (in blue) and our responses (in **black, Bold**) are listed below.

Point-by-point responses.

Reviewers' comments

Reviewer #1 (Remarks to the Author):

The authors have adequately addressed my concerns either through modifying the manuscript/adding new data or in their answers to reviewer comments. This is a very interesting paper and I support its publication.

Reviewer #2 (Remarks to the Author):

The manuscript submitted by Jeong et al. is a revised version of an initial submission that I reviewed for Nature Communications over a year ago. The work described an extensive series of experiments that ultimately allowed the investigators to conclude that a novel isoform of AMPK plays a critical role in extending lifespan in response to glucose restricted diets in *C. elegans*. It reveals multiple novel and highly interesting new findings that put AMPK at a pole position in the neurons to control several aspects of organismal ageing; two of which include the regulation of an Adiponectin linked pathway in *C. elegans* and also the activation of PPAR α to control membrane fluidity through the increase in unsaturated fatty acid composition. It is a genetic and biochemical tour de force that is both comprehensive and exciting, opening up a number of new avenues that will be interesting for lipid biochemists and investigators committed to understanding healthy ageing.

The major issue that concerned me in the initial submission was the authors' dependence on an indirect GFP-based proxy as a means of assessing membrane fluidity. I suggested that these data

be replaced with a more biophysics-based assay using C-Laurdan which they did and incorporated the new data in the revised version. All my other concerns were either experimentally addressed and included in the manuscript, or defused through description or explanation of the investigators' interpretation in the response to reviewers.

As a result I am completely satisfied with this work. I believe that it represents a refreshing step forward for the ageing field, which has become stale in recent years, lacking any fresh new discoveries that might reveal novel avenues that contribute to better healthspan, beyond the current series of lifespan extending paradigms.

There are, as always, some outstanding questions that arise from these findings, namely how *paqr-2* mechanistically induces all these non-cell autonomous changes in the organism, and how this pathway ultimately contributes to the observed changes in overall membrane fluidity. However, I am sure that experiments are currently ongoing to address these questions which will undoubtedly be a focus in the group's next manuscript. Overall, these exciting new questions only attests to the overall importance of the current submitted work. I really only have a few concerns that I have listed below, most of which are editorial in nature. In addition, there are a number of minor grammatical issues that I presume could be addressed by the authors following a solid proofread by a native English speaker.

Minor concerns:

The discussion of RNAi refractoriness in neurons and then the eventual neuron-specific RNAi experiments is contradictory (Lines 341-358). If indeed the neurons are refractory to RNAi, which I agree with, how does the neuron-specific *aak-2* RNAi give a different result than that obtained with whole animal *aak-2* RNAi? The RNAi is still taking place in the neurons. Although I realise that the genotype enhances the sensitivity of the neurons to RNAi, the text doesn't make sense that in one situation the neurons cannot support RNAi and in another situation they are fully capable of carrying out functional RNAi. The fundamental basis for this difference should be better rationalized, and the language in the lead up discussion should be modified to iron out the apparent contradiction.

We appreciate this request for clarification. Once dsRNAs are uptaken and processed in the intestinal cells, they are transferred to other tissues in *C. elegans*. SID-1, RNA transmembrane transporter, is essential for the spread of dsRNA and subsequent silencing of genes in distal cells. Of note, *C. elegans* neuron is refractory to RNAi due to the lack of the *sid-1* expression in neurons. Therefore, to knock-down genes of interest in neurons, we utilized TU3401 transgenic animals which express the SID-1 in neurons; its genotype is *sid-1(pk3321); uls69 [pCFJ90 (myo-2p::mCherry) + unc-119p::sid-1]*, in which pCFJ90 (*myo-2p::mCherry*) is co-injection marker for pharyngeal RFP and *unc-119p* is used to neuronal expression of SID-1 (DOI:

10.1038/nmeth.1463). Accordingly, *aak-2* RNAi exhibits no effect on neuronal expression of *aak-2* in wild-type N2 backgrounds but knock-down the neuronal expression of *aak-2* in TU3401. For better understanding, we added the explanation of TU3401 in revised manuscript.

Line 52-54;

here are a number of small grammatical issues that should be attended to. I have listed many of them below:

Line 27 onsets->onset

Line 68 it appears that *clk-1* was not included in this list. Is that because you assume that *clk-1* (mitochondrial paradigm) is working through AMPK?

Line 73 eukaryote->eukaryotes

Line 79 whole body metabolisms->metabolism

Line 98 regulations->regulation

Line 124 through a series of actions

Line 158 glycine and lysine accumulation increased by...

Line 214 could extend lifespan

Line 219-220 Taken altogether (the rest of this sentence does not really say anything...it is awkward as it currently reads)

Line 248 enabled->enables

Line 294 by introducing an *icd* mutation

Line 324 a beneficial effect

Line 385 demonstrating that the *AAK-2a*...

Lines 391-394 It was not clear to me if the authors could conclude that expression of the individual *aak-2* isoforms was sufficient to extend WT lifespan if expression in multi-copy arrays. That information might be useful.

Lines 493 the idea that *nhr-49* is required in the intestine and the neurons downstream of AMPK to mediate changes in organismal lipid metabolism is quite reminiscent of work published by Schmeisser et al. in Cell Reports.

Line 502 taken altogether,

Line 508 fat metabolism

Line 577 taken altogether,

Line 600 introducing the *paqr-2* mutation...

Line 611 these data imply...

Line 630 certain circumstances...

Line 634 Studies on the longevity effects of GR diets are even less common

Line 660 These data imply that sbp...

We thank the reviewer for carefully reading the manuscript. As reviewer suggested, we revised the manuscript.

Line 665 in a single tissue or alternatively in any one tissue... In its present state it is not at all clear what this is meant to convey to the reader.

We appreciate this comment for clarification. As reviewer suggested, we revised the manuscript.

Line 670-671 From the data I have seen it seems difficult to distinguish that AMPK is not acting through NHR-49 to bring about these changes. The current way that this is expressed suggests that AMPK and NHR-49 do not work together, but their activities are non-additive suggesting the contrary. This sentence should be reworded to correctly express the major conclusions of the work without any ambiguity.

We appreciate this request for clarification. As reviewer mentioned, our data suggest that AMPK functions through NHR-49 to modulate longevity upon GR. We realized that Line 670-671 could cause misunderstanding that AMPK and NHR-49 function independently in GR-induced longevity. To prevent any possible misunderstanding, we revised the manuscript.

Line 670-671;

Reviewer #3 (Remarks to the Author):

All points raised in the original review report have been carefully addressed by the authors, with additional material suitably provided to support their conclusions. I am ready to recommend publication of this article as an original contribution to Nature Communications.

Reviewer #4 (Remarks to the Author):

The authors have made major revisions to the manuscript, including addition of significant, new

data. In short, the authors have responded to all my concerns. All the main conclusions of the manuscript are supported by evidence. Congratulations to the authors on a very solid and extensive study.s